# Temperature-dependent jumonji demethylase modulates flowering time by targeting H3K36me2/3 in *Brassica rapa*

Xiaoyun Xin[1,2,3,4], Peirong Li[1,2,3,4], Xiuyun Zhao[1,2,3,4], Yangjun Yu[1,2,3,4], Weihong Wang[1,2,3,4], Guihua Jin[1,2], Jiao Wang[1,2], Liling Sun[1,2], Deshuang Zhang[1,2,3,4], Fenglan Zhang ◉[1,2,3,4] ✉, Shuancang Yu ◉[1,2,3,4] ✉ & Tongbing Su ◉[1,2,3,4] ✉

Global warming has a severe impact on the flowering time and yield of crops. Histone modifications have been well-documented for their roles in enabling plant plasticity in ambient temperature. However, the factor modulating histone modifications and their involvement in habitat adaptation have remained elusive. In this study, through genome-wide pattern analysis and quantitative-trait-locus (QTL) mapping, we reveal that *BrJMJ18* is a candidate gene for a QTL regulating thermotolerance in thermotolerant *B. rapa* subsp. *chinensis* var. *parachinensis* (or Caixin, abbreviated to *Par*). BrJMJ18 encodes an H3K36me2/3 Jumonji demethylase that remodels H3K36 methylation across the genome. We demonstrate that the BrJMJ18 allele from *Par* (BrJMJ18[Par]) influences flowering time and plant growth in a temperature-dependent manner via characterizing overexpression and CRISPR/Cas9 mutant plants. We further show that overexpression of BrJMJ18[Par] can modulate the expression of *BrFLC3*, one of the five *BrFLC* orthologs. Furthermore, ChIP-seq and transcriptome data reveal that BrJMJ18[Par] can regulate chlorophyll biosynthesis under high temperatures. We also demonstrate that three amino acid mutations may account for function differences in BrJMJ18 between subspecies. Based on these findings, we propose a working model in which an H3K36me2/3 demethylase, while not affecting agronomic traits under normal conditions, can enhance resilience under heat stress in *Brassica rapa*.

Temperature plays a crucial role in determining the pace of plant development. Given the current scenario of climate change, rising temperatures can accelerate flowering and shorten developmental phases in crops. This may lead to significant reductions in agricultural yields, posing a widespread risk of food insecurity. Primary production needs to increase, and crops that use resources more efficiently and display increased resilience to unpredictable climatic events need to

be developed urgently. The use of genetically modified crops are largely limited; therefore, the use of natural genetic variations in plant breeding continues to underpin the improvement of all major crops. It's vital to uncover the mechanisms underlying the selection of genetic variants—both derived and preexisting—that enable crops to thrive in new and challenging environments. By comprehending these mechanisms, we can gather knowledge and germplasms essential for

[1]State Key Laboratory of Vegetable Biobreeding, Beijing Vegetable Research Center, Beijing Academy of Agriculture and Forestry Science, Beijing, China. [2]National Engineering Research Center for Vegetables, Beijing Vegetable Research Center, Beijing Academy of Agriculture and Forestry Science, Beijing, China. [3]Beijing Key Laboratory of Vegetable Germplasms Improvement, Beijing, China. [4]Key Laboratory of Biology and Genetics Improvement of Horticultural Crops (North China), Beijing, China. ✉e-mail: zhangfenglan@nercv.org; yushuancang@nercv.org; sutongbing@nercv.org

crafting resilient crop varieties capable of confronting the ongoing threats posed by global warming.

*Brassica rapa* L. (*B. rapa*) is an important oilseed and vegetable crop and is the subgenome donor of two other important *Brassica* crops, *Brassica napus* and *Brassica juncea*. *B. rapa* demonstrates remarkable adaptability, thriving across diverse habitats spanning from sea level to elevations exceeding 4500 meters and also from cold to tropical areas. All these subspecies or variants occur over a wide range of ecological conditions, and some individual species have adapted to particular microclimates to meet human needs or their survival[1,2]. Plant flowering time is an important life history trait underlying reproductive fitness and is sensitive to local growing conditions. Wide variation in flowering time exists between and within *B. rapa* morphotypes; therefore, the widely cultivated *B. rapa* crops are thus ideal objects to illustrate the diversity that can be created by domestication or/and breeding[3,4]. Among these morphotypes, *B. rapa* subsp. *chinensis* var. *parachinensis* (abbreviated to *Par*) is an understudied member within the realm of *B. rapa*. *Par* is a vegetable primarily valued for its edible components, encompassing the stems, stem leaves, and terminal inflorescences, which are commonly consumed in Asia. The *Par* variety displays two distinctive domestication traits not found in other leafy *B. rapa* crops. Firstly, it exhibits vernalization-independent flowering, in alignment with its suitability as a year-round vegetable. Secondly, building upon the first characteristic, *Par* was further domesticated to maintain stable flowering time under the warm conditions in southern China. This stability is crucial for preserving the quality and yield of the commercial organ of *Par* since flowering at inappropriate times can impact them. In other words, the recent domestication of *Par* primarily focuses on its flowering time adaptation to high temperatures, however, the procedure and genetic basis of this stepwise domestication is unknown.

High temperatures can impact the flowering time of *B. rapa* in various ways. The timing of the heat stress is crucial since its impact varies during the vegetative growth period and the reproductive growth period[5]. Here, we would like to discuss the effect of high temperatures on the flowering time of *B. rapa* plants during their vegetative growth. Del Olmo et al. [6] studied *R-o-18*, an oilseed *rapa* variety, and found that plants grown at 28 °C flowered later than those at 21 °C[6]. While some studies on vegetative *B. rapa*, like summer-planting Chinese cabbage and non-heading Chinese cabbage, have shown that high-temperature exposure during the seedling stage leads to premature bolting, affecting yield and quality[7,8]. Some *Par* varieties, as reported by Lu (2022), can start flowering prematurely when temperatures exceed 30 °C, leading to stunted growth and slow development, significantly affecting their production[9]. Rameeh (2012) investigated the effects of high temperature on the flowering time of Indian mustard (*B. juncea*)[10], revealing that late-planted Indian mustard experienced early flowering due to heat stress.

The definitions of 'thermotolerance' and 'heat stress' are influenced by the plant's optimal range and the temperature conditions in its cultivation region. As reported by Morrison and Stewart[11], during the flowering stage, the critical threshold temperature causing seed yield losses for all *Brassica* species were found to be 29.5 °C and mean maximum temperatures of more than 29 °C during vegetative development led to a decline in flower numbers across all *Brassica* species. We also obtained meteorological data from the China Meteorological Administration regarding the average summer temperatures in Guangdong province, the region where *Par* domestication predominantly occurs. These temperatures have exhibited an average of 29.4 °C over the past half-century. By integrating this information, we propose that the approximate threshold for *Par* domestication can be established at around 29 °C.

The present study used var. *parachinensis*, which has adapted to gain plasticity of flowering time and growth duration under hot conditions, to explore the genetic variations recorded in its genome by sequencing a collection of different *B. rapa* subspecies and determined the genomic signatures that underlie *B. rapa's* improved high-temperature responses. We demonstrated a stepwise model for the speciation of subsp. *chinensis* var. *parachinensis* during domestication, and further identified *BrJMJ18^{Par}* (the *BrJMJ18* allele from *Par*) as a candidate gene for a QTL regulating thermotolerance in var. *parachinensis*. Through investigations involving overexpression and CRISPR/Cas9 mutant plants, we demonstrated that BrJMJ18^{Par} plays a pivotal role in mediating flowering time and plant growth in response to temperature fluctuations. Overexpression of *BrJMJ18^{Par}* can adjust the expression of *BrFLC3* and flowering time, while the outcome of *BrFLC3* is less than direct in the WT *Par* plants. These findings not only advance our understanding of speciation during domestication but also offer insights and germplasm resources for developing resilient *B. rapa* crop varieties in response to the challenges of global warming.

## Results

### A stepwise domestication model of *Brassica rapa* subsp. *chinensis* var. *parachinensis* (*Par*)

The germplasm collection used in this study included a new set of 169 accessions and a previous set of 41 subsp. *rapifera* germplasms[12] from all over the world (Supplementary Fig. 1a and Supplementary Table 1). We sequenced the genomes of the plants in the new set, resulting in a sequencing dataset from a total of 210 varieties of six *Brassica rapa* morphotypes, including 15 subsp. *oleifera* (*Ole*), 51 subsp. *rapifera* (*Raf*), 9 subsp. *chinensis var. narinosa* (*Nar*), 50 subsp. *chinensis* (*Pak-Choi*, *PC*), 41 subsp. *chinensis DG* (*Dark Green*, *DG*), and 44 subsp. *chinensis var. parachinensis* (*Par*). The dataset consisted of 2.9 Tb of 125 bp paired-end reads, with more than 10 × coverage on average for each sample. Using the *B. rapa* genome v3.0 as reference genome (downloaded from BRAD http://brassicadb.cn/), we obtained a total of 2,690,680 and 210,664 high-quality single nucleotide polymorphisms (SNPs) and insertions/deletions (InDels; Table 1), respectively.

We then determined the phylogenetic relationships of the 210 lines. Analyses of the genetic structure recovered seven clusters using S$_{TRUCTURE}$ (Fig. 1a and Supplementary Fig. 1b). The results showed that *Par* was positioned at the most distant point from the ancient *Raf* group, while *PC* was closest to the root and *DG* was found at an intermediate position between *Par* and *PC* (Fig. 1a and Supplementary Fig. 1c, d). In the PCA plot, the transitional feature of *DG* between *Par* and *PC* was more evident (Supplementary Fig. 1c). To further understand the evolutionary history of *Par*, we used ∂a∂i analysis to estimate the demographic modeling of *PC*, *DG*, and *Par* (Fig. 1b). The best model supported an initial common ancestor split from *PC* followed by a subsequent divergence into *DG* and *Par*. The time span from the initial divergence of *PC* to the split of *PC* was approximately 1220–2038 years, and the divergence of *DG* and *Par* occurred approximately 1838–3050 years ago (Fig. 1b and Supplementary Table 2). Besides, the data of gene flow evaluation and nucleotide diversity divergence of the three groups further revealed a transitional feature of *DG* during *Par* evolution (Fig. 1c and Supplementary Fig. 2).

In terms of phenotypic analyses, flowering time varied greatly among the three groups (Fig. 1d). *Par* was the only subspecies that could bloom without vernalization; however, three accessions of *DG* could still flower without cold, suggesting that some early-flowering genetic components have started to deposit in *DG* during selection (Fig. 1d, the left panel). A 4-week vernalization was then imposed on the three subspecies to determine flowering-time variations. We found that *PC* exhibited the longest bolting duration, *Par* bloomed earliest, while *DG* fell in between (Fig. 1d, the right panel). With regard to other morphological features, the three subspecies resemble each other at the seedling stage (Supplementary Fig. 3, upper panel); while at the

**Table 1 | General information on genetic variation in the 210 *B. rapa* lines**

|  |  | all | *Raf* | *Ole* | *PC+Nar* | *DG* | *Par* |
|---|---|---|---|---|---|---|---|
| Sample Size |  | 210 | 51 | 15 | 59 | 41 | 15 |
| Variants | SNP | 2,690,680 | 2,237,437 | 2,026,926 | 1,808,392 | 1,750,704 | 1,371,766 |
|  | Unique SNP[a] | NA | 142,903 | 11,606 | 25,649 | 8,434 | 7,737 |
|  | InDel | 210,664 | 169,012 | 149,831 | 1,921,857 | 135,267 | 104,795 |
|  | Unique InDel[a] | NA | 12366 | 880 | 2504 | 833 | 778 |
|  | PI[b] | 0.002507 | 0.002329 | 0.001788 | 0.002044 | 0.001896 | 0.001524 |
|  | taijima D | 3.172689 | 2.120287 | 0.783791 | 1.623865 | 1.533449 | 1.201749 |
| LD | LD[c] 0.2 | 1.72 | 0.63 | 120.4 | 6.85 | 11.8 | 37.8 |
|  | LD half | 0.11 | 0.05 | 0.19 | 0.5 | 0.98 | 2.84 |

[a]Unique single nucleotide polymorphisms (SNPs) and insertion/deletions (InDels) are variations specific to each group.
[b]PI, nucleotide diversity within each group. PI was calculated based on the genotypes of each line at the SNP positions using BioPerl.
[c]LD (linkage disequilibrium) blocks were calculated based on SNPs with minor allele frequency (MAF) greater than 0.05 using Haploview software. LD decay was calculated based on the squared correlation coefficient ($r^2$ = 0.2 or 0.5) by pairwise physical distance between SNPs.

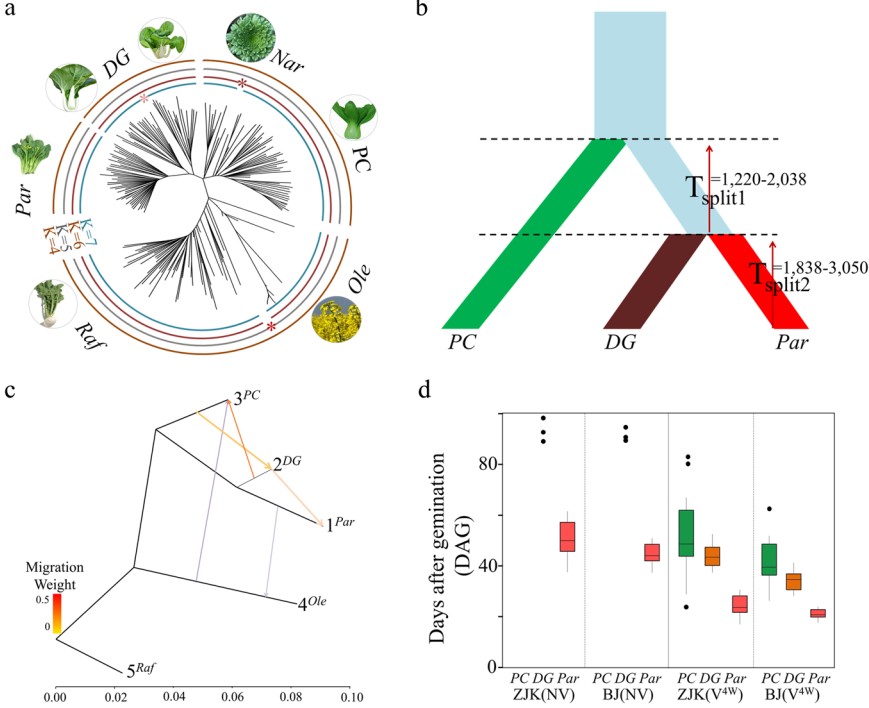

**Fig. 1 | A stepwise model for speciation of subsp. *chinensis* var. *parachinensis* (*Par*). a** Neighbor-joining (N-J) tree of the 210 lines. An unrooted phylogenetic tree of the 210 lines was constructed using the N-J method under the Kimura 2-parameter model implemented in MEGA-CC (v 7.0). Bootstrap support > 50% calculated from 1000 replicates is shown. **b** The best demographic model evaluated using ∂a∂i. At time $T_{split}$. The time of the split was estimated in the generation time, which was converted to years, assuming one generation per year. We used $S_{TRUCTURE}$ $K$ = 6 as the likelihood ratio, which can also satisfy the requirements of both $S_{TRUCTURE}$ and empirical classification. **c** Inferred *B. rapa* tree showing the directions of gene flow in each group. Arrows indicate the direction of gene flow, while the line colors represent the migration weight based on the sample number. The horizontal branch length is proportional to the amount of genetic drift that has occurred on the branch. Scale bars represent 100-fold average standard error (SE) for the entries in the sample covariance matrix. **d** Population phenotypic differentiation for the bolting time of *PC*, *DG*, and *Par* in Zhangjiakou (ZJK) and Beijing (BJ), respectively. Flowering time of 50 *PC*, 41 *DG* and 44 *Par* accessions from our germplasm collection were investigated. The flowering time was evaluated with (denoted as V^4w) or without (denoted as NV) 4 weeks vernalization in ZJK and BJ, respectively. Bolting time, days after germination (DAG), was defined as the number of days from sowing to the appearance of the visible bud. The box encompasses two middle quartiles, with a central line showing the median. Whiskers extend to the furthest data point within 1.5 times the interquartile range. *Par* represents *B. rapa* subsp. *chinensis* var. *parachinensis*, *DG* represents *B. rapa* subsp. *chinensis* var. *Dark-green*, *PC* represents *B. rapa* subsp. *chinensis* (pak choi).

adult juvenile stage, *PC* and *DG* share similar plant architecture, and *DG* and *Par* have similar the leaf shapes and color (Supplementary Fig. 3, bottom panel). All these molecular and phenotypic findings suggest a stepwise selection for *Par*'s evolution: *DG* originated from a particular *PC* population, and *Par* subsequently diverged from *DG* through the enrichment of early-flowering mutations and other adapted traits for local conditions.

## Genetic basis of subsp. *chinensis* var. *parachinensis* domestication identified via Coupling genome-wide pattern analysis and quantitative-trait-locus (QTL) mapping

We divided *Par*'s speciation into two stages: *DG/PC* (from *PC* to *DG*) and *Par/DG* (from *DG* to *Par*). The close level of nucleotide diversity (PI) of the three groups (Supplementary Fig. 2b) indicated very weak bottlenecks during selection. However, for the second step, due to

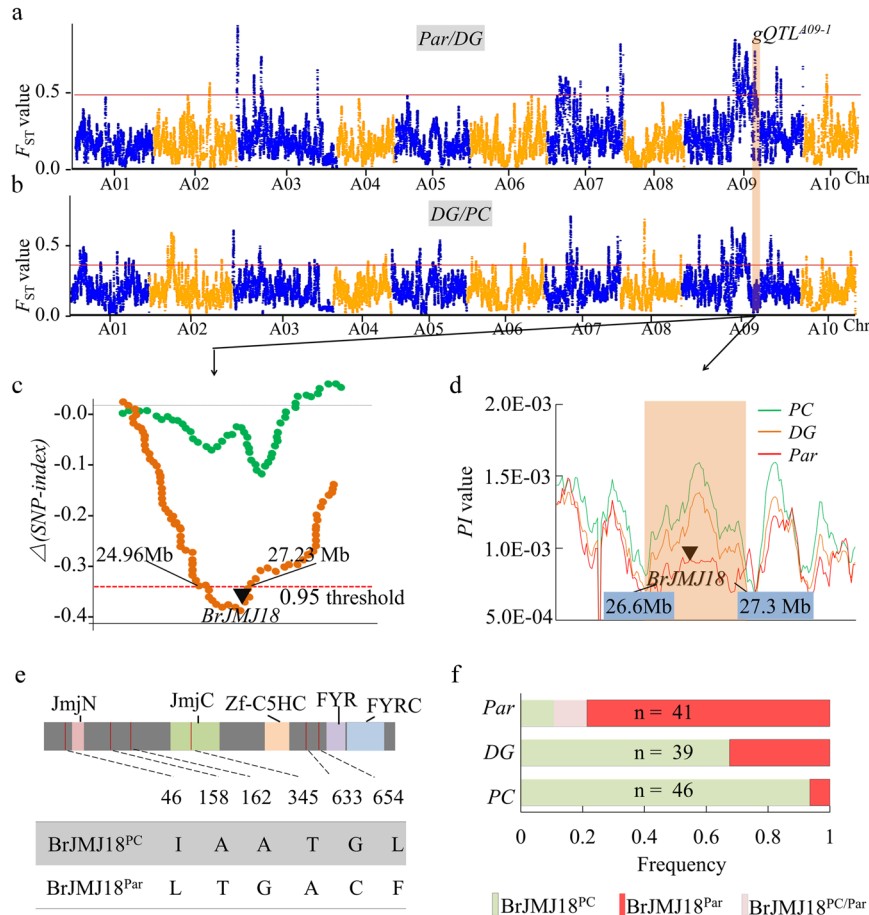

**Fig. 2 | Genetic basis of subsp. *chinensis* var. *parachinensis* domestication.**
**a**–**b** Genetic basis of subsp. *chinensis* var. *parachinensis* selection. (**a**) and (**b**) show highly divergent regions between *Par* and *DG*, and *DG* and *PC*, respectively. The $F_{ST}$ values are plotted against the position on each chromosome. The horizontal pink lines indicate the genome-wide thresholds (5%) of differentiation signals. The peach bar on Chr. A09 intersecting (**a**) and (**b**) indicates the candidate selection sweep (A09: 26,630,001...27,300,000) studied. **c** A temperature-responsive *QTL(HT)* co-locates with the region of *gQTL^{A09-1}*. An F2 population established from a cross between *Par* and *PC*, was used for the analysis. Half of the 4-week-old seedlings continued to grow under normal conditions (NC), while the other half of the seedlings were moved to high temperature (HT) conditions until flowering. Flowering-related quantitative trait loci (*ftQTLs*) responding to different temperatures were then identified. The green and brown dot lines indicate *QTLs* responding to NC and HT, respectively. The horizontal red line indicates the 95% confidence

interval of *Δ*(SNP-index). *gQTL* represents *QTLs* (quantitative trait locus) associating with *Par* domestication generated from (**a**). Black triangle, the genome position of *BrJMJ18*. **d** The local π value features for the selection sweep on chromosome A09 (26,630,001...27,300,000) of *PC*, *DG*, and *Par* around *gQTL^{A09-1}*. Black triangle, the genome position of *BrJMJ18*. **e, f** Sequence polymorphisms and haplotype analysis of *BrJMJ18* in *PC*, *DG*, and *Par*. Red lines in (**e**) indicate different nucleotide alterations and corresponding amino acid substitutions are shown. JmjN: Jumonji N-terminal domain; JmjC: Jumonji C-terminal domain; Zf-C5HC2: C5HC2 zinc finger domain; FYRN: FY-rich domain, N-terminal region; FYRC: FY-rich domain, C-terminal region. The alteration of allelic frequencies of *BrJMJ18* in *PC*, *DG*, and *Par* are shown in (**f**). *BrJMJ18^{PC}* and *BrJMJ18^{Par}* represent the *BrJMJ18* allele coming from *PC* and *Par*, respectively; n denotes the number of lines of *PC*, *DG*, and *Par* used for the analysis.

significant flowering time variation yet low PI value between *DG* and *Par*, we speculated a mild but precise selection in *Par*'s speciation. We identified candidate genomic regions selected during each step. Calculation of the $F_{ST}$ (5% threshold) detected 85 and 51 selected regions of the first and second steps, respectively. The selected regions in *DG/PC* and *Par/DG* span 21.0 Mb (4.4% of the genome) and 25.8 Mb (5.4%), harboring 4632 and 3089 genes, respectively (Fig. 2a, b and Supplementary Data 1). Among these selective genes, 1964 (42.4%) and 1184 (38.2%) showed differential expression during floral transition according to the RNA seq data (Supplementary Data 1). Gene Ontology (GO) analyses[13] revealed 171 specific ($p < 0.05$) enriched GO terms in *DG/PC* comparison, whereas only 45 terms were enriched in *Par/DG* comparison (Supplementary Data 2), confirming a precise selection during the second step.

Given *Par*'s domestication focus on stable flowering time under warm temperatures, we expected the enrichment of genes related to flowering time and temperature response in its selection. Compared

with *PC* and *DG*, the higher thermotolerance of *Par* was preliminarily phenotypically confirmed (Supplementary Fig. 4). Besides, the flowering time of *Par* was also less sensitive to cultivation environment change (Fig. 1d and Supplementary Fig. 5). We then recalled all the GO terms associated with reproduction and abiotic stress, and confirmed that these terms were specifically ($p < 0.05$) enriched in the *Par/DG* comparison but not, or less frequently, in the *DG/PC* comparison (Supplementary Fig. 6a). To define *Par*-speciation-related genes, we concentrated on loci solely distinguishing *Par* from *DG*, rather than *DG* from *PC*. This yielded 24 loci and 964 candidate genes (Supplementary Fig. 6b and Supplementary Data 3), including 373 differentially expressed genes (DEGs). To further narrow down target genes, we phenotyped *Par × PC* F2 population for flowering time under normal (22/22 °C, 16/8 h; hereafter shortened as NC) and high temperature (29/29 °C, 16/8 h; hereafter shortened as HT) chamber conditions and used bulked segregant analysis sequencing (BSA-seq) approach for mapping heat-responsive QTLs. A total of 13 HT-associated

quantitative trait loci (*QTL(HT)s*), solely responding to HT, were then identified (Supplementary Data 4).

## BrJMJ18 is a candidate gene for a QTL regulating thermo-tolerance in *Par*

We concentrated in a selection sweep on chromosome A09 (26,630,001...27,300,000, designated as *gQTL^{A09-1}*, marked in Fig. 2a–d), because 1) the flanking region of *gQTL^{A09-1}* did not co-locate within any of the reported flowering time (ft) *ftQTLs* in *B. rapa*, and 2) it overlapped with the region of one of the *QTL(HT)s* (Fig. 2d). A total of 77 genes, the expression of 18 of which, including 8 DEGs, were detectable during flowering, were mapped to the *gQTL^{A09-1}* region (Supplementary Data 3). Most of the 77 genes were independent of floral transition, except for three Jumonji C (JMJC) domain-containing proteins (Supplementary Data 3). In *Arabidopsis*, ten of the 21 *JMJ* genes are experimentally confirmed to affect flowering time, and fourteen of them respond to short-term heat stress (Supplementary Fig. 7). We then found that *BrJMJ18* (*BraA09g034190.3 C*) was the most highly expressed *BrJMJ*, while the other two were undetectable; and *BrJMJ18* was differently expressed during floral transition in *DG* and *Par*, but not *PC*, under NC conditions (Supplementary Data 5). Therefore, we proposed *BrJMJ18* as the best candidate gene contributing to the speciation of *Par* in *gQTL^{A09-1}*.

We then used the sequenced genomes of 58 *PC*, 41 *DG*, and 44 *Par* accessions (Supplementary Table 1) to scan for selection signatures surrounding *BrJMJ18*. A sharp reduction of nucleotide diversity was found between *Par* and *DG* at the *BrJMJ18* locus, while by contrast, no significant reduction was found between *DG* and *PC* (Fig. 2d). In addition, *BrJMJ18* was found to be located in a linkage disequilibrium (LD) block of *Par/DG* (Supplementary Fig. 8). Furthermore, we conducted a haplotype analysis of *BrJMJ18* in *PC*, *DG*, and *Par*. A total of 71 DNA polymorphisms, comprising 13 in the promoter and 26 in the exons, were found in the gene body of *BrJMJ18*. Since no significant differences of the expression pattern of *BrJMJ18* were found during flowering (Supplementary Data 5) and in the response to high temperature (Supplementary Fig. 9) between *DG* and *Par*, therefore, we only used the six nonsynonymous SNPs, resulting in six amino acid substitutions of BrJMJ18, for haplotype classification (Fig. 2e). A total of 126 of the 135 genotypes were then classified into two major haplotype groups: *BrJMJ18^{PC}* and *BrJMJ18^{Par}* (*BrJMJ18^{PC}* is the prominent haplotype in *PC* and *DG*, while *BrJMJ18^{Par}* mainly existed in *Par*) (Fig. 2f). The frequency of *BrJMJ18^{Par}* was very low (6.5%, *n* = 46) in *PC*, and increased to 23.1% (*n* = 39) in *DG*, and finally expanded to 90.2% (*n* = 41) in *Par* (Fig. 2f). We further noticed that *BrJMJ18^{PC}* and *BrJMJ18^{Par}* were evenly represented in the ancient subsp. *rapifera* and subsp. *oleifera* groups, respectively (Supplementary Fig. 10). These observations suggested that the variations at *BrJMJ18^{Par}* were not important for early *B. rapa* speciation, but conferred advantages to subsequent speciation or/and local adaptation.

## Overexpression of *BrJMJ18^{Par}* delays flowering under both greenhouse and field high temperature conditions

The effect of *BrJMJ18^{PC}* and *BrJMJ18^{Par}* on thermotolerance was then preliminary confirmed in the natural *DG* and *Par* group and the "*Par* × *PC*" F2 population mentioned above under NC and HT conditions, respectively, expressing as high temperature exerted a stronger effect of flowering on *BrJMJ18^{PC}*-carrying lines than on *BrJMJ18^{Par}*-carrying lines (Supplementary Fig. 11).

To characterize *BrJMJ18^{PC}* and *BrJMJ18^{Par}*, *BrJMJ18^{PC}*- and *BrJMJ18^{Par}*-OX plants of *Par* grown in a greenhouse were used for analysis (Fig. 3). The open reading frames (ORFs) of *BrJMJ18^{PC}* and *BrJMJ18^{Par}* driven by the cauliflower mosaic virus *35 S* promoter (*35 S*), respectively, were transformed into *Par* plants, Transgenic T1 lines with similar protein expressions were used for study (Fig. 3d). Under NC, both *BrJMJ18^{PC}*- and *BrJMJ18^{Par}*-OX plants bolted earlier than the *Par* controls (Fig. 3a, c).

While under HT, high temperature significantly delayed bolting in *BrJMJ18^{Par}*-OX, but did not affect bolting in *BrJMJ18^{PC}*-OX plants (Fig. 3b, c). To further characterize *BrJMJ18^{Par}*, the *BrJMJ18* knockout lines of *Par* plants, denoted as *BrJMJ18^{Par}*-CR, generated by Clustered regularly interspaced palindromic repeats (CRISPR)/CRISPR-associated protein 9 (Cas9), were used for analysis (Fig. 3e–h). Loss of *BrJMJ18^{Par}* induced late flowering under both NC and HT conditions, and the flowering time delay induced by high temperature was also significantly broadened in *BrJMJ18^{Par}*-CR plants, which is surprisingly similar to the changing trend of *BrJMJ18^{Par}*-OX plants (*BrJMJ18^{Par}*-CR did not bolt in 120 days under HT, and DAS was then set as 120). We also analyzed the flowering time of *BrJMJ18* transgenic lines (*AtJMJ18::BrJMJ18^{orf}*-GFP) in *Arabidopsis* (Supplementary Fig. 12), and found that under HT, the *AtJMJ18::BrJMJ18^{Par}*-GFP plants flowered much later than the *AtJMJ18::BrJMJ18^{PC}*-GFP plants.

We then planted *Par*, *BrJMJ18^{PC}*- and *BrJMJ18^{Par}*-OX plants in a tunnel greenhouse in the summer (15^{th}, Jul. … 29^{th}, Sept. 2021) to simulate extreme field high temperature conditions for further flowering time characterization. In the tunnel, daytime temperatures reached up to 52 °C and dropped to a low of 31 °C, while nighttime temperatures ranged from 33 °C to 24 °C. We found that *BrJMJ18^{Par}*-OX plants flowered later than controls, in line with the changing direction in the greenhouse under HT (Fig. 3i, j). Besides, the *BrJMJ18^{Par}*-OX plants demonstrated superior commercial quality compared with other plants (Fig. 3i).

All the above results established that both alleles of BrJMJ18 promote flowering under NC, but BrJMJ18^{Par} evolved to obtain the function of delaying flowering under HT conditions. Moreover, because of the consistent flowering phenotype of *BrJMJ18^{Par}*-OX and *BrJMJ18^{Par}*-CR *Par* plants under high-temperature conditions, we hypothesized that *JMJ18^{Par}* could be deactivated by high temperature, which indicated that BrJMJ18^{Par} functions in flowering time control in a temperature-dependent manner.

## BrJMJ18 is a histone H3 lysine 36 demethylase

BrJMJ18 was classified in the clade of the H3K4me3/2 demethylase family, with the highest homology (87%) at the amino acid level to *Arabidopsis* AtJMJ18 (Supplementary Fig. 13). AtJMJ18 displayed demethylase activity toward H3K4me3/2[14]. We then employed affinity-purified His-BrJMJ18^{PC/Par} proteins and synthesized histone H3 peptides containing specific modifications, including H3K4me3, H3K9me3, H3K27me3, and H3K36me2/3, to perform demethylase activity assays. The His-BrJMJ18 protein demonstrated efficient demethylation of H3K36me2/3 peptides while keeping the methylation levels of H3K4me3, H3K9me3, and H3K27me3 peptides unchanged; and there was no difference in demethylation activity between the BrJMJ18^{PC} and BrJMJ18^{Par} proteins at H3K36me3/2 (Fig. 4a and Supplementary Fig. 14a). We next immunoaffinity-purified the two allelic BrJMJ18-GFP proteins from *BrJMJ18* transgenic *Arabidopsis* lines (*AtJMJ18::BrJMJ18^{orf}*-GFP) and conducted in vitro demethylase analysis. Both allelic BrJMJ18-GFP proteins demethylated H3K36me3 and H3K36me2, but not at H3K4me3/2, H3K9me3/2, and H3K27me3/2 (Fig. 4b and Supplementary Fig. 14b). Finally, to confirm whether BrJMJ18^{PC} and BrJMJ18^{Par} can exert this demethylase activity in vivo, we transiently expressed BrJMJ18^{PC} and BrJMJ18^{Par} proteins (*35 S::BrJMJ18^{orf}*-GFP) in tobacco leaves. We showed that H3K36me3/2 underwent a dramatic global reduction in both *BrJMJ18^{PC}* and *BrJMJ18^{Par}* overexpression leaves under NC (Fig. 4c); meanwhile, we noticed that high temperature led to stronger H3K36me3/2 reduction in *BrJMJ18^{PC}*-OX than in *BrJMJ18^{Par}*-OX leaves (Fig. 4c). We used *BrJMJ18^{Par}*-CR plants for further analysis and observed that signals at H3K36me3/2 were markedly increased in the *BrJMJ18^{Par}*-CR plants (Fig. 4d).

Nuclear localization is a necessary context to enable a Jumonji protein to conduct its function. The two allelic BrJMJ18

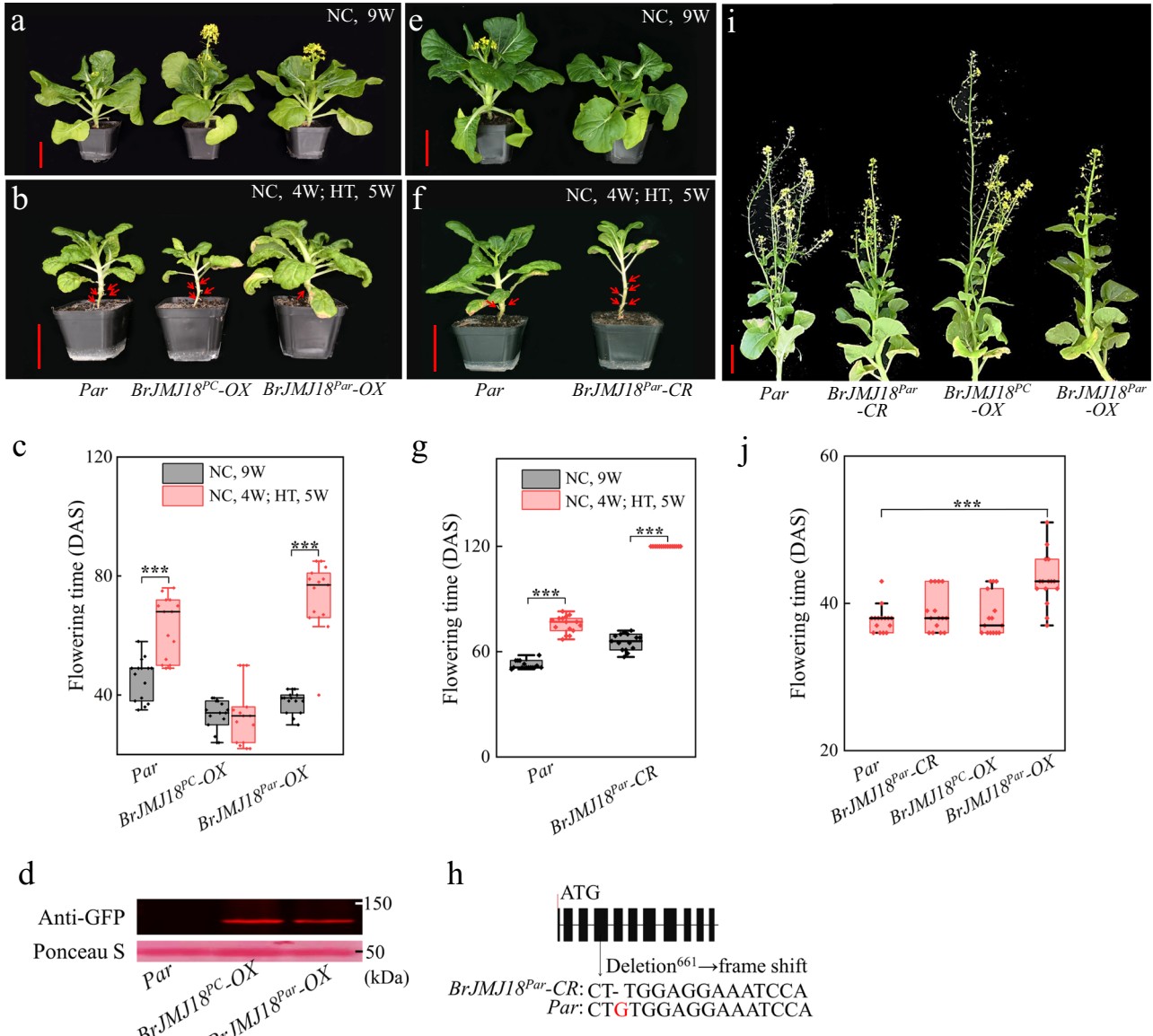

**Fig. 3 | Flowering characterizations of the *BrJMJ18* transgenic *Par* plants under greenhouse and field conditions, respectively. a–h** Phenotypes of the *BrJMJ18* transgenic *Par* plants grown in the greenhouse under different temperature conditions. **a** *BrJMJ18* transgenic *Par* plants grown under normal conditions (NC) for 9 weeks. Scale bar, 5 cm. **b** Phenotypes of the *BrJMJ18* transgenic *Par* plants grown under NC conditions for 4 weeks, following another 5 weeks under high temperature (HT) conditions. Scale bar, 5 cm. Red arrows, withered leaves. **c** Flowering time of plants is shown in (**a**) and (**b**). DAS, days after sowing. Data are means ± SD, $n = 15$. Asterisks indicate significant differences between NC and HT, two-tailed Student's t-test (***$p < 0.001$) *Par*, $p = 2.53 \times 10^{-5}$; *BrJMJ18$^{Par}$-OX*, $p = 9.28 \times 10^{-9}$. **d** Confirmation of BrJMJ18 protein expression in the transgenic plants by immunoblotting analysis. Ponceau S staining was used to assess equal loading. **e, f** Flowering characterizations of *BrJMJ18* knockout *Par* line, *BrJMJ18$^{Par}$-CR*, in greenhouse under different temperatures. **g** Flowering time of plants shown in (**e**) and (**f**). Red arrows, withered leaves. Data are means ± SD, $n = 15$. Asterisks indicate significant differences between NC and HT, two-tailed Student's t-test (***$p < 0.001$) *Par*, $p = 2.32 \times 10^{-10}$; *BrJMJ18$^{Par}$-CR*, $p = 1.47 \times 10^{-16}$. The experiments in (**a–f**) were repeated three times with similar results. **h** Confirmation of *BrJMJ18$^{Par}$-CR* by Sanger sequencing. A single base pair deletion (G → /) at the + 661 position after the ATG leads to a frame-shift mutation of the BrJMJ18 protein. **i, j** *Par*, *BrJMJ18$^{PC}$-* and *BrJMJ18$^{Par}$-OX* plants were planted in a tunnel greenhouse under natural field conditions in the summer (15th, Jul. … 29th, Sept.) of 2021 to simulate extremely high-temperature conditions for flowering time characterization. Data are means ± SD, $n = 20$. Asterisks indicate significant differences between *Par* and *BrJMJ18$^{Par}$-OX*, two-tailed Student's t-test (***$p < 0.001$), $p = 8.19 \times 10^{-5}$. For box plots (**c, g**, and **j**), the box encompasses two middle quartiles, with a central line showing the median. Whiskers extend to the furthest data point within 1.5 times the interquartile range. Source data are provided as a Source Data file.

proteins localized to the nucleus in transiently expressed tobacco leaves (Fig. 4e, f). Moreover, we revealed that the nuclear localization of BrJMJ18$^{PC}$ or BrJMJ18$^{Par}$ were not affected by HT (Fig. 4e, f). Taken together, our results showed that BrJMJ18 is a nuclear H3K36me3/2 demethylase under different temperatures; however, the activity amplitude between different temperatures is weaker in *BrJMJ18$^{Par}$*-expressing plants than that of *BrJMJ18$^{PC}$*-expressing plants.

## Overexpression of *BrJMJ18$^{Par}$* alters the expression of *BrFLC3* and modulates flowering

To assess the downstream genes of *BrJMJ18*, we conducted a chromatin immunoprecipitation (ChIP)-seq experiment using a polyclonal antibody recognizing BrJMJ18 in *Par* plants under NC (the specificity of the antibody was shown in Supplementary Fig. 15). We identified 6456 genes (12,547 loci) as candidate targets of BrJMJ18 (Supplementary Data 6). *FLOWERING LOCUS C* (*BrFLC*) family genes,

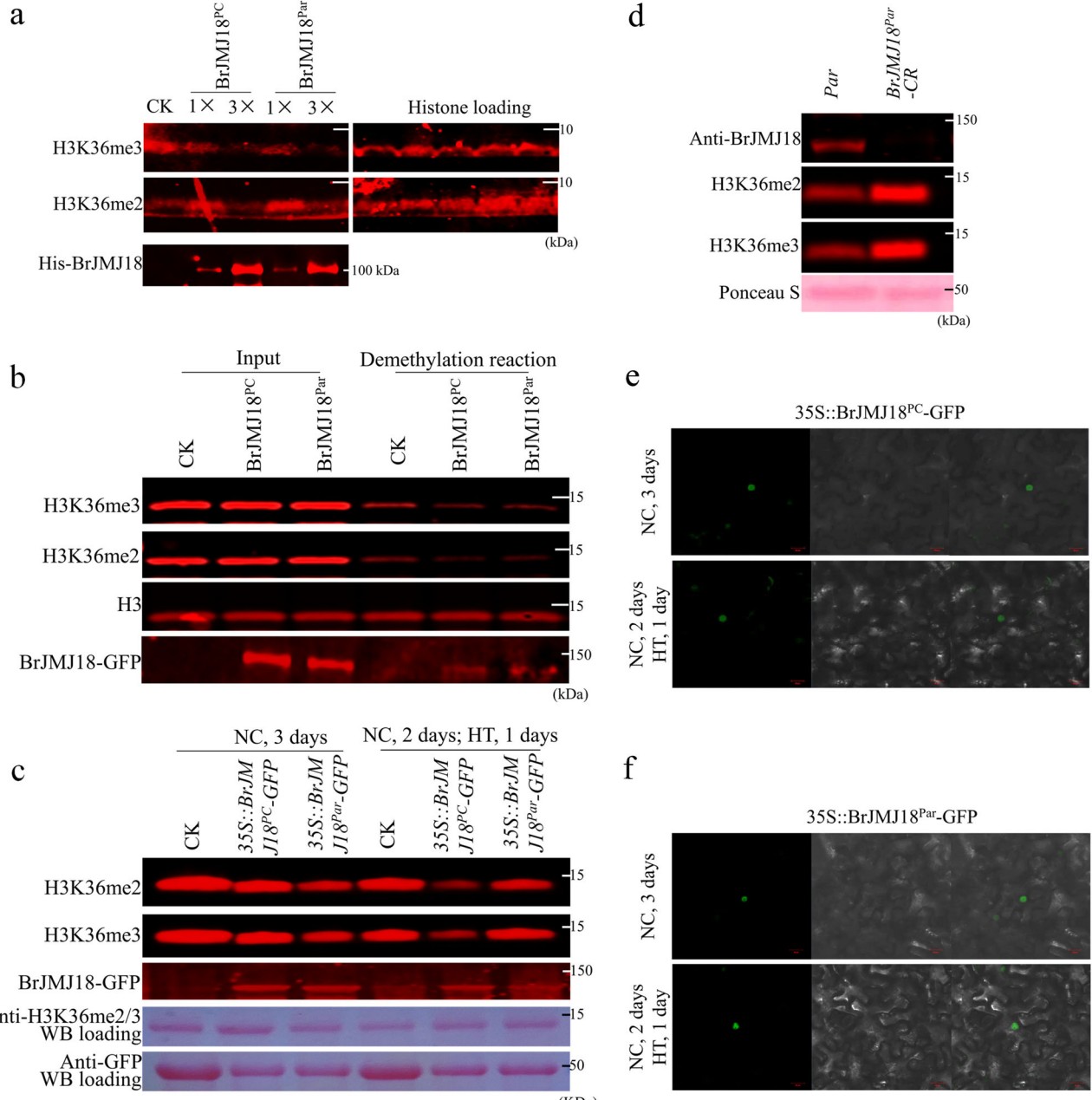

**Fig. 4 | BrJMJ18 is an H3K36me2/3 demethylase. a–d** both BrJMJ18[PC] and BrJMJ18[Par] demethylate H3K36me3 and H3K36me2 in vitro and in vivo. **a** *E. coli* expressed His-BrJMJ18[PC] and His-BrJMJ18[Par] proteins demethylate H3K36me3 and H3K36me2 in vitro. Synthesized Histone H3 peptide with H3K4me3, H3K9me3, H3K27me3, H3K36me2, and H3K36me3 modifications were used as substrate. Purified His was used as a negative control. **b** Both BrJMJ18[PC]-GFP and BrJMJ18[Par]-GFP demethylate H3K36me3 and H3K36me2 in vitro. The two allelic BrJMJ18-GFP proteins were immunoaffinity-purified from *BrJMJ18 transgenic Arabidopsis* lines (*AtJMJ18::BrJMJ18[orf]*-GFP) and subjected to in vitro demethylase analysis using histone from calf thymus as a substrate. Col-0 was used as a negative control. Immunoaffinity-purified BrJMJ18-GFP proteins were detected with anti-GFP antibodies to confirm equal loading of BrJMJ18[PC]-GFP and BrJMJ18[Par]-GFP. H3, detected by blotting with anti-H3 antibodies, served as a loading control. **c** H3K36me3 and H3K36me2 status in *35S::BrJMJ18[PC/Par]-GFP* expressing tobacco leaves under normal conditions (NC) for 3 days and NC for 2 days following 1-day of HT, respectively

*35S::GFP* expressing tobacco leaves were used as a negative control. BrJMJ18-GFP proteins were detected with anti-GFP antibodies to confirm equal expression of exogenous genes. Ponceau S staining was used to assess equal loading. **d** H3K36me3 and H3K36me2 status in *BrJMJ18* loss-of-function mutants of *Par* under NC. Western blotting analysis of BrJMJ18 and H3K36me2/3 were conducted using anti-BrJMJ18 and anti-H3K36me2/3 antibodies in the plants. Five-week-old plants of *Par* grown under NC conditions were used for analysis. Ponceau S staining was used to assess equal loading. All the western blot experiments were repeated at least three times with similar results. **e, f** BrJMJ18[PC/Par]-GFP subcellular localization in transiently expressing tobacco plants under normal conditions (NC) for 3 days (**e**) and NC for 2 days following 1-day of HT (**f**), respectively. BrJMJ18-GFP protein was viewed using the confocal laser scanning system, Zeiss LSM510. Images of at least 30 epidermis cells expressing each allele of BrJMJ18-GFP protein under NC or HT conditions were acquired using the confocal laser scanning system Zeiss LSM510, respectively. Source data are provided as a Source Data file.

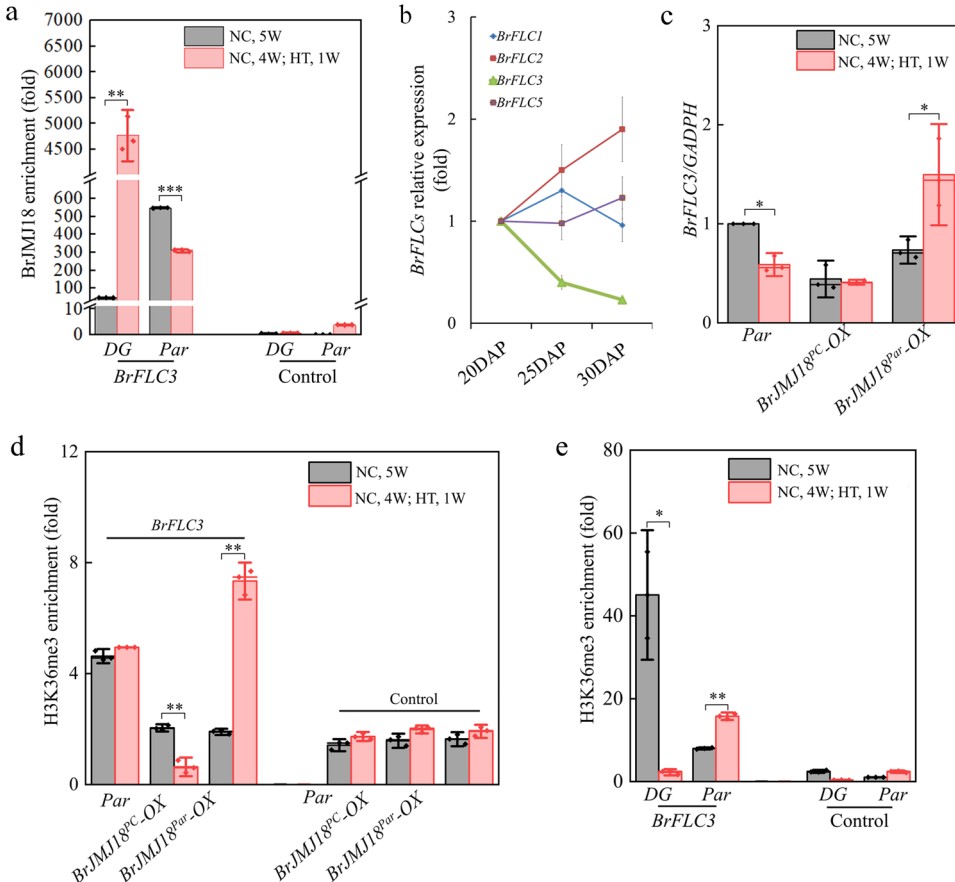

**Fig. 5 | Overexpression of *BrJMJ18^Par* moderates flowering by altering the expression of *BrFLC3*. a** Chromatin immunoprecipitation (ChIP) analysis of the BrJMJ18 level across *BrFLC3* in *BrJMJ18^Par*-carrying *Par* and *BrJMJ18^PC*-carrying *DG* plants. The 5^th–7^th young and healthy rosette leaves of five-week-old plants were used for the analysis. Rabbit IgG was used as a control. *GADPH* was used as a BrJMJ18-independent control. Control is a locus gene desert region where BrJMJ18 does not bind. The values are the mean ± standard deviation from three biological replicates. Asterisks indicate significant differences between NC and HT, two-tailed Student's t-test (**0.01 < *p* < 0.05, ***p* < 0.001). *DG*, *p* = 0.0016; *Par*, *p* = 0.00027. **b** Expression patterns of *BrFLC1, 2, 3* and *5* in *Par* before and after bolting under natural field conditions. The 5^th–7^th young and healthy rosette leaves were used for qPCR test. The values are the mean ± standard deviation from three biological replicates. The *Par* bolted at 25DAP. DAP, days after planting (14-day *Par* seedlings grown in cultivation pots in greenhouse were transplanted into natural field).

**c** *BrFLC3* expression patterns in *BrJMJ18^PC/Par*- overexpressing plants. The 5^th–7^th young and healthy rosette leaves of *Par* plants were used for qPCR test. The values are the mean ± standard deviation from three biological replicates. Asterisks indicate significant differences between NC and HT, two-tailed Student's *t* test (*0.01 < *p* < 0.05). *BrJMJ18^PC*-*OX*, *p* = 0.0032; *BrJMJ18^Par*-*OX*, *p* = 0.0023. **d** ChIP analysis of H3K36me3 enrichment on the *BrFLC3* locus in *BrJMJ18* overexpression *Par* lines. The values are the mean ± standard deviation from three biological replicates. Asterisks indicate significant differences between NC and HT, two-tailed Student's *t* test (** 0.001 < *p* < 0.01). *Par*, *p* = 0.012; *BrJMJ18^Par*-*OX*, *p* = 0.034. **e** The H3K36me3 level at *BrFLC3* in *DG* (*BrJMJ18^PC*-carrying) and *Par* (*BrJMJ18^Par*-carrying). The values are the mean ± standard deviation from three biological replicates. Asterisks indicate significant differences between NC and HT, two-tailed Student's t-test (*p* < 0.05, **0.001 < *p* < 0.01,). *DG*, *p* = 0.021; *Par*, *p* = 0.0016. Source data are provided as a Source Data file.

including *BrFLC1-3*, were among the list of the flowering genes pulled down by the BrJMJ18 protein. We then validated the ChIP-seq results by carrying out quantitative PCR to quantify the gene bodies of *BrFLCs* in *DG* (*BrJMJ18^PC*-carrying) and *Par* (*BrJMJ18^Par*-carrying) plants, respectively. We found that BrJMJ18^PC bound strongly to *BrFLC1-3*, and high temperature aggravated this binding markedly in *DG*. While in *Par*, BrJMJ18^Par only bound strongly to *BrFLC3*, and opposite of that in *DG*, high temperature thoroughly disassociated the binding of BrJMJ18^Par and *BrFLC3* (Fig. 5a and Supplementary Fig. 16a). Similar results were obtained in the anti-GFP ChIP-qPCR using *BrJMJ18^PC/Par*-*OX Par* plants grown under NC and HT, respectively (Supplementary Fig. 16b).

Furthermore, we noticed that *BrFLC3* was the only downregulated *BrFLC* during floral transition in *Par* (Fig. 5b) and was the dominant expressed *BrFLC* during flowering in *Par* (Supplementary Fig. 17). Besides, we noticed that *BrFLC3* is an effective *FLC* regulating flowering in *yellow sarson*, another type of *B. rapa* that can flower without vernalization[15]. Thus we hypothesized that *BrFLC3* is an indispensable

regulator in adjusting flowering time in *Par*. The linking of BrJMJ18 and *BrFLC3* was further supported by testing *BrFLC3* expression in transgenic *Par* plants. Under NC, *BrFLC3* expression decreased markedly in both *BrJMJ18^PC*- and *BrJMJ18^Par*-*OX* plants, while under HT, it was decreased in *BrJMJ18^PC*-*OX* but increased in *BrJMJ18^Par*-*OX* plants (Fig. 5c), which is in line with their flowering time variation under HT (Fig. 3b, c). In *Arabidopsis*, similar results for *AtFLC* were observed in transgenic plants (Supplementary Fig. 18). Therefore, all these findings implicated that in *BrJMJ18^Par*-*OX* plants, *BrFLC3* levels increase to delay flowering at high temperatures.

As described in Fig. 5c, we speculated that BrJMJ18^Par dissociates from *BrFLC3* via a mechanism by which the expression of *BrFLC3* is consequently activated upon heat in *BrJMJ18^Par*-OX plants. We were then interested in verifying whether the *BrFLC3* expression pattern is reflected in its H3K36me3 context. ChIP-PCR was carried out and we found that the H3K36me3 level at the *BrFLC3* locus was consistent with the expression pattern of *BrFLC3* in *BrJMJ18^PC*- and *BrJMJ18^Par*-overexpressing plants under both temperature conditions (Fig. 5d). We

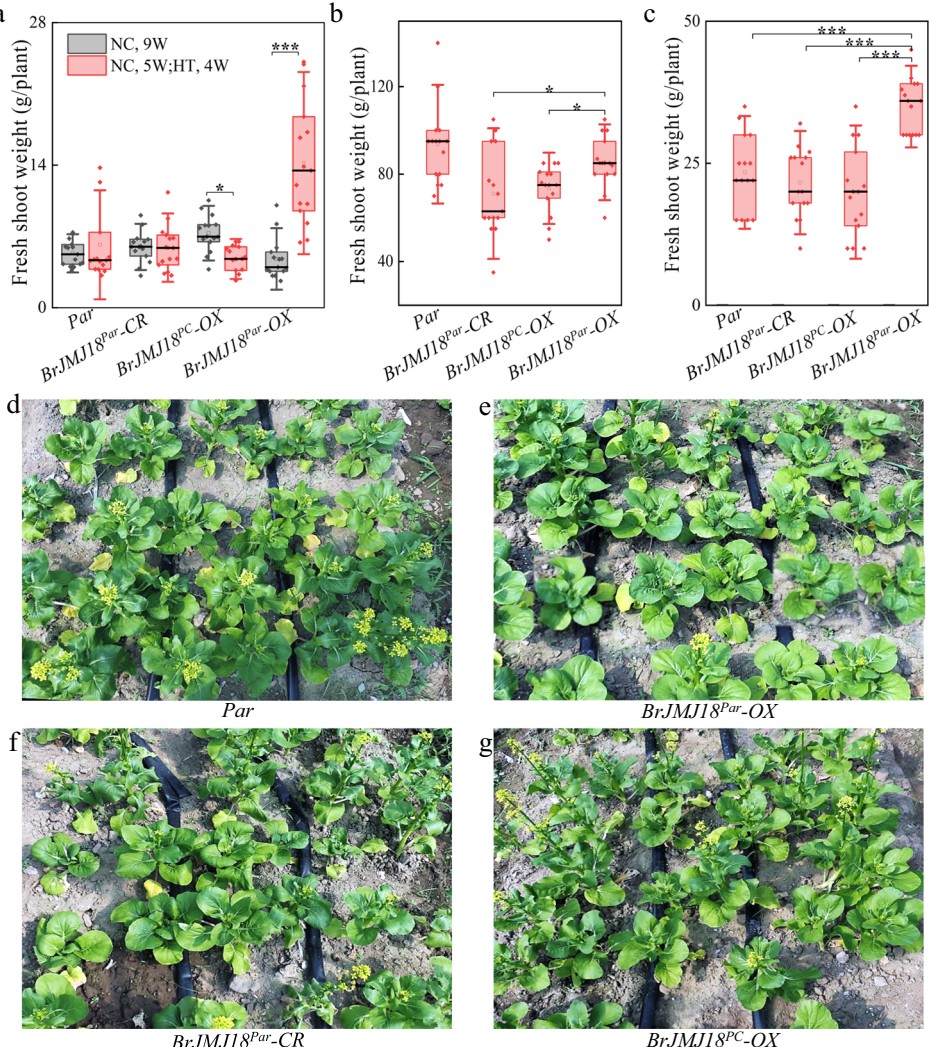

**Fig. 6 | Overexpression of *BrJMJ18^Par* mediates plant growth under both greenhouse and field high temperature conditions. a** The aboveground biomass evaluation of the *BrJMJ18* transgenic *Par* plants grown in the greenhouse under different temperature conditions. *BrJMJ18* transgenic *Par* plants grown under NC conditions for 9 weeks, and under NC conditions for 4 weeks, following another 5 weeks under high temperature (HT) conditions, respectively, were used for yield test. Data are means ± SD, *n* = 15. Asterisks indicate significant differences between NC and HT, two-tailed Student's t-test (**p* < 0.05, ***p* < 0.01, ****p* < 0.001). *BrJMJ18^PC-* *OX*, *p* = 0.0011; *BrJMJ18^Par-OX*, *p* = 8.51×10⁻⁵. **b, c** The aboveground biomass evaluation of the *BrJMJ18* transgenic *Par* plants grown in tunnel greenhouse under field conditions. The day maximum and minimum temperatures in the tunnel greenhouse were 52 °C and 31 °C, respectively, while the night maximum and minimum temperatures were 33 °C and 24 °C, respectively. **b** Plants were harvested at 55 days after sowing. **c** Plants were harvested at their proper picking stages. Data are means ± SD, *n* = 15. Asterisks indicate significant differences between *BrJMJ18^Par-OX* and the other three plants, two-tailed Student's *t* test (**p* < 0.05). In (**b**), *BrJMJ18^Par-OX* and *BrJMJ18^PC-OX*, *p* = 0.029; *BrJMJ18^Par-OX* and *BrJMJ18^Par-CR*, *p* = 0.03. In (**c**), *BrJMJ18^Par-OX* and *BrJMJ18^PC-OX*, *p* = 4.87E×10⁻⁶; *BrJMJ18^Par-OX* and *BrJMJ18^Par-CR*, *p* = 2.5 × 10⁻⁶; *BrJMJ18^Par-OX* and *Par*, *p* = 7.56 × 10⁻⁵. **d–g** Photos of the *BrJMJ18* transgenic *Par* plants grown in tunnel greenhouse under field conditions. **d**, *Par* controls; **e**, *BrJMJ18^Par-OX* plants; **f**, *BrJMJ18^Par-CR* plants; **g**, *BrJMJ18^PC-OX* plants. For box plots (**a–c**), the box encompasses two middle quartiles, with a central line showing the median. Whiskers extend to the furthest data point within 1.5 times the interquartile range. Source data are provided as a Source Data file.

also observed consistent results of the expression pattern and H3K36me3 methylation status at the *AtFLC* locus in transgenic *AtJMJ18::BrJMJ18^{PC/Par}-GFP Arabidopsis* plants (Supplementary Fig. 19). To further confirm if this was the case in non-transgenic *DG* and *Par* plants, we explored the H3K36me3 level at the *BrFLC3* locus and found that the H3K36me3 level was downregulated in *DG* (*BrJMJ18^PC-carrying*) but upregulated in *Par* (*BrJMJ18^Par-carrying*) under HT, which agreed with the enrichment of H3K36me3 at *BrFLC3* demonstrated in *BrJMJ18^PC-* and *BrJMJ18^Par-OX* plants in Fig. 5e. These results proposed that in *BrJMJ18^Par-OX* plants the late flowering primarily results from the dissociation of BrJMJ18^Par from *BrFLC3*, leading to the inability to repress its expression, while the effect of *BrFLC3* is less than straightforward in the WT *Par* plants. (Fig. 3g, j).

**Overexpression of *BrJMJ18^Par* mediates plant growth under both greenhouse and field high- temperatures conditions**

We evaluated the effects of the overexpression of the two allelic *BrJMJ18s* on major commercial qualities of different transgenic *Par* plants under both greenhouse and field conditions. For the plants grown in greenhouse under NC, *BrJMJ18^Par-* and *BrJMJ18^PC-OX* and *BrJMJ18^Par-CR* plants displayed no or slight decrease in the aboveground biomass (fresh shoot weight) and leaf number compared with *Par* controls (Fig. 6a and Supplementary Fig. 20). However, under HT, increased biomass and leaf number was found for *BrJMJ18^Par-OX* but not the other three plants (Fig. 6a and Supplementary Fig. 20). Besides, *BrJMJ18^Par-OX* had fewer aging leaves than the other plants under HT (Fig. 3b). In addition, we observed that *BrJMJ18^Par-OX* plants exhibited

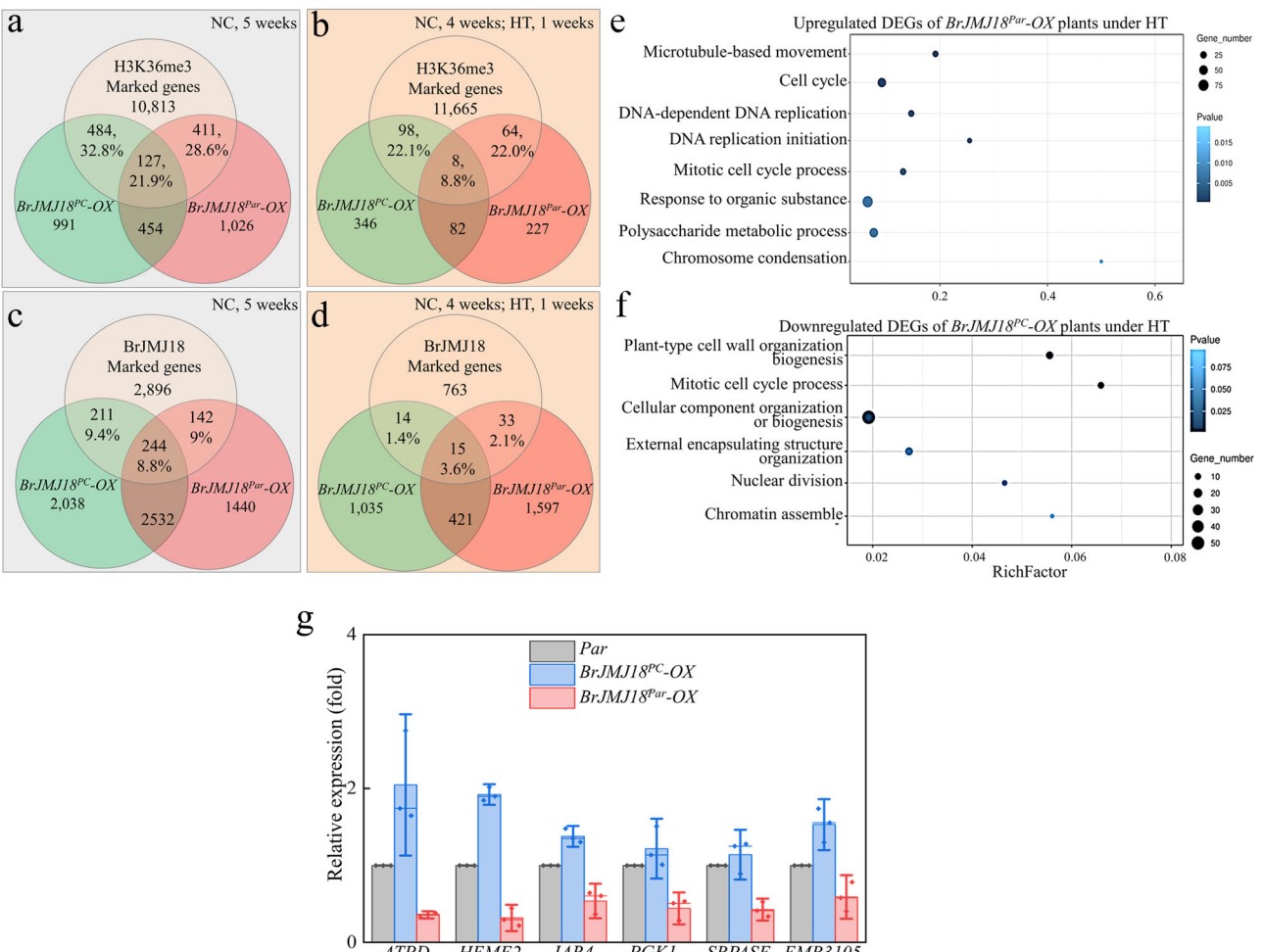

**Fig. 7 | Integrated analysis of RNA-seq and ChIP-seq data reveled BrJMJ18<sup>Par</sup> modulates chlorophyll biosynthesis ′ under high temperature. a** Diagram showing the overlap of genes targeted by BrJMJ18<sup>PC</sup> and BrJMJ18<sup>Par</sup>, respectively, and reported H3K36me3-regulated genes under NC. **b** Diagram showing the overlap of genes targeted by BrJMJ18<sup>PC</sup> and BrJMJ18<sup>Par</sup>, respectively, and reported H3K36me3-regulated genes under HT. **c** Diagram showing the overlap of the DEGs of *BrJMJ18<sup>PC</sup>-OX* and BrJMJ18<sup>Par</sup>-*OX* plants, respectively, and the identified BrJMJ18-targeted genes under NC. Genes showing a 2-fold change within a 95% confidence interval were considered to be differentially expressed. **d** Diagram showing the overlap of the DEGs of *BrJMJ18<sup>PC</sup>-OX* and BrJMJ18<sup>Par</sup>-*OX* plants, respectively, and the identified BrJMJ18-targeted genes under HT. Genes showing a 2-fold change within a 95% confidence interval were considered to be differentially expressed. **e, f** Functional categorization by Gene Ontology of the DEGs of *BrJMJ18<sup>PC</sup>-OX* and BrJMJ18<sup>Par</sup>-*OX* plants under HT. Only significantly enriched entries are shown. **e**, GO analysis demonstrated that the upregulated DEGs in *BrJMJ18<sup>Par</sup>-OX* plants under HT were

mainly enriched in cell division. **f**, the cell division-related GO entries were enriched in the downregulated DEGs in the *BrJMJ18PC-OX* line. **g** Expression of heat stress-related genes reported by Zhang et al. in *Par* and *BrJMJ18-OX* plants. 5-week-old plants grown under NC, and 4-week-old plants grown under NC followed by 1 week under HT were used for Q-PCR. Relative to *Par* plants, all six genes exhibited slight induction in *BrJMJ18<sup>PC</sup>-OX* plants under high temperatures, while they were significantly downregulated in *BrJMJ18<sup>Par</sup>-OX* plants. Photosynthesis-related gene *BraA9g029800.3 C* (ATP synthase DELTA-subunit gene, *ATPD*), porphyrin and chlorophyll metabolism-related gene *BraA04g028660.3 C* (Uroprophyrinogen decarboxylase, *HEME2*) and carbon metabolism-related genes *BraA07g034950.3 C* (Pyruvate dehydrogenase E1a-like subunit, *IAR4*), BraA03g035490.3 C (Phosphoglycerate kinase 1, *PGK1*), *BraA07g0022160.3 C* (Sedoheptulose- bisphosphatase, *SBPASE*), BraA08g004460.3 C (Embryo defective 3105, *EMB3105*). *GADPH* was used as the internal control. The values are the mean ± standard deviation from three biological replicates. Source data are provided as a Source Data file.

---

little or no changes in morphology under HT comparing with the other plants (Fig. 3a, b).

We then evaluated the yield of *BrJMJ18* transgenic plants grown in the tunnel greenhouse under natural field conditions. We harvested the edible parts of the plants in two different ways. Firstly, we picked all the plants at the same time, and the production of *BrJMJ18<sup>Par</sup>-OX* plants was comparable with that of control *Par*, but higher than *BrJMJ18<sup>Par</sup>-CR* and *BrJMJ18<sup>PC</sup>-OX* plants (Fig. 6b). However, at this point only *BrJMJ18<sup>Par</sup>-OX* plants exhibited appropriate exterior quality (Fig. 3i). We alternatively harvested the plants at their proper picking stages, and the *BrJMJ18<sup>Par</sup>-OX* line displayed much higher yield than the other plants (Fig. 6c). Interestingly, despite exhibiting a similar delayed flowering phenotype to *BrJMJ18<sup>Par</sup>-OX* plants, *BrJMJ18<sup>Par</sup>-CR* plants did not display

increased aboveground biomass and leaf numbers under HT (Fig. 6 and Supplementary Fig. 20). This suggests that the enhanced vegetable growth of *BrJMJ18<sup>Par</sup>-OX* under HT is not simply due to the deactivation of BrJMJ18<sup>Par</sup>. We hypothesized that high temperatures enable BrJMJ18<sup>Par</sup> to acquire new functions to mediate plant growth.

### BrJMJ18<sup>Par</sup> modulates chlorophyll biosynthesis under high temperatures

Given the phenotypic differences between *BrJMJ18<sup>Par</sup>-* and *BrJMJ18<sup>PC</sup>-OX* plants, we conducted a ChIP-seq assay with anti-GFP antibody using *BrJMJ18<sup>PC</sup>-OX*, *BrJMJ18<sup>Par</sup>-OX*, and *Par* plants grown under NC and HT, respectively, to explore their functional differences comprehensively. A total of 2056 and 2018 genes were enriched by BrJMJ18<sup>PC</sup> and

*BrJMJ18^Par* under NC, respectively, while the corresponding numbers under HT were 535 and 382 (Supplementary Data 7). The decreased number of enriched genes in both allelic *BrJMJ18s* under HT revealed that high temperature impeded their binding activities; while compared with BrJMJ18^PC, BrJMJ18^Par was affected more. Previous studies identified that at least 11,835 *B. rapa* genes are targeted by H3K36me3[16]. We found that under NC, 32.8% of enriched genes in *BrJMJ18^PC-OX* and 28.6% in *BrJMJ18^Par-OX* overlapped with H3K36me3-targeted genes. Under HT, these percentages dropped to 22.1% and 20.0% (Fig. 7a, b and Supplementary Data 7). We further found that 71.7% and 71.2% of the enrichment genes under NC were specific to *BrJMJ18^PC-OX* and *BrJMJ18^Par-OX* plants, respectively, while the corresponding frequencies under HT increased to 82.9% and 76.2% (Fig. 7a, b). Notably, GO analysis revealed that under NC, the enriched terms for BrJMJ18^PC and BrJMJ18^Par showed remarkable consistency (Supplementary Fig. 21a, b). In contrast, under HT conditions, the enriched GO terms diverged. Specifically, BrJMJ18^PC's targets were more enriched in entries related to cellular microscopic structure, while BrJMJ18^Par was more centralized in items associated with macromolecule metabolism (Supplementary Fig. 21c, d). We then conducted a RNA-seq assay using the same samples above. A total of 5025 and 4358 DEGs were identified in *BrJMJ18^PC-OX* and *BrJMJ18^Par-OX* plants under NC, respectively, while the corresponding DEGs under HT were 1457 and 2019, respectively (Supplementary Data 8). Over one-third of DEGs in *BrJMJ18^PC-OX* and *BrJMJ18^Par-OX* plants under HT overlapped with heat-responsive genes from Yue et al. [17] (Supplementary Data 8). Under NC, 40.5% of the DEGs in *BrJMJ18^PC-OX* plants and 45.6% in *BrJMJ18^Par-OX* plants overlapped with reported H3K36me3-targeted genes, while under HT the corresponding frequencies decreased to 31.2% and 27.9% (Supplementary Data 8). We further found that 18.2% (455 genes) of the DEGs in *BrJMJ18^PC-OX* plants and 17.8% (386 genes) of the DEGs in *BrJMJ18^Par-OX* plants are *BrJMJ18*-target genes identified by ChIP-seq under NC, while the corresponding frequencies under HT decreased to 5.0% (29 genes) and 5.7% (38 genes) (Supplementary Data 8). 44.8% and 36.3% of the DEGs under NC were specific to *BrJMJ18^PC-OX* and *BrJMJ18^Par-OX* plants, respectively, while the corresponding frequencies under HT increased to 71.0% and 79.1% (Fig. 7c, d and Supplementary Data 8), further indicating a function divergence of BrJMJ18^PC and BrJMJ18^Par under HT. We then performed GO analysis of the DEGs in *BrJMJ18^PC-* and *BrJMJ18^Par-OX* plants under NC and HT (Fig. 7e, f and Supplementary Fig. 22), respectively, to characterize BrJMJ18^PC and BrJMJ18^Par. Under NC, both the upregulated and downregulated DEGs in *BrJMJ18^PC-OX* and *BrJMJ18^Par-OX* plants showed enrichment in similar GO categories (Supplementary Fig. 22a). However, there's a notable difference under HT conditions. In *BrJMJ18^Par-OX* plants, the upregulated DEGs were predominantly associated with cell division. Surprisingly, the downregulated DEGs in the *BrJMJ18^PC-OX* line now showed enrichment in cell division-related GO terms (Fig. 7e, f and Supplementary Fig. 22b). We also observed that the downregulated DEGs in *BrJMJ18^Par-OX* were notably enriched in processes such as chlorophyll and tetrapyrrole biosynthesis, cofactor metabolic processes, and chlorophyll biosynthetic processes (Supplementary Fig. 22b). Interestingly, this enrichment pattern closely mirrors the GO items enriched among the downregulated genes in the heat-resistant Chinese cabbage under heat treatment reported by Zhang et al. [18]. We performed qPCR analysis on six heat-responsive genes associated with chlorophyll biosynthesis and carbon metabolism to assess BrJMJ18^Par's role in thermotolerance. These genes, identified by Zhang et al. [18], were also found to be DEGs in *BrJMJ18^Par-OX* plants under HT. Compared with control *Par*, all six genes showed slight induction in *BrJMJ18^PC-OX* plants under HT, while they were significantly downregulated in *BrJMJ18^Par-OX* plants. These findings suggest significant functional differences between BrJMJ18^PC and BrJMJ18^Par under high temperatures, with BrJMJ18^Par demonstrating greater effectiveness in regulating chlorophyll biosynthesis and plant growth.

## Amino acid mutations of T345A, Y633C and L654F contribute to functional divergence of BrJMJ18^PC and BrJMJ18^Par

Compared with BrJMJ18^PC, BrJMJ18^Par displayed decreased binding and catalytic activities under HT (Figs. 4–7). BrJMJ18 contains five distinct functional domains, including the JmjN, JmjC, C5HC2 zinc finger (Zf-C5HC2), F/Y-rich N terminus (FYRN), and F/Y-rich C-terminal (FYRC) domains (Fig. 2f). Among all the six amino acid mutations, T345A occurred in the known catalytic JmjC domain, while both Y633C and L654F localized in the interspace between the DNA-binding Zf-C5HC2 domain and the FYRN domain, commonly found in chromatin-associated proteins[19,20]. Therefore, we reasoned that the functional diversity of BrJMJ18^PC and BrJMJ18^Par was embodied in the mutations of T345A, Y633C, and L654F. We then transiently expressed BrJMJ18^PC, BrJMJ18^Par, and variant proteins of BrJMJ18^Par(A345T), BrJMJ18^Par(C633Y), BrJMJ18^Par(F654L), and BrJMJ18^Par(C633Y)/(F654L) in tobacco leaves to explore their catalytic functions. These variants were generated by introducing specific substitutions (A345T, C633Y, and F654L) into BrJMJ18^Par to match the BrJMJ18^PC sequence. Under NC, the demethylase activities of BrJMJ18^Par(A345T) and BrJMJ18^Par(C633Y) were found to be higher than that of BrJMJ18^PC, whereas BrJMJ18^Par(F654L) and BrJMJ18^Par(C633Y)/(F654L) exhibited lower activities (Fig. 8). However, under HT, BrJMJ18^Par(A345T) and BrJMJ18^Par(C633Y633)/(F654L) displayed similar H3K36me3/2 demethylase activity to BrJMJ18^PC, but higher than that of BrJMJ18^Par (Fig. 8). These findings indicate that mutations T345A, Y633C, and L654F contribute to the catalytic differences between BrJMJ18^PC and BrJMJ18^Par, which was also supported by our phenotypic assessments of transgenic *Arabidopsis* plants (Supplementary Fig. 23).

## Discussion

It is reported that East Asian leafy *B. rapa* crops likely evolved from turnips[2,21,22]. We cross-referenced the estimated demographic modeling with historical records of leafy *B. rapa* domestication events in China. This comparison confirmed that the genomic inferences of domestication events of *PC*, *DG* and *Par* are corroborated by documented historical records, as depicted in Supplementary Fig. 24. *Par* has acquired the capacity to maintain its flowering time and nutritional growth with minimal disruption in response to temperature fluctuations during its domestication in the warm regions of southern China. However, this crop has received limited attention. Heat tolerance seems to be polygenic, which might explain why the genetic basis of heat stress tolerance in plants is poorly understood[23–25]. Although different mapping methodologies and populations have been used to define QTLs and genes involved in thermotolerance in different plants and crops (e.g., in *Arabidopsis*, azuki bean, barley, brassica, cowpea, maize, potato, rice, sorghum, tomato, and wheat)[26–29], only a few causal genes have been identified. Here, we employed various genetic and biochemical approaches to demonstrate the thermotolerant characteristics of an epigenetic remodeler, BrJMJ18, during the domestication of *B. rapa* crops. We show a model that summarizes our hypothesis (Supplementary Fig. 25). Briefly, BrJMJ18^PC functions as a "throttle" for flowering, regardless of whether it is under NC conditions or HT. On the other hand, BrJMJ18^Par exhibits functional divergence at different temperatures. Under NC conditions, BrJMJ18^Par promotes flowering, but under high temperatures, it acts as a "brake" for flowering. This working mechanism can explain why under HT conditions, *BrJMJ18^PC-OX* or *BrJMJ18* knockout *Par* plants appear stressed, whereas *BrJMJ18^Par-OX* plants did not (Figs. 3 and 6).

### BrJMJ18 is an H3K36me2/3 demethylase

Tri-methylation at H3K36me3/2 is widely distributed in the whole genome in various species[30–33]. In *Arabidopsis*, more than four-fifths of genes are enriched with H3K36me3[34]. These facts suggest that the establishment and maintenance of H3K36me3 is indispensable during organic evolution. However, compared with the well-documented establishment of H3K36me3, the H3K36me3 erasure is barely

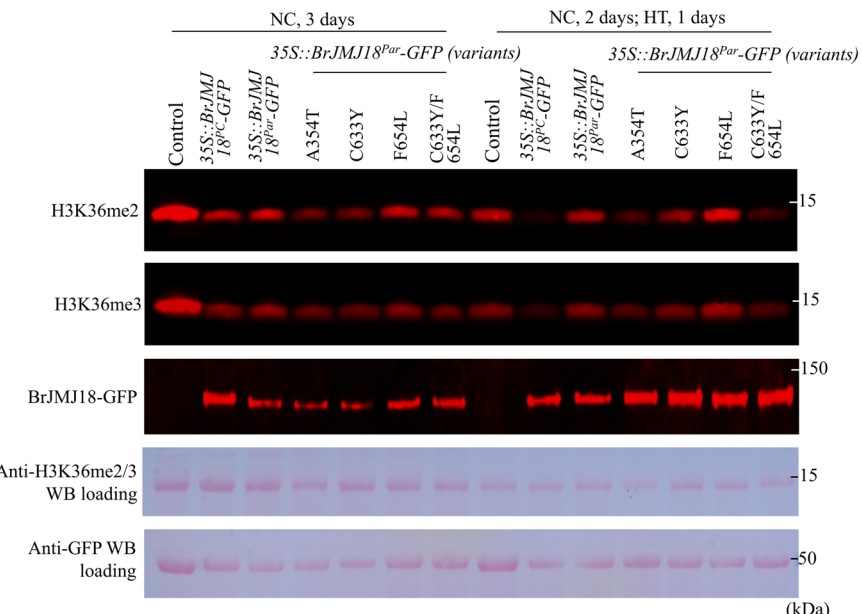

**Fig. 8 | Amino acid mutations of T345A, Y633C and L654F contribute to functional divergence of BrJMJ18[PC] and BrJMJ18[Par].** (BrJMJ18[Par(A345T)] is identical to the sequence of BrJMJ18[Par] except with a substitution at position 345 at which A was replaced with T, which is specific to BrJMJ18[PC]; the same goes for BrJMJ18[Par(C633Y)], BrJMJ18[Par(F654L)], and BrJMJ18[Par(C633Y)/(F654L)]). BrJMJ18[PC]-GFP, BrJMJ18[Par]-GFP, and its variant proteins were transiently overexpressed in tobacco leaves. H3K36me3 and H3K36me2 were detected using anti-H3K36me3 and anti-H3K36me2 antibodies in 35S::BrJMJ18-GFP expressing leaves under NC for 3 days and NC for 2 days following 1-day of HT, respectively. BrJMJ18-GFP proteins were detected with anti-GFP antibodies to confirm equal expression of exogenous genes. Ponceau S staining was used to assess equal loading of total protein extracts. All the western blot experiments were repeated at least three times with similar results. Source data are provided as a Source Data file.

described. To the best of our knowledge, only two Jumonji C-domain containing proteins, *Arabidopsis* JMJ30 and INCURVATA11 (ICU11), were reported as possible H3K36me3 demethylases in plants[34–36]; however, it is also interesting to note that AtJMJ30 and its homolog AtJMJ32 are primarily associated with inhibiting early flowering at higher temperatures by eliminating the H3K27me3 on the *AtFLC* locus[37]. Here, we showed that BrJMJ18 is an H3K36me3/2 demethylase using in vivo and in vitro demethylation activity analysis, and the increased and decreased levels at H3K36me3/2 in *BrJMJ18* over-expressing and knockout lines of *Par*, respectively. The H3K36me3/2 demethylation activity of *BrJMJ18* was also confirmed by the fact that: 1) *BrJMJ18-OX* and *sdg8*, loss-of-function mutation of the H3K36me3 methylase *SDG8*, *Arabidopsis* plants show similar flowering phenotype[34], and 2) ~ 30% the immunoprecipitated genes by BrJMJ18[PC] or BrJMJ18[Par] are reported H3K36me3-targeted genes (Fig. 7 a, b). H3K27me3 has an opposing relationship with H3K36me3 accumulation. According to Mehraj et al.[16], there are at least 10,445 *B. rapa* genes marked by H3K27me3. We further found that under NC, only 15% of enriched genes immunoprecipitated by BrJMJ18[PC] or BrJMJ18[Par] overlap with the reported H3K27me3-marked genes, and these proportions decrease to 13.0% and 10.5% under HT (Supplementary Data 7). These data indicate that the genes precipitated by BrJMJ18 are specifically targeted by H3K36me2/3.

Epigenetic mechanisms regulate diverse signaling pathways in response to environmental stresses. The flowering integrator gene, *AtFLC*, offers a classic example demonstrating the interaction between epigenetic modifications and environmental adaptation[38]. Under standard growth conditions, FLC chromatin is enriched in a number of histone modifications associated with actively transcribed genes[39], including H3K4me3/2 and H3K36me3/2. Upon cold, these markers are dynamically removed from the locus and are replaced with H3K27me3/2[40,41]. We noticed that H3K27me3 and H3K36me3 show opposing profiles under cold[42], and this antagonism is thought to be functionally important in the establishment of mutually exclusive chromatin states.

This model also predicted a physical association between antagonistic histone modifiers, specifically the H3K36me3 methyltransferase SDG8 and the H3K27me3 demethylase ELF6; such a prediction was recently confirmed experimentally[42,43]. Recently, it was reported that H3K27me3 is also important for flowering time control under ambient temperature conditions, in which Jumonji proteins (JMJ30/JMJ32/ JMJ13) mediate H3K27me3 demethylation at *FLC*, constituting one of the balancing mechanisms[37]. In addition, studies in *A. thaliana* revealed that heat memory is also partially mediated by the sustained demethylation of H3K27me3 on small HSPs[44,45]. Compared with H3K27me3, the establishment and maintenance of H3K36me3 and its biological significance are relatively unreported. It is interesting to find that Pajoro et al. report that H3K36me3 methyltransferase mutants (*sdg8-2* and *sdg26-1*) exhibited comparable flowering times under varying temperature conditions[34], while H3K36me3 demethylase mutant *jmj30-1* showed heightened sensitivity to high temperatures, indicating the necessity of H3K36me3 modification for temperature-responsive flowering. We speculated that BrJMJ18 and the H3K27me3 demethylases might form a balanced or feedback loop in histone methylation, contributing to a resilient mechanism for plant growth.

### Functional divergence of *BrFLC*s

In *Arabidopsis*, FLC is a common regulator in vernalization-mediated flowering. Recent reports suggested that FLC moderates flowering at elevated temperatures[37,46–48]. Beyond flowering control, pleiotropic functions of FLC were assigned to the development of flowers[49], leaf shape, trichome number[50], seed dormancy/germination[51,52], water usage regulation[53], and some fitness-related traits[54]. Brassicaceae genomes have undergone three rounds of whole genome duplication (WGD) after speciation from *Arabidopsis*[55,56]. We believe that during the evolution of *B. rapa*, *BrFLC* paralogs with specific neofunctions could have diverged from the ancestor *FLC*. Multiple lines of evidence show *BrFLC*s (*BrFLC1–3* and *BrFLC5*) functionally diverging. First, all four functional *BrFLC*s have been genetically mapped to flowering time

loci in *B. rapa*, varying based on G × E (population × growth conditions) (Supplementary Table 3). Secondly, all four *BrFLCs* can produce different splice variants[57–59]. Thirdly, *COOLAIR* is a cold-induced antisense RNA transcribed from the *FLC* locus which has been proposed to facilitate *FLC* silencing. In *B. rapa*, however, only *COOLAIRs* of *BrFLC2* have been detected, suggesting that *BrFLC2* might have a more important function than the other *BrFLCs* in vernalization-mediated flowering[58]. Whether the remaining *BrFLC* genes have distinct *COOLAIRs* and their respective functions remain unexplored. Lastly, different *BrFLCs* were genetically linked with traits other than flowering. For instance, a key QTL that regulates turnip development was assigned to the *BrFLC2* gene locus[60], while *BrFLC1* was proposed as a candidate influencing reproductive fitness traits[51,61]. The above information strongly suggests *BrFLCs*' functional divergence after gene duplication in *B. rapa* evolution.

Our data, together with other pieces of evidence, implied that *BrFLC3* is an active player in flowering time control: (1) *BrFLC3* is a dominant expressed *BrFLC* in *Par* (Supplementary Fig. 17); (2) *BrFLC3* expression decreased rapidly upon cold in different *B. rapa* morphotypes[58,62]; (3) *BrFLC3* overexpression caused a comparable flowering delay to *BrFLC1* and *BrFLC2* overexpressing *Arabidopsis* plants; and 4) *BrFLC3*-overexpressing Chinese cabbage plants showed a dramatically delayed flowering time[63], and (5) in *yellow sarson*, a type of *B. rapa* that can flowering without vernalization, ELF6 controls flowering by regulating *BrFLC3* expression[15]. However, the genetic correlation of *BrFLC3* with flowering time in *B. rapa* was only reported by two studies[57,64]. *BrFLC3*'s distinct genomic dissimilarity to *AtFLC* among the four *BrFLCs* (Supplementary Fig. 26) raises the possibility of neo-functionalization during evolution. Beyond, we noticed that *BrFLC3* is the only downregulated *BrFLC* in *Par*. We speculated that *Par* may have developed a pathway involving *BrFLC3* for sensing specific environmental cues. Based on our findings, under HT conditions, *BrFLC3* levels increase to delay flowering, as evidenced by the comparison of H3K36me2/3 enrichment and *BrFLC3* expression in *BrJMJ18^Par^-OX* plants. This correlation is specifically observed in *BrJMJ18^Par^-OX* lines. However, in non-transgenic *Par* plants, the scenario differs: the H3K36me3 level at *BrFLC3* remains largely unchanged (Fig. 5d), and there is no significant increase in *BrFLC3* levels at high temperatures (Fig. 5c). We hypothesize that these disparities in response likely stem from the multi-level regulation of *FLC*, including chromatin remodeling, transcriptional control, and co-transcriptional RNA metabolism. Specifically, in *BrJMJ18^Par^-OX* plants, an overabundance of BrJMJ18^Par^ proteins may disrupt the balanced regulation of *BrFLC3* seen in wild-type *Par* plants. This disruption thereby exacerbates BrJMJ18^Par^'s influence and leading to pronounced increases in H3K36me2/3 levels and *BrFLC3* expression under HT conditions. In contrast, in WT *Par* plants, the regulation of *BrFLC3* expression involves a complex interplay of mechanisms, resulting in less pronounced changes compared to the *BrJMJ18*-overexpressing lines. Moreover, both WT *Par* and *BrJMJ18^Par^-OX* plants exhibit consistent flowering time variations under HT conditions, but the latter experience more significant changes. All these findings suggested that in *BrJMJ18^Par^-OX* plants BrJMJ18^Par^ can adjust the expression of *BrFLC3* and flowering time, while the effect of *BrFLC3* is less than straightforward in the WT *Par* plants. Additionally, specific *BrFLC3* haplotypes for a *B. rapa* crop were not identified (Supplementary Fig. 27). This could elucidate the limited genetic mapping of *BrFLC3*, suggesting its expression plasticity as a potential determinant for thermotolerance.

Breeding high-yielding and resilient cultivars without significant fitness costs poses challenges. Ideally, resistance responses would remain inactive in the absence of stimuli, and stress-related gene expression would be tightly activated only during stress. Natural variants aligning with the operational model of *BrJMJ18*, as outlined here, can offer advantages for stress-resistant plant breeding. Furthermore, considering the reversible nature of H3K36me3/2 on chromatin,

epigenetic modifications can introduce adaptable, short- or long-term gene expression changes in response to varying environmental stresses, potentially enhancing adaptive capabilities in the face of environmental shifts[65,66].

## Methods

### Plant materials and growth condition
**Genotype selection, planting, and phenotyping of natural *B. rapa* collection.** A collection of 210 varieties of different *B. rapa* morphotypes, including 15 subsp. *oleifera* (*Ole*), 51 subsp. *rapifera* (*Raf*), 9 subsp. *chinensis var. narinosa* (*Nar*), 50 subsp. *chinensis* (*Par-Choi*, *PC*), 41 subsp. *chinensis DG* (*DarkGreen*, *DG*), and 44 subsp. *chinensis var. parachinensis* (*Par*) were used for genetic structure and selection analyses. All of these lines were self-pollinated for at least six generations. A separate F2 population generated from *PC* and *Par* was used to investigate the impact of allelic effects on flowering time.

The flowering time of the varieties collection was evaluated under NV and V⁴ᵂ conditions in Zhangjiakou (ZJK, 114:55E/40:51 N) and Beijing (BJ,116:28E/39:54 N), China, respectively, from 20ᵗʰ June to 20ᵗʰ September, 2018. NV, 14-day seedlings grown under a long-day (LD) regime (16/8 h day/night) and 22/22 °C at a photon flux density of 100 mol/m²/s (denoted as NC (Normal Conditions) hereafter), were moved to natural field conditions in ZJK and BJ, respectively, until bolting; V⁴ᵂ, 14-day seedlings grown under NC conditions were moved to a long-day regime (16/8 h day/night) and 4/4 °C conditions for 4-weeks (denoted as V⁴ᵂ (Vernalization for 4 weeks) hereafter), and then transplanted to natural field conditions in ZJK and BJ respectively, until bolting. Flowering time, as days after germination (DAG), was defined as the number of days from sowing to the appearance of the visible bud.

The flowering time of the F2 population derived from *Par × PC* was investigated in the Beijing Vegetable Research Center (BVRC) greenhouse. The germinated seeds were planted in soil and cultivated under NC conditions at 22 /22 °C with a 16-h light and 8-h dark period for 4 weeks. Following this, half of the seedlings were kept under NC conditions until their flowering time data were recorded, while the other half were transferred to HT conditions at 29 /29 °C with a 16-h light and 8-h dark period until they flowered. Flowering-related quantitative trait loci (ftQTLs) responding to different temperatures were then identified by using BSA analysis.

**Planting of transgenic *B. rapa* and *Arabidopsis* plants.** Seeds of inbred lines or transgenic plants of different *B. rapa* morphotypes, including *PC*, *DG*, and *Par*, were germinated under NC for 36 h, and then transferred to soil and grown under NC until the collection of phenotypic data. For high-temperature treatment, 4-week-old seedlings grown under NC were transferred to HT conditions until the collection of phenotypic data.

*Arabidopsis thaliana* Col-0 and transgenic plant seeds were surface sterilized, treated in the dark for 4 days at 4 °C, and sown on 0.5 × Murashige and Skoog (MS) medium. Seven-day-old seedlings were transferred from the plates to the soil and grown under NC. For HT treatment, 2-week-old seedlings grown under NC were transferred to a 29/29 °C growth room with an LD photoperiod at a photon flux density of 100 mol/m²/s.

### Phenotyping of transgenic plants
Flowering time: The flowering time of the transgenic *Arabidopsis* and *Col-0* plants was determined by the number of rosette leaves at the time of bolting. At least 15 individual plants of each line were measured. The flowering time of transgenic *Par* plants was determined by the days from germination to bolting. At least 15 individual plants of each line were measured.

Fresh weight: Shoots of transgenic *Arabidopsis* or *Par* plants were excised and weighed as the fresh weight.

Leaf number: The number of *Arabidopsis* or *Par* plants true leaves was counted as leaf number.

Uncropped photos of plants phenotype are provided in the Source Data file.

## Data processing and SNP calling

We conducted data processing and SNP calling as described by Su et al.[67], except that we used the *B. rapa* genome v3.0 (http://brassicadb.cn/#/Download/) as the reference.

## Population genetic analyses and gene flow estimates

Population genetic analyses and gene flow estimates were conducted as described by Su et al.[67].

## Demographic modeling

Models of the demographic history of *PC*, *DG*, and *Par* were evaluated using diffusion approximations to the allele frequency spectrum (AFS) with ∂a∂i[68]. We implemented a model testing hierarchy to accommodate ∂a∂i's limit of analyzing three populations at a time. We first evaluated one- and two-population models for each morphotype group to estimate the timing of divergence among the derived groups. The results of these initial analyses informed our subsequent demographic modeling of three different three-population model groups. In our model, we tested whether the *DG* and *Par* groups were independently derived from *PC* or whether a group ancestral to both *DG* and *Par* split from *PC* and then split into *DG* and *Par*.

## Analyses of putative selective sweeps

To detect genomic regions that have potentially differentiated during domestication or improvement of *Par*, the $F_{ST}$ scores were calculated for *DG/PC* and *Par/DG* groups using PopGenome[69] to estimate population differentiation. The $F_{ST}$ scores were estimated for 200 kb sliding windows with a step size of 5 kb. The average $F_{ST}$ of all sliding windows was considered as the value at the whole-genome level across different groups. Sliding windows with $F_{ST}$ values greater than the 95th percentile of the genome-wide $F_{ST}$ values were selected and regarded as significantly different windows. Overlapping significance windows were then merged into one fragment. These fragments were regarded as highly diverged regions across the groups. Nucleotide diversity, π, is often used as a measurement of the degree of genotype variability within a population or species. To improve the prediction accuracy, we then evaluated the nucleotide diversity of the *DG/PC* and *Par/DG* comparisons, respectively. Values of π were calculated for 200 kb overlapping windows (5 kb steps) across the genome using the BioPerl module PopGen (Stajich and Hahn, 2005). We then calculated the π ratios for each chromosome and identified potential candidate selection regions following the method described in[70].

## RNA and DNA extraction

Total RNA was isolated from the 5th–7th healthy rosette leaves of transgenic *Arabidopsis* and *PC*, *DG*, and *Par* plants and corresponding controls using an RNAprep pure plant kit (DP441, Tiangen, Beijing, China).

Total DNA of *Arabidopsis*, and *PC*, *DG*, and *Par* plants, was isolated from rosette leaves using the cetyltrimethylammonium bromide (CTAB) method[71].

## RNA sequencing

For *Arabidopsis*, the 5th–7th healthy rosette leaves were collected from Col-0 and overexpression plants (*AtJMJ18::BrJMJ 18^PC-GFP* and *AtJMJ18::BrJMJ18^Par-GFP*) grown under NC for 3 weeks or 2 weeks under NC following 1-week of HT. For *Brassica*, *PC* (PC261, ZYC), *DG* (HN129, ZJHYQ), and *Par* (CX268, SNLBLY70TCX) plants were germinated and planted in the BVRC under natural field conditions from March to June, 2019. *PC*, *DG*, and *Par* bolted at the 10th week, 6th week, and 4th week,

respectively. *BrJMJ18^PCT/Par-OX* and wildtype *Par* plants were germinated and planted in the BVRC greenhouse under NC for 5 weeks or 4 weeks following 1 weeks under HT. The 5th–7th healthy rosette leaves were collected from the third week after planting at 3 to 4 pm every Monday weekly. The leaf samples were collected at the end of the day. RNA sequencing analyses were performed by Igenecode Company (Beijing, China).

RNA-seq transcriptome library was prepared following TruSeq TM RNA sample preparation Kit from Illumina (San Diego, CA) using 1 μg of total RNA. After quantified by TBS380, the paired-end RNA-seq sequencing library was sequenced with the Illumina HiSeq xten/NovaSeq 6000 sequencer (2 × 150 bp read length). The raw paired-end reads were trimmed and quality-controlled by SeqPrep (https://github.com/jstjohn/SeqPrep) and Sickle (https://github.com/najoshi/sickle) with default parameters. Then clean reads were separately aligned to the reference genome with orientation mode using HISAT2 (http://ccb.jhu.edu/software/hisat2/index.shtml software. The mapped reads of each sample were assembled by StringTie (https://ccb.jhu.edu/software/stringtie/index.shtml? t = example) in a reference-based approach. To identify DEGs (differential expression genes) between two different samples, the expression level of each transcript was calculated according to the transcripts per million reads (TPM) method. RSEM was used to quantify gene abundances. Essentially, differential expression analysis was performed using the DESeq2[4]/DEGseq[5]/EdgeR[6]with Q value ≤ 0.05, DEGs with |log2FC | >1 and Q value < = 0.05(DESeq2 or EdgeR) /Q value < = 0.001(DEGseq) were considered to be significantly different expressed genes). In addition, functional-enrichment analysis including GO and KEGG were performed to identify which DEGs were significantly enriched in GO terms and metabolic pathways at Bonferroni-corrected P-value ≤ 0.05 compared with the whole-transcriptome background. GO functional enrichment and KEGG pathway analysis were carried out by Goatools (https://github.com/tanghaibao/Goatools) and KOBAS.

## Bulked segregation analyses (BSA)

Bulked segregation analyses were performed by BoYunHuaKang Company (Beijing, China). Sequencing libraries were generated using NEB Next® Ultra DNA Library Prep Kit for Illumina®(NEB, USA) following manufacturer's recommendations and index codes were added to attribute sequences to each sample. The clustering of the index-coded samples was performed on a cBot Cluster Generation System using HiSeq 4000 PE Cluster Kit (Illumia) according to the manufacturer's instructions. After cluster generation, the library preparations were sequenced on an Illumina Hiseq 4000 platform and 150 bp paired-end reads were generated. Two parameters, SNP-index and Δ (SNP-index)[72] were calculated to identify candidate regions. An SNP-index is the proportion of reads harboring the SNP that are different from the reference sequence. Δ (SNP-index) was obtained by subtraction of the SNP-index of two pools. For each read depth, 95 % confidence intervals of Δ (SNP-index) were obtained.

## Quantitative real-time reverse transcription PCR (qRT-PCR)

First-strand cDNA was synthesized using a Prime Script™ reagent Kit with gDNA Eraser (RR047A, Takara, Dalian, China). qPCR was performed using the SYBR Green PCR master mix (04887352001, Roche, Basel, Switzerland), and a LightCycler 480 Real-Time PCR system (Roche, Basel, Switzerland). *Actin2* (*Arabidopsis*) and *GAPDH* (*B. rapa*) were used as internal controls. qPCR primers are listed in Supplemental Supplementary Table 4.

## Vector and transgenic plants constructions

The 2 kb upstream region of *AtJMJ18* and the coding region of *BrJMJ18^PC* and *BrJMJ18^Par* were obtained by PCR and confirmed by sequencing. The 2 kb upstream region of *AtJMJ18* was used as a promoter and inserted into vector *p*CMBIA1300[73] to substitute for the CaMV35S

promoter. The SpeI/KpnI fragments containing the open reading frames of *BrJMJ18^PC* and *BrJMJ18^Par* were inserted into a plasmid with a green fluorescent protein (GFP) tag at the 3′ end. For the overexpression vectors, XbaI/SalI fragments of *BrJMJ18^PC* and BrJMJ18^Par coding regions were inserted into vector *pCMBIA2300* vector with CaMV35S promoter and a GFP tag at the 3′ ends. The resultant constructs were transformed into *Agrobacterium tumefaciens* strain GV3101. *pCMBIA1300-AtJMJ18pro-BrJMJ18^PC/Par* vectors were used to generate of *Arabidopsis BrJMJ18^PC/Par* overexpression transgenic plants, denoted as *AtJMJ18::BrJMJ18^PC-GFP* and *AtJMJ18::BrJMJ18^Par-GFP*, respectively; while *pCMBIA2300-35S-BrJMJ18^PC/Par* vectors were used to generate *BrJMJ18^PC/Par* overexpression plants of *Par*, denoted as *35S::BrJMJ18^PC-GFP* (*BrJMJ18^PC*-OX) and *35S::BrJMJ18^Par-GFP* (*BrJMJ18^Par*-OX), respectively. For prokaryotic expression vectors, the EcoRI/SalI fragments of BrJMJ18^PC/Par coding region were inserted into the pET28a (+) vector with His Tag at the 3′ end. The resultant constructs were transformed into *Escherichia coli* strain BL21. To construct CRISPR/Cas9 vectors, single guide RNA (sgRNA) sequences were designed using the web server CRISPR-P3. Using pCBC-DT1T2 as the template, two *AtU6* promoter-sgRNA-AtU6 terminator cassettes were amplified using PCR. The PCR fragments were inserted into *pKSE401*[74] and confirmed using Sanger sequencing. The primers used are listed in Supplementary Table 4.

***Arabidopsis* transgenic plants were generated using the flower-dipping method**[75]. The T1 *BrJMJ18* transgenic plants were screened using 15 mg/L hygromycin. Transgene expression was detected by immunoblotting using an anti-GFP antibody (HT801-01, TransGene, Beijing, China). Progeny of *AtJMJ18::BrJMJ18^PC-GFP* lines 4#, 6#, and N1#; and *AtJMJ18::BrJMJ18^Par-GFP* lines 2#, 8#, and N3# revealed a 3:1 separation ratio against hygromycin. T2 seeds of these lines were used for further experiments.

### *B. rapa* transgenic plants were constructed as following

Agrobacterium Preparation: Transform *Agrobacterium* strain GV3101 via electroporation with the constructs of interest. Inoculate single colonies into 10 mL LB cultures (50 mg/L kanamycin and 50 mg/L rifampicin) and incubate at 28 °C, 220 rpm for 48 h. Centrifuge the culture at 3000 × *g* for 15 min, and re-suspend the pellet in liquid MS at OD650 = 0.05.

Seed Sterilization and Germination: Sterilize *Par* inbred line 16A-1 seeds with 75% ethanol (1 min) and 5% sodium hypochlorite (15 min). Germinate seeds on GM plates at 25 °C for 4 days.

Explant Isolation, Inoculation, and Co-cultivation: Cut 1–2 mm cotyledonary petioles of 4-day-old seedlings and place them on CIM-C plates. Inoculate by dipping petiole ends in *Agrobacterium* suspension. Return explants to the same CIM-C plates and Incubate in a growth chamber at 25 °C in dim darkness for 72 h.

Selection and Transfer of Plants to Greenhouse. After 72 h on CIM-C plates, the explants were transferred to the SIM plates and moved to a growth chamber at 25/22 °C with a 16/8-h (light/dark) photoperiod (photon flux density 100 mol/m²/s) for 2 weeks. Then the explants were moved to fresh SIM plates for a further 1-3 weeks until the emergence of green shoots. Green shoots were transferred to SEM plates for a further 1-2 weeks in the same growth chamber. The well-developed green shoots were transferred to the RM plates in the same growth chamber for 1-3 weeks. Rooted green plants were acclimatized in a growth chamber under 25 °C with 70–80% humidity for 2 days and then transferred into the soil.

Transgenic Plant Verification: For overexpression lines, transgene expression was confirmed by immunoblotting with an anti-GFP antibody (HT801-01, TransGene, Beijing, China) for *BrJMJ18^PC/Par*-OX plants. T0 plants of both lines were self-crossed to generate seeds. In the case of CRISPR/Cas9 plants, the target gene *BrJMJ18* from T0 plants was

amplified, purified, and cloned into a vector using the pMD™18-T Vector Cloning Kit (Takara) for sequencing. Progeny T1 seedlings were grown on 0.5 × MS medium without kanamycin. Loss-of-function *BrJMJ18* seedlings, confirmed through PCR and Sanger sequencing, were used for subsequent analysis.

Plant culture Media are listed as below:

Liquid MS: 4.3 g/L Murashige and Skoog (MS) basal salts (M519, Phytotech, USA), 30 g/L Sucrose, pH 5.7.

Germination media (GM): 4.3 g/L Murashige and Skoog (MS) basal salts, 30 g/L Sucrose, pH 5.7, 8 g/L agar.

Callus induction media for cocultivation (CIM-C): GM plus 500 mg/L 2-(N-morpholino) ethanesulfonic acid (MES), 4 mg/L 6-BA, 0.667 mg/L IAA.

Shoot induction media (SIM): CIM-C medium plus 160 mg/L Timentin, 2 mg/L AgNO₃ and 25 mg/L kanamycin.

Shoot elongation media (SEM): GM plus 500 mg/L MES, 160 mg/L Timentin, 50 µg/L 6-BA, 2 mg/L AgNO3 and 25 mg/L kanamycin.

Rooting medium (RM): 3.05 g/L Gamborg's B5 salts (G398, PhytoTECH, USA), 10 g/L sucrose, pH5.7, 8 g/L agar, 160 mg/L Timentin, 1 mg/L IBA, 2 mg/L AgNO₃ and 25 mg/L kanamycin.

### Transient transformation

Tobacco (*Nicotiana benthamiana*) leaf transient transformation was performed as described previously[76]. Infiltrated tobacco plants grown under NC for three days and under NC for two days following 1 day of HT, respectively, were used for further analysis. BrJMJ18-GFP protein expression was viewed using the confocal laser scanning system Zeiss LSM510 (Carl Zeiss, Jena, Germany). BrJMJ18-GFP protein and Histone H3K36 methylation levels were determined using immunoblotting analysis using anti-GFP (HT801-01, TransGene, Beijing, China) and anti-Histone H3 di/tri methyl K36 antibodies (ab176921/ab195489, Abcam, Cambridge, MA, USA).

### Protein extraction and immunoblotting analysis

For BrJMJ18 protein detection, total proteins were extracted from *Arabidopsis* and *Par* young rosette leaves or transiently transformed tobacco leaves using extraction buffer (50 mM Tris-HCl, pH 7.5, 150 mM NaCl, 10 mM MgCl₂, 1% Triton X-100 and 1 mM Phenylmethanesulfonyl fluoride). For histone methylation detection, total proteins were extracted using histone extraction buffer (50 mM Tris-HCl, pH 7.5, 0.5 M sucrose, 1 mM MgCl₂, 1 mM EDTA, 5 mM DTT, 1 mM Phenylmethanesulfonyl fluoride and 1 × protease inhibitor cocktail). After centrifuged at 14,000 × *g* for 30 min at 4 °C, the proteins in the supernatant were separated by SDS-PAGE and transferred to a 0.4 µm nitrocellulose blotting membrane (10600002, GE Healthcare, Chicago, IL, USA) using a Bio-Rad Tran-Blot Turbo transfer system (Bio-Rad, Hercules, CA, USA).

Antibodies against the following proteins were used: GFP (HT801-01, TransGen Biotech, Beijing, China), β-Tubulin (HC101, TransGen Biotech), Histone H3 (ab1791, Abcam, Cambridge, MA, USA), Histone H3 mono/di/tri methyl K4 (ab8895/ab32356/ab213224, Abcam, Cambridge, MA, USA), Histone H3 mono/di/tri methyl K9 (ab176880/ab32521/ab176916, Abcam), Histone H3 Histone H3 mono/di/tri methyl K27 (ab194688/ab24684/ab195477, Abcam), and Histone H3 mono/di/tri methyl K36 (ab176920/ab176921/ab195489, Abcam). β-tubulin and Ponceau S staining were used as loading controls. IRDye 680RD Goat anti-mouse (926-68070, LI-COR, Lincoln, NE, USA) or anti-rabbit (926-68071, LI-COR) antibodies were used as secondary antibodies. Anti-BrJMJ18 antibodies were produced by immunizing rabbits with a recombinantly expressed BrJMJ18 protein fragment (amino acids 1–133) expressed in *Escherichia coli* and purified using an BrJMJ18 antigen column (HuaBio, Hangzhou, China). Uncropped scans of immunoblotting blots are provided in the Source Data file.

## Preparation of recombinant protein

*Escherichia coli* cells transformed with pET28a (+)-*BrJMJ18*[PC/Par] constructs were cultured, and isopropyl-D-thiogalactopyranoside was added to induce the recombinant protein expression. His-BrJMJ18[PC/Par] proteins were purified using a Ni2 + -NTA agarose column (40724, QIAGEN, Germany).

**In vitro histone demethylase assay using recombinant BrJMJ18-GFP protein.** The histone demethylase assay was performed as previously reported[37] with some modifications. Total proteins of young rosette leaves of *BrJMJ18* transgenic plants were extracted using extraction buffer (50 mM Tris-HCl, pH 7.5, 150 mM NaCl, 10 mM MgCl$_2$, 1% Triton X-100 and 1 mM Phenylmethanesulfonyl fluoride) and centrifuged at $14,000 \times g$ for 30 min at 4 °C. The supernatant was incubated with anti-GFP monoclonal antibody (mAb) agarose (D153-8, MBL, Woburn, MA, USA) at 4 °C for 3 h. The agarose was collected by centrifugation and washed with wash buffer (50 mM Tris-HCl, pH 7.5, 150 mM NaCl, 10 mM MgCl$_2$) three times. Then, the agarose was added to demethylation buffer (20 mM Tris-HCl, pH 7.5, 150 mM NaCl, 50 μM (NH4)$_2$Fe(SO$_4$)$_2$, 1 mM α-ketoglutarate, 2 mM ascorbic acid, and 0.1 mg/mL Histone from calf thymus (1223565001, Roche, Basel, Switzerland) and incubated at 37 °C for 4 h.

**In vitro histone demethylase assay using recombinant His-BrJMJ18 protein.** The histone demethylase assay was performed as previously reported[77] with some modifications. 2.5 μg synthesized Histone H3 peptide with H3K4me3, H3K9me3, H3K27me3, H3K36me2, H3K36me3 modification (R-1023, R-1028, R-1034, R-1038, R-1039, Epigentek, USA) were incubated with affinity-purified recombinant His-BrJMJ18[PC/Par] protein (approximately 0.8 μg-2.4 μg) in 30 μL reaction buffer (20 mM Tris-HCl, pH 7.5, 150 mM NaCl, 50 μM (NH4)$_2$Fe(SO$_4$)$_2$, 1 mM α-ketoglutarate, 2 mM ascorbic acid) for 4 h at 37 °C. Purified His was used as a negative control.

The proteins in the reaction solution were separated by SDS-PAGE and transferred to 0.4 μm nitrocellulose membrane (10600002, GE Healthcare, Chicago, IL, USA) to detect histone H3 methylation using immunoblotting analysis with anti-histone H3 mono/di/tri methyl K4, mono/di/tri methyl K9, mono/di/tri methyl K27, and mono/di/tri methyl K36 antibodies. The amount of H3 in each sample (detected with anti-histone H3 antibody (ab1791, Abcam, Cambridge, MA, USA)) served as a loading control. His-BrJMJ18[PC/Par] proteins were detected with anti-BrJMJ18 antibody. The agarose was boiled in 1 × SDS buffer for 5 mins and the proteins in the supernatant were separated by SDS-PAGE and transferred to 0.4 μm nitrocellulose membranes (10600002, GE Healthcare, Chicago, IL, USA) to detect BrJMJ18-GFP protein using immunoblotting analysis with anti-GFP antibodies.

## Chromatin immunoprecipitation (ChIP)

For ChIP-Sequencing using anti-BrJMJ18 antibodies, the 5th–7th healthy rosette leaves of 5-week-old *Par* plants grown under NC were used. And the leaf samples were collected at the end of the day. For ChIP-seq using anti-GFP antibodies (ab290, Abcam, Cambridge, MA, USA), the 55th–7th healthy rosette leaves of *BrJMJ18*[PC/Par]-OX and wild-type *Par* plants grown under NC for 5 weeks or 4 weeks following 1 weeks under HT were used. The chIP-seq experiment was carried out by IGENE-BOOK Biotechnology Company (Wuhan, China) using rabbit polyclonal anti-BrJMJ18 antibodies and according to a previously described method[78]. Two biological replicates were performed for each sample. Briefly, 1.5 g samples were washed twice in cold phosphate-buffered saline (PBS), cross-linked with 1% formaldehyde for 10 min at room temperature, and then quenched by the addition of glycine (125 mmol/L final concentration). Afterwards, samples were lysed and chromatin was obtained on ice. Chromatin was sonicated to get soluble sheared chromatin (average DNA length of 200–500 bp). Then, 20 μL of chromatin was reserved at −20 °C as input DNA, and 100 μL of chromatin was used for immunoprecipitation using 10 μg of anti-BrJMJ18 at 4 °C overnight. The next day, 30 μL of protein beads were added and the samples were further incubated for 3 h. The beads were then washed once with 20 mM Tris/HCL (pH 8.1), 50 mM NaCl, 2 mM EDTA, 1% Triton X-100, 0.1% SDS; twice with 10 mM Tris/HCL (pH 8.1), 250 mM LiCl, 1 mM EDTA, 1% NP-40, 1% deoxycholic acid; and twice with TE buffer 1 × (10 mM Tris-Cl at pH 7.5. 1 mM EDTA). Bound material was then eluted from the beads in 300 μL of elution buffer (100 mM NaHCO$_3$, 1% SDS), treated first with RNase A (final concentration 8 μg/mL) for 6 h at 65 °C and then with proteinase K (final concentration 345 μg/mL) overnight at 45 °C. Immunoprecipitated DNA was used to construct sequencing libraries following the protocol provided by the I NEXTFLEX® ChIP-Seq Library Prep Kit for Illumina® Sequencing (NOVA-5143-02, Bioo Scientific, Austin, TX, USA) and sequenced on Illumina Xten with the PE 150 method (Illumina Inc. San Diego, CA, USA). Trimmomatic (version 0.38) was used to filter out low-quality reads (Bolger et al., 2014). Clean reads were mapped to the *B. rapa* genome v3.0 using Bwa (version 0.7.15) (Li and Durbin, 2009). Samtools (version 1.3.1) was used to remove potential PCR duplicates[79]. MACS2 software (version 2.1.1.20160309) was used to call peaks using default parameters (bandwidth, 300 bp; model fold, 5, 50; *q* value, 0.05) using input as control. If the summit of a peak is located closest to the transcription start site (TSS) of one gene, the peak was assigned to that gene[80]. Gene Ontology (GO) enrichment analysis was performed using the EasyGO gene ontology enrichment analysis tool (http://bioinformatics.cau.edu.cn/easygo/)[81]. And BrJMJ18 target genes under NC/HT were obtained using wild-type *Par* plants under NC/HT as control. The GO term enrichment was calculated using hypergeometric distribution, with a *P* value cutoff of 0.01. *P* values obtained by Fisher's exact test were adjusted by the false discovery rate (FDR) for multiple comparisons to detect overrepresented GO terms.

For ChIP-QPCR, experiments were performed using an Imprint Chromatin Immunoprecipitation Kit (CHP1, Sigma-Aldrich, St. Louis, MO, USA). For *Brassica* crops, plants grown under NC for 5 weeks or 4 weeks under NC following 1-week of HT, respectively, were used as materials. For *Arabidopsis*, Col-0 and *AtJMJ18::BrJMJ18*[PC]-GFP, and *AtJMJ18::BrJMJ18*[Par]-GFP plants grown under NC for 3 weeks or 2 weeks under NC following 1-week of HT, respectively, were used for analysis. Young and healthy rosette leaves (40 mg) of the above mentioned plants were used for ChIP qPCR experiments. Rabbit polyclonal anti-BrJMJ18 antibodies and rabbit polyclonal anti-Histone H3 di/tri methyl K36 antibodies (ab176921/ab195489, Abcam) were used in the immunoprecipitations and normal rabbit IgG (12-370, Sigma-Aldrich) was use as the negative control. *Actin2* (*Arabidopsis*) was used as the internal control. The level of the protein in the negative control was set as 1 and enrichment represented the relative fold-change fold in relation to the negative control. Primers used are listed in Supplementary Table 4.

## Statistical analysis

For bar graphs, error bars represent the SD, data are presented as means ± SD from $n = 3$. For box-and-whisker plots, boxes show the interquartile range (i.e., the 1st and 3rd quartile) framing the median. The whiskers show the minimal and maximal values except if these values are higher than the interquartile range multiplied by 1.5. In this case, the whiskers show the interquartile range by 1.5 and the dots are outliers. The significance of the difference in the columns/boxplots were calculated with a two-tailed Student's *t* test for the pairwise comparisons (*$p < 0.05$; **$p < 0.01$; ***$p < 0.001$).

## Reporting summary

Further information on research design is available in the Nature Portfolio Reporting Summary linked to this article.

## Data availability

ChIP-seq data have been deposited on NCBI with accession number GSE223969. All the raw sequencing data are archived at the Genome Sequence Archive of China National Center for Bioinformation with the accession number PRJCA025632. Source data are provided with this paper.

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

## Acknowledgements

This work was supported by grants from the National Key Research and Development Program of China (2022YFF1003003, T.S.), the National Natural Science Foundation of China (32172557, T.S.), the Innovation and Capacity-Building Project of BAAFS (KJCX20240408, S.Y., and KJCX20230431, X.X.), the Outstanding Scientists Training program of BAAFS (JKZX202406, T.S.), and the China Agriculture Research System of MOF and MARA (CARS-A03, T.S.).

## Author contributions

T.S., X.X., S.Y., and F.Z. designed the experiments. T.S., X.X., and P.L. carried out the analyses. W.W., X.Z., G.J., J.W., L.S., Y.Y., and D.Z. collected the phenotypes. T.S. and X.X. wrote and revised the paper. All authors discussed the results and commented on the manuscript.

## Competing interests

The authors declared no competing interests.
