## [Peer Review File · Nature Communications]

Temperature-Dependent Jumonji Demethylase Modulates Flowering Time by Targeting H3K36me2/3 in *Brassica rapa*REVIEWER COMMENTS

Reviewer #1 (Remarks to the Author):

This manuscript study the response of *B. rapa* to high temperatures. Through QTL and genomic analyses of a set of *B. rapa* accessions, the authors found a number of SNPs in an homolog of *At.JMJ18* in the thermotolerant cultivar *B. rapa* var. *parachinensis*. Then, they performed an impressive work including enzymatic assays, transgenics in *B. rapa* and *A. thaliana*, RNA-seq, ChIP-seq experiments to characterize *Br.JMJ18* as a novel regulator of the thermosensory pathway in plants. They conclude that in most *B. rapa* accessions under, heat promotes *Br.FLC3* downregulation by *Br.JMJ18*. However, in *B. rapa* var. *parachinensis*, heat impairs *Br.JMJ18* target-binding promoting vegetative growth and late flowering time.

Although the work is very relevant and the results are quite complete. I found that conclusion are not satisfactory and some points require clarification prior publication.

Major concerns

- 1) First, it is not clear to me what happen with *B. rapa* (in general) flowering time at high temperature? The final model suggest that heat may accelerate flowering in “*Br.JMJ18-wt*” accessions (pack choi, dark green, etc.) but no data is provided. Current published data shows that some *B. rapa* varieties delay flowering at high ambient e temperature (Chen et al genes 2022, Del Olmo et al Plan Journal 2019).
- 2) Across the manuscript, the authors talk about “Heat Stress” treatment. But it believe it is a treatment at 29C. Is that really heat stress? Should it not be called heat treatment or warm temperature? Please explain this in the text and include the temperature of the treatment in the figure legends.
- 3)To my eyes, plants that carry *Br.JMJ18-par* or *CRISP* allele look stressed under high temperature (figure 3). However thsi phenotype is not discussed in the text.
- 4)Line301. The transgenic line production should be explained and introduced in the main text not in the figure 3 legend .
- 5)From line 488-556. All this section, is inconclusive. The genomic analyses are great, but the concoctions are not solid and needs rewriting. For example did they explored is there is a correlation between heat responsive genes and *Br.JMJ18* binding or misregulated genes? In my opinion the sentences “We thus speculated that *Br.JMJ18Par* promotes vegetative growth probably through regulation of cell division under heat stresses” is too much speculation. At least, to be repeated as a solid conclusion of the work.
- 6)Line 356 *ICU11* is also proposed to be an *H3K36me3*, please cite Bloomer et al PNAS 2022.
- 7)Line 350: *Br.FLC3* is also crucial to explain the flowering phenotype of *Br.ELF6* and *Br.REF6* in yellow sarson. Please cite Poza-Viejo et al. Plant Cell&Env. 2022

8) Lines 677-680 need to be rewritten, I do not see the point. ELF6 and REF6 flowering time is not confusing: ELF6 regulates FLC directly, REF6 does not regulate FLC directly but is late because it has high levels of FLC. Triple and double H3K27 histone demethylase mutants have severe developmental phenotypes (see Yan et al. Nat Plants 2018)

9) Line 749. Please provide details about the RNA-seq methods (library preparation, sequencing length, bioinformatics analyses)

10) Line 985. It is prodigious how many *B. rapa* transgenic lines were produced. Please provide a more detailed protocol. This will help the entire scientific community.

11) Line 1008: Could you explain why the genomic data is not deposited in public databases?

Minor points

12) References 1 and 2 are about tomato. I think there are not related to this work.

13) Line 18. The authors write about *B. rapa parachinensis* as Chinese kale, I believe that a kale is *B. oleracea*. I understand the in English name for *parachinensis* is Chinese cabbage or Caixin. Please revise that.

14) Fig1 Could you please put a bigger world-wide map in Fig1?

15) Fig2 There are two QTL (A3 and A7) that the authors do not even mention. Any speculation about that.

16) Fig3C I think that "x-axis" legend is missing

17) I found the discussion a bit long. I think that some parts of the discussion could be shortened a few sentences and the paper would be improved.

Reviewer #2 (Remarks to the Author):

Flowering time is essential for crop yield and the response to global warming. This manuscript aims to identify the genetic mechanisms of flowering time regulation and heat stress tolerance in *Brassica rapa* with considerable work and multiple approaches. However, after reading the manuscript, I feel the key message of the story, and the information flow is not clear, leading to a lot of issues in understanding the story. Here are my main comments:

1. The story starts with the domestication analysis of Par, DG, and PC. When we look at the place of the collected accession, we can see most of the par accessions are from the south of China, which differs

from DG and PC. Therefore, I doubt the different flowering time of Par is domestication from DG and PC or a local adaptation of collected Par accession. Moreover, it is also unclear whether the results in Figure 1F indicated the flowering time in all collected Par, DG, and PC accessions or only representative accessions.

2. The authors first aimed to identify the genes responsible for inducing flowers in Par under normal conditions (NC). However, from line 185, the authors suddenly switch the topic to flowering time at heat-stressed (HT) conditions, which disrupts the information flow. And it is not clear whether the authors performed the QTL analysis based on early or delayed flowering at HT. After reading the whole manuscript, I can see the delayed flowering time in Par is the key phenotype. Then why don't start the story directly from this phenotype? As I can see, the mapping of BrJMJ18 relies on the QTL analysis. Then why the domestication part is necessary for this story?

3. In line 238, the authors showed that BrJMJ18 is differentially expressed during floral transition among three tested subspecies, but the author further showed that BrJMJ18 doesn't have differential expression during flowering. Considering the mechanism of flowering time regulation, the expression during floral transition is, of course, more important. Then why do authors focus on the amino acid changes but not the expression level if that is the case?

4. The authors always want to discuss the function of BrJMJ18 in flowering time regulation at both NC and HT conditions, making it difficult to follow the story. BrJMJ18Par at NC is not crucial if the authors mainly want to analyze BrJMJ18Par in HT. I think they should simplify the NC part and focus on the function of BrJMJ18Par at HT.

5. There are a lot of data generated in Arabidopsis. I doubt the necessity since they can create mutants or overexpression lines in Par. I think they should remove these data from this story.

6. The authors claim that BrJMJ18Par delays flowering time at HT. If this is the case, why does losing BrJMJ18 result in late flowering under HT? It is very confusing.

7. According to the level of H3K36me3, the authors believe that BrJMJ18Par dissociates from the BrFLC3 locus. The author has already generated the antibody of BrJMJ18; then, they should confirm the dissociation of BrJMJ18 at HT by CHIP-PCR?

8. I am also confused why the authors don't perform the CHIP-seq with the BrJMJ18 antibody at both NC and HT. I think such results are more informative than BrJMJ18-GFP CHIP-seq in overexpression lines.

9. The figures do not always clear the number of biological replicates or the number of analyzed samples. Therefore, it is sometimes difficult to judge the conclusion.

10. I expect a clear mechanism to explain how BrJMJ18Par delay flowering time at HT in Par; a model will be helpful.

Reviewer #3 (Remarks to the Author):

This manuscript identifies BrJMJ18 as a targeted locus for the domestication of a thermotolerant *B. rapa* crop, Chinese kale (*B. rapa* subsp. *chinensis* var. *parachinensis*). The authors utilized various domesticated strains of *B. rapa* to identify loci that are adapted to tropical regions. In this manuscript, the authors focused on gQTLA109-1 and identified BrJMJ18 as a candidate gene that confers local

adaptation of *parachinensis* (Par). The authors presented data to show that BrJMJ18 is an H3K36me_{2/3} demethylase and nonsynonymous variations in BrJMJ18 protein confer thermotolerant traits of Par. This study may provide interesting insights into knowledge on the domestication of crops in various local environments, including different climate regimes. However, several issues need to be addressed to warrant the conclusion. Major issues are listed below.

Major points.

1. Re: Fig. 4 BrJMJ18 as an H3K36me_{2/3} demethylase

In this manuscript, the authors used two methods to argue that BrJMJ18 is an H3K36me_{2/3} demethylase. 1) The authors used transgenic *Arabidopsis* plants in which BrJMJ18 is under the control of the *Arabidopsis* JMJ18 promoter. Affinity purified protein extract of BrJMJ18 from *Arabidopsis* was incubated together with calf thymus histones to detect activity (Fig. 4A; Fig. S14). 2) transiently expressed two variants of BrJMJ18 in tobacco and extracted histone to detect the level of H3K36me_{2/3} (Fig. 4B).

1-a. A previous study of *Arabidopsis* JMJ18 showed that the *Arabidopsis* ortholog of JMJ18 is an H3K4me_{2/3} demethylase (Yang et al., 2012). Neither of the two methods the authors used is appropriate to characterize the specificity of demethylases. Purified recombinant proteins with careful assays should be necessary.

1-b. It is my understanding that Fig. 4B used tobacco with transiently expressed BrJMJ18. Then, how can the plants be exposed to 3 weeks of NC and 3 days of HS? Are these stable transgenic lines?

1-c. I do not think the designation of polymorphic alleles of BrJMJ18 as wt vs. par is appropriate. The "Par" version of BrJMJ18 is not a mutant. Perhaps DG vs. Par (or PC vs. Par) should be considered.

2. Re: Table S8: CHIP-seq of BrJMJ18.

2-a. It appears that the authors generated a polyclonal antibody against BrJMJ18. However, no data is presented to show the specificity of the antibody used in CHIP-seq experiments.

2-b. It was not possible for me to evaluate the quality of this CHIP-seq experiment. There is no indication of the number of replicates, and it is also unclear what was used for the control.

2-c. Fig. 5A used GADPH as a loading control for BrJMJ18 CHIP-qPCR validation at FLC3. Is the GADPH also enriched by BrJMJ18? From CHIP-seq data, the authors should have been able to extract loci that are enriched with BrJMJ18 regardless of conditions, and those loci are better controls.

3. Re: Flowering time of various genotypes and transgenic plants

3-a. Fig. S11-B: It is unclear why the authors compared the flowering times of vernalized plants with high temperature-grown plants, and it cannot be interpreted as "Line 267 - in which heat stress (HS) conditions caused early flowering in BrJMJ18WT-carrying plants, but delayed flowering in BrJMJ18Par-

carrying plants". The authors are comparing two different treatments.

3-b. Flowering time as DAS (the day after sowing). Do HS conditions the authors used (either greenhouse or field conditions) cause any differences in growth (or developmental) -rate? HS condition the authors described (Fig. S4) shows apparent detrimental effects on plant growth.

4. Fig. 7

4-a. Similar to point #2, I could not evaluate the quality of this ChIP-seq experiment.

4-b. Line 498: "32.8% of the enriched genes in BrJMJ18WT-OX plants and 28.6% in BrJMJ18Par-OX plants under NC overlapped with reported H3K36me3-targeted genes, while the corresponding frequencies under HS decreased to 22.1% and 20.0%: Have the authors evaluated the association with other modifications (i.e., H3K4me2/3, H3K27me2/3, etc.). It is also quite surprising that more than 70% of BrJMJ18WT and BrJMJ18-Par are unique. Are the authors suggesting two variations of BrJMJ18 have evolved to target quite different sets of genes?"

4-c. Line 515: "18.2% of the DEGs in BrJMJ18WT-OX plants and 17.8% of the DEGs in BrJMJ18Par-OX plants are BrJMJ18-target genes identified by ChIP-seq under NC, while the corresponding frequencies under HS decreased to 5.0% and 5.7%." It is not clear whether these overlaps are significant. How many genes that are identified as BrJMJ18 targeted are also DEGs?

5. Fig. 8

5-a. Using a heterologous system to show biochemical specificity is not desirable, and it is hard to interpret the result. The assumption here is that in vivo system of Arabidopsis and *B. rapa* is similar to each other, but the authors concluded that BrJMJ18 is different from its Arabidopsis counterpart. Are they affecting AtFLC by acting as H3K36me2/3 demethylases or as H3K4me2/3 demethylases?

5-b. BrJMJ18Par-OX dissociates more from FLC3 in HS conditions, but the catalytic activity is better. Are these variations contributing to BrJMJ18 bindings?

6. Figure legends should be carefully revised. For example, Fig. 2 legend: There are no horizontal blue lines – they are pink. There is no pink bar(perhaps peach bar?). There is no indication of what blue and orange Fst values represent.

7. Line 312: "We also analyzed the flowering time of BrJMJ18 overexpression lines (AtJMJ18pro::BrJMJ18CDS: GFP) in Arabidopsis".

These are not "overexpression lines". The authors used the Arabidopsis JMJ18 endogenous promoter.

8. One of the conclusions by the authors is that FLC3 is adapted to mediate HS-specific delay in flowering in Par variations. flc3 mutation in Par varieties can support it.

Reviewer #4 (Remarks to the Author):

This is a very interesting paper, in which the authors identify an H3K36me3 demethylase in *Brassica rapa*. Natural variation in this protein contributes to the temperature response of flowering time, specifically in relation to elevated temperatures. Through a considered, step-by-step approach, the authors show first that there is a difference in flowering time specific to the Par genotype in warm conditions. They then identify the gene and show it has H3K36me3 demethylase activity. They record flowering time, gene expression, histone modifications and protein binding to show that the action of this gene strongly affects flowering response in high temperatures, likely through the demethylation of H3K36me3 at the BrFLC3 gene.

Major comments

1. Can you clarify a bit more in the introduction what the desired trait is in Par? It seems that flowers are wanted in warm conditions, without vernalization, but at the same time a late flowering natural allele is favourable. Is longer vegetative growth desirable?
2. Related to that and to point 3 below, it is not clear if the focus is thermotolerance/heat stress or flowering time.
3. I am not sure if the term “thermotolerance” is appropriate for the HS conditions used, at 29°C. Heat stress studies as far as I am aware are usually done in much higher temperatures (above 30°C, and normally around 40°C, as in Fig. S4 in this manuscript). The result of Par being more thermotolerant as shown in Fig. S4 is convincing, however the link between this increased tolerance and BrJMJ18 is not made. The terms “thermotolerance” and “heat stress” are not absolute and they depend on the optimum range of the plant in question as well as the temperature range in the region where it is grown. I would suggest some more explanation to be added to justify the choice of these temperatures and possibly changing the terms “thermotolerance” and “heat stress” if appropriate.
4. It needs to be clarified which of the different versions of HS was actually used in the different experiments (e.g. 29/29°C or 29/22°C). Not consistent night temperature and pre-treatment between Methods section, main text, Fig. 2D and table S6. Vernalization treatment is likely to be significant and should be clarified where this was used.
5. Some results are hidden in tables (e.g. different expression of BrJMJ) while there are supplementary figures that are not necessarily adding anything more than the numbers in the text could give (e.g. S6 and perhaps also S10).
6. If BrJMJ18Par is deactivated by warm, then why do the CRISPR, OX and wt Par not all have the same phenotype? Or if it is not fully deactivated, why is wt Par not intermediate between these? In fact, why would overexpression of a K36 demethylase, even an inactive one, lead to an increase in FLC expression and H3K36me3 levels? Could you add some more discussion around these points and any possible explanations? What other genes are affected? (Table s9?)

7. Claims of tight association between BrJMJ18 and BrFLC3 based on diurnal expression must be weakened, since it not excluded (and indeed seems likely) that they could both be under circadian regulation by the clock. Expression of clock genes could be shown to distinguish between claim and heat-shock misregulation of circadian clock. Furthermore, strong diurnal regulation of these genes suggests claims about expression in different subspecies should be tested with measurements at different times of the day, since expression is so sensitive to this, or at least report time of sampling for all work and use the same for comparisons.

Minor comments

Table 1: Not consistent use between π and PI.

Fig. S1A: missing X-axis label. Also, it is unclear if the finally selected K is 5, 6 or 7.

Fig 2 A, B: No horizontal blue line. Also “ π values of the corresponding comparisons are shown at the bottom of (A) and (B), respectively” I could not find these. Also, no x-axis labels.

Lines 188-190: I could not find the data related to the BSA-seq approach including the flowering time phenotypes in different conditions. If authors feel it is beyond the scope or unnecessary to report this fully, then perhaps some summary information could be added in text or diagram form.

Fig. 2D: 99% or 95% threshold?

Fig. S7: No “24h after treatment”.

Table S7 s4, line 238 and line 248: In table S7 s4, BrJMJ18 is not differently expressed in PC during flowering, unlike the statement in line 238. Also is there no significant difference among subspecies during flowering (as stated in line 248)? This is surprising considering the large difference between PC and the other two.

Fig. S11: What is different in S11A compared to 1F that means the DG accessions are now flowering?

Fig. 3A: Is the scalebar 2cm?

Fig. 3G: Where is the broadened flowering time delay of ParBrJMJ18CR plants under HS shown? In HS, the CRISPR line never flowered, so how do you explain that in the tunnel greenhouse in the summer of 2021 it did? And why did the BrJMJ18WT-OX not flower early there?

Fig. 4B: Activity under NC also appears different in tobacco leaves.

Fig. 5B: Please indicate bolting time here.

Fig. S16: As the role of FLC is in the initiation of flowering (at least in Arabidopsis), why is FLC investigated after bolting? Also, no y-axis numbers.

Fig. S17: AtFT levels do not seem to match flowering time pattern. Can you comment?

Fig. 5F: It is surprising that there is no FLC3 expression in DG (according to Fig. S16) but high K36. Can you comment?

Fig. S19: Claims are too strong since the results are not consistent in the two lines.

Fig. 7A, B: Why are there so few overlapping? Is this expected? In NC the plants generally behaved similarly, right?

Fig. S23: Not clear what is shown here. Can you clarify in text and with arrows or clearer image?

Fig. 8A: It seems that maybe the terms "activity" and "H3K36 signal" are mixed up in the discussion of this figure and in figure legend. e.g. where it is said that in NC F654L had higher activity than WT.

Reviewer #5 (Remarks to the Author):

Note to the authors: I have been asked to review the population genetics part only, as there was obviously a gap in the reviews before. I therefore read the whole manuscripts, but focus my review to the first part.

Some typos:

Abstract

l27: provide, not provides

Introduction:

First sentence confusing, please rewrite

l52 exists, not exits

More critical:

MM:

The methods used to perform the QTL study are not described. There is a hint that population genetics were performed as described in Su et al 2018, but the description there does not fit to the presented figure (2D), as this presents physical distances and SNP indices instead of genetic distances (in cM) and a LOD score. So please either clarify the figure or add a proper description of the method.

Moreover, looking at Supplementary Figure S8, it looks as if the JMJ8 locus is not closely linked to the QTL, at least falls out of the major LD group. Figure S8 is also not really described, the description keeps talking about a qQTL (which indicates a phenotype-genotype-association) but only π -values (nucleotide diversity) is shown.

Table S6 also does not show LOD scores for the respective peaks. In other words, the statement that this

QTL was the highest is not shown here.

Figure 1A: really small, not really readable. I would advise either to move to Supplementary or to change size.

Figure 1B: please add the phylogenetic outgroup to the caption.

Figure 1C: no black arrows as indicated in the caption

Figure 1D: do the estimated times since the respective splits make sense in a historical context? This should possibly be added to the discussion.

Figure 1 E: I personally find it hard to grasp the message here. What's the information in the vertical direction?

Figure 2A. the horizontal line is red, not blue

Figure 2C this does not show genes, but the results of GO enrichment, caption has to be adapted

Figure 2D: this is at least an unusual QTL representation, why isn't it presented as LOD score, and why not over the full chromosome?

Figure 2F: the caption should explain what JmjN, JmjC, Zf-C5HC, FYR and FYRC stands for.

Figure 2G; Was there a difference in FTi in the DG lines carrying the Par allele? Did the three DG lines flowering without vernalisation (Figure 1F) carry a Par allele?

Reviewer #1 (Remarks to the Author):

This manuscript study the response of B. rapa to high temperatures. Through QTL and genomic analyses of a set of B. rapa accessions, the authors found a number of SNPs in an homolog of At.JMJ18 in the thermotolerant cultivar B. rapa var. parachinensis. Then, they performed an impressive work including enzymatic assays, transgenics in B. rapa and A. thaliana, RNA-seq, CHIP-seq experiments to characterize Br.JMJ18 as a novel regulator of the thermosensory pathway in plants. They conclude that in most B. rapa accessions under, heat promotes Br.FLC3 downregulation by Br.JMJ18. However, in B. rapa var. parachinensis, heat impairs Br.JMJ18 target-binding promoting vegetative growth and late flowering time.

Although the work is very relevant and the results are quite complete. I found that conclusion are not satisfactory and some points require clarification prior publication.

RE: We thank you for your insightful comments and suggestions. We have implemented all of your applicable recommendations to improve the quality of the presented work, please see details in our point-by-point response below. With these improvements, we hope that you will find the manuscript much improved.

Major concerns

1) First, it is not clear to me what happen with B. rapa (in general) flowering time at high temperature? The final model suggest that heat may accelerate flowering in "Br.JMJ18-wt" accessions (pack choi, dark green, etc.) but no data is provided. Current published data shows that some B. rapa varieties delay flowering at high ambient e temperature (Chen et al genes 2022, Del Olmo et al Plan Journal 2019).

RE: According to the requirement for vernalization, *B. rapa* crops can be divided into two classes: vernalization-requiring and non-vernalization-requiring morphotypes. For vernalization- requiring plants, the majority of this morphotypes experience late flowering when exposed to high temperatures because high temperatures induce de-vernalization (Saito and Saito, 2003; Xiao-Li et al., 2013). On the other hand, those morphotypes that do not require vernalization may either experience early or late flowering when exposed to high temperatures. In this scenario, the timing of high temperatures is crucial, as the effects of heat stress during the seedling stage (vegetative growth period) and flowering (reproductive growth period) stage differ. Generally, heat stress during the seedling stage affects flowering time, while heat stress during the flowering stage only impact seed yield.

Currently, there is more research focusing on the effects of heat stress during the flowering stage on agronomic traits such as seed yield (Chen et al., 2022). Chen found a significant genetic variation in heat tolerance and sensitivity for above-ground biomass, whole plant seed yield, harvest index, and seed yield of five pods on the main stem at maturity. However, it didn't address flowering time changes due to heat

stress. While in the study by Del Olmo et al. (2019), the authors compared the flowering time of R-o-18 plants grown at 21°C and 28°C. R-o-18 is an *oilseed rapa* variety. Interestingly, they observed that plants grown at 28°C flowered, on average, up to 8 days later than those grown at 21°C and produced more leaves at bolting. Therefore, the *oilseed rapa* morphotype grown at higher temperatures retained the ability to flower but experienced a delay in floral transition.

In some studies on vegetative rapa, such as summer-sowing Chinese cabbage and *B. rapa* subsp. *chinensis* (PC), exposure to high temperatures during the seedling stage resulted in premature bolting, affecting yield and quality (Yui and Yoshikawa, 1992; Wei et al., 2022; 花萌 and 李恺文, 2022). This finding aligns with our results. Additionally, study by Rameeh (2012) that investigated the effects of high temperature on the flowering time of Indian mustard (*B. juncea*). The study found that late-planted Indian mustard experienced early onset of flowering due to terminal heat stress.

We added the statement above into the introduction (lines 53-74).

References:

- Saito, H., and Saito, T. (2003). Effects of High Temperature Interruption during Vernalization on the Inflorescence Formation in Turnip Plants. *Engei Gakkai Zasshi* 72, 329-334.
- Xiao-Li, W., Yong, Z., and Yao-Wei, Z. (2013). Effects of Chinese Cabbage Devernalization and DNA Methylation Analysis before and after Devernalization. *China Vegetables*.
- Chen, S., Hayward, A., Dey, S.S., Choudhary, M., Witt Hmon, K.P., Inturrisi, F.C., Dolatabadian, A., Neik, T.X., Yang, H., Siddique, K.H.M., et al. (2022). Quantitative Trait Loci for Heat Stress Tolerance in *Brassica rapa* L. Are Distributed across the Genome and Occur in Diverse Genetic Groups, Flowering Phenologies and Morphotypes. *Genes* 13.
- Del Olmo, I., Poza-Viejo, L., Piñeiro, M., Jarillo, J.A., and Crevillén, P. (2019). High ambient temperature leads to reduced FT expression and delayed flowering in *Brassica rapa* via a mechanism associated with H2A.Z dynamics. *The Plant journal : for cell and molecular biology* 100, 343-356.
- Wei, Q., Hu, T., Xu, X., Tian, Z., Bao, C., Wang, J., Pang, H., Hu, H., Yan, Y., Liu, T., et al. (2022). The New Variation in the Promoter Region of FLOWERING LOCUS T Is Involved in Flowering in *Brassica rapa*. *Genes* 13.
- Yui, S., and Yoshikawa, H. (1992). Breeding of Bolting Resistance in Chinese Cabbage. Critical Day Length for Flower Induction of Late Bolting Material with No Chilling Requirement. *Engei Gakkai Zasshi* 61, 565-568.
- 花萌, and 李恺文 (2022). 夏季大白菜未熟抽薹现象研究. *青海农林科技*, 60-63.
- Rameeh, V. (2012). Correlation analysis in different planting dates of rapeseed varieties. *Journal of Agricultural Sciences*

With regarding of the concern of “*The final model suggest that heat may accelerate flowering in “BrJMJ18^{PC}” accessions (pack choi, dark green, etc.) but no data is provided*”, we randomly selected four accessions (PC032, a BrJMJ18^{PC}-carrying PC line (BrJMJ18^{WT} was replaced BrJMJ18^{PC} in the new text as suggested by reviewer 3); DG109, a BrJMJ18^{PC}-carrying DG line, DG016, a BrJMJ18^{Par}-carrying DG line, and Par110, a BrJMJ18^{Par}-carrying Par line) from our germplasm collection (Table S1) to further explore the impact of high temperatures on flowering time in the PC and DG groups. After germination, seedlings were grown in NC conditions (21°C/21°C, 16 h/8 h light period) for 4 weeks. Half of the seedlings were then transferred to HS conditions (29°C/29°C, 16 h/8 h light period) until flowering. We found that BrJMJ18^{PC}-carrying PC and DG accessions flowered earlier under HS conditions compared to NC. Meanwhile the BrJMJ18^{Par}-carrying Par accession delayed flowering under HS conditions compared to NC. And the BrJMJ18^{Par}-carrying DG accession did not bolt under HS in our experiment. These results are consistent with our working model

in which heat delays flowering in BrJM18^{Par} accessions. We put the data in the new figure S4 and a description in the text (lines 179-181).

Therefore, the influence of high temperatures on flowering time depends on multiple factors, including the timing of the treatment, the morphotype and the variety being studied.

Figure S4. *Par* showed higher thermotolerance comparing with that of *PC* and *DG*.

(A) For heat shock treatment, 14-day seedlings grown under normal conditions (NC) were moved to NC and heat-shock conditions (16/8 h day/night, 42°C/42°C), respectively, for another one week. After the heat treatment, the plants were recovered at 22°C for 5 days. At the end of recovery, photographs were taken. (B) For heat stress treatment, PC032, a BrJM18^{PC}-carrying *PC* line, DG109, a BrJM18^{PC}-carrying *DG* line, DG016, a BrJM18^{Par}-carrying *DG* line, and Par110, a BrJM18^{Par}-carrying *Par* line were randomly selected from our germplasm collection. 4-week-old seedlings grown under NC were moved to NC and heat stress (16/8 h day/night, 29°C/29°C, HS) conditions, respectively until flowering. DG106 under HS condition did not flower within the observe window of 120 days. (C) Flowering time of plants shown in (B). Flowering time of DG106 under HS condition was set to 120 days. The box encompasses two middle quartiles, with central line showing median. Whiskers extend to the furthest data point within 1.5 times the interquartile range. Asterisks indicate significant differences, Student's t-test (* $p < 0.05$, ** $p < 0.01$, *** $p < 0.001$).

2) Across the manuscript, the authors talk about "Heat Stress" treatment. But it believe it is a treatment at 29C. Is that really heat stress? Should it not be called heat treatment or warm temperature? Please explain this in the text and include the temperature of the treatment in the figure legends.

RE: Most of the *Brassica rapa* crops originated from the Mediterranean coast, belonging to temperate climate zones, and they are highly sensitive to heat stress. According to "The Plant Family Brassicaceae" book (Anjum et al., 2012), temperatures with a day-night average higher than 30°C can be considered as heat stress for Brassicaceae plants. While as reported by Morrison and Stewart (2002), during the flowering stage, the critical threshold temperature causing seed yield losses for all Brassica species was found to be 29.5°C. Mean maximum temperatures more than 29°C during vegetative development led to a decline in flower numbers across all Brassica species. As the intensity of heat stress during flowering increased, seed yield showed a corresponding decrease. Also as you reminded, the definitions of "thermotolerance" and

"heat stress" are influenced by the plant's optimal range and the temperature conditions in its cultivation region. To address this, we referred to data from the China Meteorological Administration regarding average temperatures in Guangdong province, the area of *Par* domestication, for the summer months from July to September. Over the last decade, the average summer temperature has stood at 29.4°C. By integrating this information, we propose that around 29°C can be regarded as the critical temperature for *Par* domestication. In this study, our heat treatment involved maintaining a consistent day-night temperature of 29°C, which can be deemed as inducing heat stress for *Par*.

We revised and added the statement above into the introduction (lines 75-84)

References:

Anjum, N., Ahmad, I., Me, P., Duarte, A., Umar, S., and Khan, N. (2012). The Plant Family Brassicaceae: Contribution Towards Phytoremediation.
Morrison, M.J., and Stewart, D.W. (2002). Heat Stress during Flowering in Summer Brassica. *Crop Science* 42, 797-803.

3) *To my eyes, plants that carry BrJMJ18-par or CRISP allele look stressed under high temperature (Figure 3).*

However this phenotype is not discussed in the text.

RE: The reviewer's comments appear to refer to the comparison between the *BrJMJ18^{PC}* and *BrJMJ18^{Par-CR}* plants. From the Figure 3, we can observe that under NC conditions, there is little difference in the growth among *BrJMJ18^{PC-OX}*, *BrJMJ18^{Par-OX}*, and *BrJMJ18^{Par-CR}* plants. However, under HS conditions, *BrJMJ18^{PC-OX}* and *BrJMJ18^{Par-CR}* exhibit signs of stress.

In terms of mechanisms, under NC conditions, *BrJMJ18^{PC}* and *BrJMJ18^{Par}* share a strong overlap in their regulated genes. GO analysis shows that these genes function predominantly in flower development, stress response, and plant hormone signaling pathways. While under HS conditions, the target genes bound by *BrJMJ18^{PC}* and *BrJMJ18^{Par}* are evidently different. In summary, *BrJMJ18^{PC}* functions as a "throttle" for flowering, regardless of whether it is under NC conditions or HS. On the other hand, *BrJMJ18^{Par}* exhibits functional divergence at different temperatures. Under NC conditions, *BrJMJ18^{Par}* promotes flowering, but under high temperature, it acts as a "brake" for flowering. This working mechanism can explain why under HS conditions, *BrJMJ18^{PC-OX}* or *BrJMJ18* knockout *Par* plants appear stressed, whereas *BrJMJ18^{Par-OX}* plants did not. We revised and added the statement above into the result and discussion (lines 444-446, 453-459 and 605-610.)

4) *Line301. The transgenic line production should be explained and introduced in the main text not in the figure 3 legend.*

RE: The description of "The open reading frames (ORFs) of *BrJMJ18^{PC}* and *BrJMJ18^{Par}* driven by the cauliflower mosaic virus 35S promoter (35S), respectively, were transformed into *Par* plants, Transgenic T1 lines with similar protein expressions were used for study" was moved to the main text. Please see lines

5) From line 488-556. All this section, is inconclusive. The genomic analyses are great, but the concoctions are not solid and needs rewriting. For example did they explored is there is a correlation between heat responsive genes and BrJMJ18 binding or misregulated genes? In my opinion the sentences "We thus speculated that BrJMJ18Par promotes vegetative growth probably through regulation of cell division under heat stresses" is too much speculation. At least, to be repeated as a solid conclusion of the work.

RE: Thank you for the insightful comment. As you pointed out, we recognize that the conclusion "We thus speculated that BrJMJ18^{Par} promotes vegetative growth probably through regulation of cell division under heat stresses" might be fairly presumptuous based on the current data and analysis, and we deleted in the new text.

To enhance the robustness of our findings, we incorporated a gene set related to heat stress in *B. rapa* from the study by Yue et al. (2021) and Zhang et al. (2022). In their research, Yue et al. identified 5,559 DEGs in a heat-resistant Chinese cabbage (268) after 8 days of heat stress. By comparing our RNA-seq results with theirs, we found that 35.07% and 29.32% of the BrJMJ18^{PC-OX} and BrJMJ18^{Par-OX} DEGs under NC conditions overlapped with the heat response genes reported by Yue et al. These additional findings

Table S10

Differentially expr

have been integrated into Table S10. According to Zhang et al.'s report, downregulated genes under heat stress in the heat-resistant Chinese cabbage (268) were notably enriched in pathways such as photosynthesis, porphyrin and chlorophyll metabolism, and carbon metabolism. These findings also align with the GO analysis of downregulated genes in BrJMJ18^{Par-OX} under heat stress.

Subsequently, we conducted Q-PCR to assess the expression of six heat stress-related genes associated with chlorophyll biosynthesis and carbon metabolism, as reported by Zhang et al., in *Par* and BrJMJ18^{PC/Par-OX} plants. Relative to control *Par* plants, all six genes exhibited slight induction in BrJMJ18^{PC-OX} plants under heat stress, while they were significantly downregulated in BrJMJ18^{Par-OX} plants. These results validate our hypothesis that BrJMJ18^{Par} enhances thermotolerance by inhibiting photosynthesis and chlorophyll biosynthesis. We have included these data and details in both the text (lines 494-496 and 514-523) and Figure 7 for clarity.

Figure 7 (G) Expression of heat stress-related genes reported by Zhang et al in *Par* and *BrJMJI8-OX* plants. 5-week-old plants grown under NC, and 4-week-old plants grown under NC followed by 1 week under HS were used for Q-PCR. Relative to *Par* plants, all six genes exhibited slight induction in *BrJMJI8^{PC}-OX* plants under heat stress, while they were significantly downregulated in *BrJMJI8^{Par}-OX* plants. Photosynthesis related gene: BraA9g029800.3C (ATP synthase DELTA-subunit gene, ATPD)", porphyrin and chlorophyll metabolism related gene: BraA04g028660.3C (Uroporphyrinogen decarboxylase, HEME2) and carbon metabolism related genes: BraA07g034950.3C (pyruvate dehydrogenase E1a-like subunit, IAR4), BraA03g035490.3C (phosphoglycerate kinase 1, PGK1), BraA07g0022160.3C (sedoheptulose- biphosphatase, SBPASE), BraA08g004460.3C (Embryo defective 3105, EMB3105). *GADPH* was use as internal control. The values are the mean \pm standard deviation from three biological replicates.

References:

Yue, L., Li, G., Dai, Y., Sun, X., Li, F., Zhang, S., Zhang, H., Sun, R., and Zhang, S. (2021). Gene co-expression network analysis of the heat-responsive core transcriptome identifies hub genes in *Brassica rapa*. *Planta* 253, 111.
 Zhang, L., Dai, Y., Yue, L., Chen, G., Yuan, L., Zhang, S., Li, F., Zhang, H., Li, G., Zhu, S., et al. (2022). Heat stress response in Chinese cabbage (*Brassica rapa* L.) revealed by transcriptome and physiological analysis. *PeerJ* 10, e13427.

6) Line 356 *ICU11* is also proposed to be an *H3K36me3*, please cite Bloomer et al PNAS 2022.

RE: In Bloomer et al PNAS 2022, they found that *Arabidopsis ICU11* encodes a 2-oxoglutarate-dependent dioxygenase (2OGD). The 2OGD domain of *ICU11* belongs to the same enzymatic superfamily as Jumonji C-domain histone demethylases. Furthermore, *ICU11* demonstrates a strong interaction with PRC2 core components and binds to the nucleation region of *FLC*, thereby facilitating the demethylation of *H3K36me3*. This is substantiated by the observation that the *icu11* mutant exhibits a slight elevation in *H3K36me3* levels. We have amended the aforementioned description and incorporated it into lines 616-618.

Reference:

Bloomer, Rebecca H., et al. "The *Arabidopsis* epigenetic regulator *ICU11* as an accessory protein of Polycomb Repressive Complex 2." *Proceedings of the National Academy of Sciences* 117.28 (2020): 16660-16666.

7) Line 350: *Br.FLC3* is also crucial to explain the flowering phenotype of *Br.ELF6* and *Br.REF6* in yellow sarson. Please cite Poza-Viejo et al. *Plant Cell&Env.* 2022

RE: In Poza-Viejo et al. (2022), they found that in *yellow sarson*, ELF6 controls flowering by regulating *BrFLC3* expression. The early flowering of *braA.elf6* mutants is linked to elevated H3K27me3 levels at the *BrFLC3* locus. In contrast, delayed flowering in *braA.ref6* mutants is associated with impaired GA production, rather than *BrFLC3* induction. Interestingly, *yellow sarson* is also a type of *B. rapa* that can flower without vernalization.

We have amended the aforementioned description and incorporated it into lines 384-385 and 679-680.

Reference:

Poza-Viejo, Laura, et al. "Conserved and distinct roles of H3K27me3 demethylases regulating flowering time in *Brassica rapa*." *Plant, cell & environment* 45.5 (2022): 1428-1441.

8) Lines 677-680 need to be rewritten, I do not see the point. ELF6 and REF6 flowering time is not confusing: ELF6 regulates FLC directly, REF6 does not regulate FLC directly but is late because it has high levels of FLC. Triple and double H3K27 histone demethylase mutants have severe developmental phenotypes (see Yan et al. *Nat Plants* 2018).

RE: I apologize for the imprecise presentation. After a review of the literatures, we found that the loss-of-function mutants of two known H3K27me3 demethylases, ELF6/JMJ11 and REF6/JMJ12, exhibited early and late flowering phenotypes, respectively, in *Arabidopsis*. Notably, the *ref6-1 jmj30-2 jmj32-1* triple mutant displayed a late-flowering phenotype without significant developmental defects. And it is also worth noting that triple and double mutants of H3K27 histone demethylases show substantial developmental phenotypes (Yan et al., *Nat Plants* 2018).

However, as pointed out by the reviewer, this section of the text lacks a strong logical connection with the core topic we intend to discuss "*BrJMJ18* is an H3K36me3/2 demethylase". Consequently, we have decided to omit the related description in the new text (please see line 647 in the Word's tracking mode).

Reference:

Yan, W., Chen, D., Smaczniak, C., Engelhorn, J., Liu, H., Yang, W., Graf, A., Carles, C.C., Zhou, D.X., and Kaufmann, K. (2018). Dynamic and spatial restriction of Polycomb activity by plant histone demethylases. *Nature plants* 4, 681-689.

9) Line 749. Please provide details about the RNA-seq methods (library preparation, sequencing length, bioinformatics analyses)

RE: Thank you for the insightful comments. RNA-seq transcriptome library was prepared following TruSeq™ RNA sample preparation Kit from Illumina (San Diego, CA) using 1µg of total RNA. After quantified by TBS380, paired-end RNA-seq sequencing library was sequenced with the Illumina HiSeq

xten/NovaSeq 6000 sequencer (2 ×150bp read length). The raw paired end reads were trimmed and quality controlled by SeqPrep (<https://github.com/jstjohn/SeqPrep>) and Sickle (<https://github.com/najoshi/sickle>) with default parameters. Then clean reads were separately aligned to reference genome with orientation mode using HISAT2 (<http://ccb.jhu.edu/software/hisat2/index.shtml>) software. The mapped reads of each sample were assembled by StringTie ([https://ccb.jhu.edu/software/stringtie/index.shtml? t=example](https://ccb.jhu.edu/software/stringtie/index.shtml?t=example)) in a reference-based approach. To identify DEGs (differential expression genes) between two different samples, the expression level of each transcript was calculated according to the transcripts per million reads (TPM) method. RSEM (<http://deweylab.biostat.wisc.edu/rsem/>) was used to quantify gene abundances. Essentially, differential expression analysis was performed using the DESeq2[4]/DEGseq[5]/EdgeR[6] with Q value ≤ 0.05 , DEGs with $|\log_2FC| > 1$ and Q value ≤ 0.05 (DESeq2 or EdgeR) / Q value ≤ 0.001 (DEGseq) were considered to be significantly different expressed genes). In addition, functional-enrichment analysis including GO and KEGG were performed to identify which DEGs were significantly enriched in GO terms and metabolic pathways at Bonferroni-corrected P-value ≤ 0.05 compared with the whole-transcriptome background. GO functional enrichment and KEGG pathway analysis were carried out by Goatools (<https://github.com/tanghaibao/Goatools>) and KOBAS (<http://kobas.cbi.pku.edu.cn/home.do>).

We add the above details about the RNA-seq in the Method section (lines 795-813).

10) Line 985. *It is prodigious how many B. rapa transgenic line were produced. Please provide a more detailed protocol. This will help the entire scientific community.*

RE: *B. rapa* transgenic plants were constructed as following:

Agrobacterium Preparation: Transform *Agrobacterium* strain GV3101 via electroporation with the constructs of interest. Inoculate single colonies into 10 mL LB cultures (50 mg/L kanamycin and 50 mg/L rifampicin) and incubate at 28 °C, 220 rpm for 48 h. Centrifuge the culture at 5,000 rpm for 15 min, re-suspend the pellet in liquid MS at OD₆₅₀=0.05.

Seed Sterilization and Germination: Sterilize *Par* inbred line 16A-1 seeds with 75% ethanol (1 min) and 5% sodium hypochlorite (15 min). Germinate seeds on GM plates at 25°C for 4 days.

Explant Isolation, Inoculation, and Co-cultivation: Cut 1–2 mm cotyledonary petioles of 4-day-old seedlings and place on CIM-C plates. Inoculate by dipping petiole ends in *Agrobacterium* suspension. Return explants to the same CIM-C plates and incubate in a growth chamber at 25°C in dim darkness for 72 h.

Selection and Transfer of Plants to Greenhouse. After 72 h on CIM-C plates, the explants were transferred to the SIM plates and moved to a growth chamber at 25/22°C with a 16/8-h (light/dark) photoperiod (photon flux density 100 mol/m²/s) for 2 weeks. Then the explants were moved to fresh SIM

plates for further 1-3 weeks until the emergence of green shoots. Green shoots were transferred to SEM plates for further 1-2 weeks at the same growth chamber. The well-developed green shoots were transferred to the RM plates at the same growth chamber for 1-3 weeks. Rooted green plants were acclimatized in a growth chamber under 25°C with 70–80% humidity for 2 days and then transferred into soil.

Transgenic Plant Verification: For overexpression lines, transgene expression was confirmed by immunoblotting with anti-GFP antibody (HT801-01, TransGene, Beijing, China) for *BrJMJ18^{PC/Par}-OX* plants. T0 plants of both lines were self-crossed to generate seeds. In the case of CRISPR/Cas9 plants, the target gene *BrJMJ18* from T0 plants was amplified, purified, and cloned into a vector using the pMDTM18-T Vector Cloning Kit (Takara) for sequencing. Progeny T1 seedlings were grown on 0.5 × MS medium without kanamycin. Loss-of-function *BrJMJ18* seedlings, confirmed through PCR and Sanger sequencing, were used for subsequent analysis.

Plant culture Media were listed as below:

Liquid MS: 4.3 g/L Murashige and Skoog (MS) basal salts, 30 g/L Sucrose, pH 5.7.

Germination media (GM): 4.3 g/L Murashige and Skoog (MS) basal salts, 30 g/L Sucrose, pH 5.7, 8 g/L agar.

Callus induction media for cocultivation (CIM-C): GM plus 500 mg/L 2-(N-morpholino)ethanesulfonic acid (MES), 4mg/L 6-BA, 0.667mg/L IAA.

Shoot induction media (SIM): CIM-C medium plus 160 mg/L Timentin, 2 mg/L AgNO₃ and 25 mg/L Kan.

Shoot elongation media (SEM): GM plus 500 mg/L MES, 160 mg/L Timentin, 50 µg/L 6-BA, 2 mg/L AgNO₃ and 25 mg/L Kan.

Rooting medium (RM): 3.05 g/L Gamborg's B5 salts, 10 g/L sucrose, pH5.7, 8 g/L agar, 160 mg/L Timentin, 1mg/L IBA, 2 mg/L AgNO₃ and 25 mg/L Kan.

We have amended the aforementioned description and incorporated it into lines 858-896.

11) *Line 1008: Could you explain why the genomic data is not deposited in public databases?*

RE: The genomic data is now accessible on NCBI under the GEO accession number GSE223969.

Minor points

12) *References 1 and 2 are about tomato. I think there are not related to this work.*

RE: Sorry for the incorrect citations. We have made the necessary corrections to references 1 and 2.

References:

- Elvis, K., Quezada, D., Ihien, E., Vasquez-Teuber, P.&Mason, A. Interspecific Hybridization for Brassica Crop Improvement. *Crop Breeding, Genetics and Genomics* 2020).10.20900/cbagg20190007
- Qi, X., et al. Genomic inferences of domestication events are corroborated by written records in *Brassica rapa*. *Mol Ecol* 26, 3373-3388(2017).10.1111/mec.14131

13) Line 18. The authors write about *B. rapa parachinensis* as Chinese kale, I believe that a kale is *B. oleracea*. I understand the in English name for *parachinensis* is Chinese cabbage or Caixin. Please revise that.

RE: Sorry for the mistake. We corrected Chinese kale to Caixin in the text (line 19).

14) Fig1 Could you please put a bigger world-wide map in Fig1?

RE: As suggested by the reviewer 5, the original Figure 1A and C have been moved to the supplementary Figure S1A and C, respectively. We have also revised Figure 1 to ensure its readability by enlarging it appropriately.

15) Fig2 There are two QTL (A3 and A7) that the authors do not even mention. Any speculation about that.

RE: Among the 24 loci, there are 4 that exhibit strong signals: A03 (635,000-685,000, encompassing 8 genes), A07 (28,335,001...28,930,000, including 69 genes), A09 (18,815,000...18,935,001, with 14 genes), and A09 (26,630,001...27,505,000, containing 77 genes) (Fig 2A, Table S5). Within the A03 loci, 3 out of the 8 genes are differentially expressed during flowering of *DG* and *Par* (Table S3), all of which play a role in DNA topological changes. In the case of the A07 loci, 23 out of the 69 genes exhibit differential expression during flowering of *PC*, *DG* and *Par* (Table S3). These genes are associated with various processes such as oxidation-reduction, protein phosphorylation, multicellular organism development, proteolysis, base-excision repair, regulation of transcription, cellular response to DNA damage stimulus, and arginine metabolic processes. Whether these genes are involved in flowering and/or heat stress response remains unclear.

16) Fig3C I think that "x-axis" legend is missing

RE: The *x-axis* represents DAS (days after sowing), and we added it in the text (line 268).

17) I found the discussion a bit long. I think that some parts of the discussion could be shortened a few sentences and the paper would be improved.

RE: The "BrJMJ18 is an H3K36me3/2 demethylase" and "Functional divergence of BrFLCs" sections in the Discussion were significantly revised and condensed. Additionally, the content from the section "Plants respond to high temperature by remodeling chromatin on a genome-wide basis" was integrated

into the aforementioned two sections. Please see details in the Discussion section in Word's tracking mode.

Reviewer #2 (Remarks to the Author):

Flowering time is essential for crop yield and the response to global warming. This manuscript aims to identify the genetic mechanisms of flowering time regulation and heat stress tolerance in Brassica rapa with considerable work and multiple approaches. However, after reading the manuscript, I feel the key message of the story, and the information flow is not clear, leading to a lot of issues in understanding the story. Here are my main comments:

RE: We thank you for your insightful comments and suggestions. We have implemented all of your applicable recommendations to improve the quality of the presented work, please see details in our point-by-point response below. With these improvements, we hope that you will find the manuscript much improved.

1. The story starts with the domestication analysis of Par, DG, and PC. When we look at the place of the collected accession, we can see most of the par accessions are from the south of China, which differs from DG and PC. Therefore, I doubt the different flowering time of Par is domestication from DG and PC or a local adaptation of collected Par accession. Moreover, it is also unclear whether the results in Figure 1F indicated the flowering time in all collected Par, DG, and PC accessions or only representative accessions.

RE: The materials for our experimental analysis were not chosen with any geographical bias. This was due to the fact that *Par* was initially domesticated in southern China and is primarily grown there, leading to a notable reduction in its diversity (Dixon, 2006). It is a vegetable primarily valued for its edible components, encompassing the stems, stem leaves, and terminal inflorescences, which are commonly consumed in Asia. The *Par* variety displays two distinctive domestication traits not found in other leafy *B. rapa* crops. Firstly, it exhibits vernalization-independent flowering, in alignment with its suitability as a year-round vegetable. Secondly, building upon the first characteristic, *Par* was further domesticated to maintain stable flowering time under the warm conditions in the southern China environment. Thus its primary agronomic trait, flowering without vernalization, is an inherent feature independent of its geographical origin.

Fig 1F indicated the flowering time of all 44 *Par*, 41 *DG* and 50 *PC* accessions from our collection. We add the statement into Fig 1 F legend (line 122).

Reference:

Dixon, G. (2006). Vegetable Brassicas and Related Crucifers. *Vegetable Brassicas and Related Crucifers*, 1-327.

2. The authors first aimed to identify the genes responsible for inducing flowers in Par under normal conditions (NC). 1) However, from line 185, the authors suddenly switch the topic to flowering time at heat-stressed (HT) conditions, which disrupts the information flow. 2) And it is not clear whether the authors performed the QTL

analysis based on early or delayed flowering at HT. 3) After reading the whole manuscript, I can see the delayed flowering time in *Par* is the key phenotype. Then why don't starting the story directly from this phenotype? As I can see, the mapping of BrJM18 relies on the QTL analysis. Then why the domestication part is necessary for this story?

RE: We sincerely apologize for our insufficient description about *Par*. *Par* is a vegetable primarily valued for its enlarged stem, stem leaves and inflorescences as the main edible parts. This characterizes *Par* from other leafy *B. rapa* crops, where the main edible part is the leaves.

For the concern 1, we mentioned the characteristic phenotypic distinctions between *Par* and *PC* in the introduction. The *Par* variety, which originated in southern China, exhibits two unique domestication traits not observed in other leafy *B. rapa* crops: 1) *Par* flowerings rapidly without vernalization, which aligns with its fast-growing and year-round characteristics as a commercial vegetable. 2) Building upon the first characteristic, *Par* was further domesticated to maintain stable flowering time under HS conditions in the southern part of China. This stability is crucial for preserving the quality and yield of the commercial organ of *Par* since flowering at inappropriate times can impact them. Therefore the second domestication characteristic is closely related to the high temperatures. In other words, the domestication of *Par* primarily focuses on its adaptation to high temperatures, particularly in stabilizing flowering time under HS conditions in the southern China region. Before line 185, our domestication analysis concentrated on selective loci that distinguished *Par* from *DG* but not *DG* from *PC*. However, these domestication genes may be related to *Par*'s flowering, high-temperature adaptability, but also other traits, or they may be unrelated. To determine which domestication genes in *Par* are influenced by environmental temperature, we thus conducted further analysis on flowering time QTLs under different temperature conditions. We added the above statements in to the introduction (lines 53-63) to highlight the typical domestication characters of *Par*.

With regarding to the concern 2, we now present an explanation of how we carried out the QTL analysis below. We conducted QTL mapping for the flowering time loci on two occasions: one using the flowering time of the F2 population under HS conditions, and the other using the flowering time under low-temperature conditions. By comparing the mapping output, we were able to determine which loci are responsive to changes in temperature. Please see details below: An F2 population, comprising approximately 350 offspring, was generated through a cross between *Par* and *PC* for our analysis. The germinated seeds were planted in soil and cultivated under NC conditions at 22°C/22°C with a 16-hour light and 8-hour dark period for 4 weeks. Following this, half of the seedlings were kept under NC conditions until their flowering time data were recorded, while the other half were transferred to HS conditions at 29°C/29°C with a 16-hour light and 8-hour dark period until they flowered. Flowering-related quantitative trait loci (ftQTLs) responding to different temperatures were then identified by using

BSA analysis. Our focus then shifted towards QTLs that manifested solely under HS conditions and were absent under NC conditions. However, the initial data above could not determine whether these QTLs were correlated with early or delayed flowering when subjected to HS conditions. Nevertheless, subsequent experiments provided us with clearer insights, and we discovered that the genomic region housing the *BrJMJ18* gene is associated with delayed flowering under HS conditions. We revised the above statements into the sections of “Genotype selection, planting, and phenotyping of natural *B. rapa* collection” (lines 187-191) and “Bulked segregation analyses (BSA)” (lines 814-825) of the Materials and Methods.

To the concern 3, the core of the story revolves around how *Par* acquires a stable flowering trait under HS conditions during domestication in the southern region of China. As you mentioned, using QTL mapping to identify the genes responsible for this trait is technically feasible. We actually attempted this approach but encountered a relatively large mapping interval (we obtained a total of 13 QTLs, with a cumulative interval size of 25.81 Mb. Among these, the interval on A09 harboring *BrJMJ18* was 2.24 Mb (Table S6)), which is difficult to narrow down (possibly due to *Par*'s low genetic diversity (Table 1)). However, the key challenge lies in confirming whether *BrJMJ18* is a domestication gene, which is crucial for understanding the species' formation history and developing economic traits, particularly for breeding purposes. On the other hand, by integrating QTL, F_{ST} and haplotype analyses, we significantly reduced the mapping interval to 170kb by effectively utilizing numerous natural variations. This allows us to pinpoint the target gene faster and with higher precision.

As recommended by the reviewer, we recognized the need for further clarification on the rationale behind our domestication analysis in the manuscript. Additionally, the comprehensive domestication analysis could potentially lead to confusion among readers. Therefore, we now revised and emphasized the domestication process and characteristics of *Par* in the introduction (lines 53-63), while significantly reducing and reorganized the related content of the domestication analysis in the results (please see lines 152-163, and 166-169 in Word's tracking mode).

3. In line 238, the authors showed that *BrJMJ18* is differentially expressed during floral transition among three tested subspecies, but the author further showed that *BrJMJ18* doesn't have differential expression during flowering. Considering the mechanism of flowering time regulation, the expression during floral transition is, of course, more important. Then why do authors focus on the amino acid changes but not the expression level if that is the case?

RE: Sorry for the inappropriate description. We revised the text as: 1) “We then found that *BrJMJ18* (BraA09g034190.3C) was the most highly expressed *BrJMJ*, while the other two were undetectable; and *BrJMJ18* was differently expressed during floral transition in *DG* and *Par*, but not *PC*, under NC conditions (Table S7, s4) “(lines 232-234). 2) “Since no significant differences of the expression pattern of *BrJMJ18*

were found during flowering (Table S7, s4) and in the response to high temperature (Figure S9) between *DG* and *Par*" (lines 243-245).

And we fully agree with your statement that the expression during floral transition is important. The escalating expression trend of *BrJMJ18* during flowering in *DG* and *Par* also supports this point. However, concerning the fact that there is no significant difference in the expression pattern in *DG* and *Par*, we propose that the allelic functional disparity of *JMJ18^{PC}* (*BrJMJ18^{WT}* was replaced *BrJMJ18^{PC}* in the new text as suggested by reviewer 3) and *JMJ18^{Par}* does not lie in the expression pattern. Importantly, our enzymatic assays revealed that heat stress leads to a stronger reduction of H3K36me3/2 in *BrJMJ18^{PC}*-expressing than in *BrJMJ18^{Par}*-expressing tobacco leaves (Figure 4C). This strongly suggests that amino acid changes may be a crucial factor contributing to the functional differences between *JMJ18^{PC}* and *JMJ18^{Par}*.

4. *The authors always want to discuss the function of BrJMJ18 in flowering time regulation at both NC and HT conditions, making it difficult to follow the story. BrJMJ18Par at NC is not crucial if the authors mainly want to analyze BrJMJ18Par in HT. I think they should simplify the NC part and focus on the function of BrJMJ18Par at HT.*

RE: In the study, we demonstrated that *BrJMJ18^{PC}* functions as a "throttle" for flowering, regardless of whether it is under NC conditions or HS. On the other hand, *BrJMJ18^{Par}* exhibits functional divergence at different temperatures. Under NC conditions, *BrJMJ18^{Par}* promotes flowering, but under high temperature, it acts as a "brake" for flowering. In these descriptions and discussions, we can consider the function analysis of *BrJMJ18^{Par}* at NC as a control, which allows for a clear comparison of its functional divergence under different temperatures.

We really appreciate the reviewer's suggestion, and realize that we may have described too much about the functions of *BrJMJ18* at NC, which could have distracted readers' attention. We therefore have condensed the descriptions concerning 'NC'. Furthermore, we have moved certain descriptions from the initial section of the Results "*BrJMJ18Par* delays flowering under both greenhouse and field high-temperature conditions," to Figure S11 and have revised it. Please refer to section "*Results-BrJMJ18^{Par} delays flowering under both greenhouse and field high temperature conditions*" in Word's tracking mode.

5. *There are a lot of data generated in Arabidopsis. I doubt the necessity since they can create mutants or overexpression lines in Par. I think they should remove these data from this story.*

RE: We appreciate the reviewer's suggestion, the data generated in *Arabidopsis* are intended to illustrate the equivalent function of *JMJ18s* in both *Par* and *Arabidopsis*. However, we now realize that we have discussed too much of the data. We therefore deleted most *Arabidopsis* data in the main text and

moved most of the *Arabidopsis* data to supplementary Figures (Figure S12, Figure S17, S18, Figure S23). Please see details in lines 300-303, 426-428, 390-391 and 571-572.

6. The authors claim that *BrJMJ18^{Par}* delays flowering time at HT. If this is the case, why does losing *BrJMJ18* result in late flowering under HT? It is very confusing.

RE: The late flowering of *BrJMJ18^{Par}* under HS conditions is primarily due to the dissociation of *BrJMJ18^{Par}* from downstream flowering genes, such as *BrFLC3*, which results in the inability to regulate their expression, leading to delayed flowering in plants (Figure 5). Knocking out *BrJMJ18^{Par}* produces a similar effect on this process, resulting in a comparable flowering phenotype under HS conditions.

We added the statement above into the Result “*BrJMJ18* moderates flowering through *BrFLC3*” (lines 432-436).

7. According to the level of *H3K36me3*, the authors believe that *BrJMJ18^{Par}* dissociates from the *BrFLC3* locus. The author has already generated the antibody of *BrJMJ18*; then, they should confirm the dissociation of *BrJMJ18* at HT by ChIP-PCR?

RE: In Figure 5A, we confirmed the dissociation of *BrJMJ18* from *BrFLC3* at HS by ChIP-PCR. In the study, Chromatin immunoprecipitation (ChIP) analysis of the *BrJMJ18* level across *BrFLC3* were performed in *BrJMJ18^{Par}*-carrying *Par* and *BrJMJ18^{PC}*-carrying *DG* plants under both NC and HS conditions. The chromatin immunoprecipitation was conducted using the polyclonal antibody recognizing *BrJMJ18*.

Figure 5 A Both *BrJMJ18^{PC}* and *BrJMJ18^{Par}* bind to the chromatin of *BrFLC3*, and high temperature aggravates the binding of *BrJMJ18^{PC}*, but not *BrJMJ18^{Par}*, to *BrFLC3*. Chromatin immunoprecipitation (ChIP) analysis of the *BrJMJ18* level across *BrFLC3* were performed in *BrJMJ18^{Par}*-carrying *Par* and *BrJMJ18^{PC}*-carrying *DG* plants. Five-week old plants grown under NC conditions, and four-week old plants grown under NC conditions following 1-week of HS conditions, were used for the analysis. Rabbit IgG was used as control. GADPH was used as a *BrJMJ18*-independent control. Control is a locus gene desert regions where *BrJMJ18* does not bind. The values are the mean \pm standard deviation from three biological replicates. Asterisks indicate significant differences, Student's t-test (* $p < 0.05$, ** $p < 0.01$, *** $p < 0.001$).

8. I am also confused why the authors don't perform the ChIP-seq with the *BrJMJ18* antibody at both NC and

HT. I think such results are more informative than *BrJMJ18-GFP* ChIP-seq in overexpression lines.

RE: We very much agree with your point of view. In fact, our initial attempt encompassed ChIP-seq utilizing *BrJMJ18*'s native antibody in *DG* and *Par* plants. Unfortunately, the outcomes were unsatisfactory. We speculate that this could be attributed to moderate expression of the *BrJMJ18* (as indicated by RNA-seq data) and substantial genetic background differences of *DG* and *Par*, resulting in less-than-optimal results.

9. The figures do not always clear the number of biological replicates or the number of analyzed samples. Therefore, it is sometimes difficult to judge the conclusion.

RE: Thanks for reminding. For phenotyping of *Arabidopsis* and *Par* plants, 15 individual plants of each line were measured. For tobacco leaf transient transformation experiment, images of at least 30 epidermis cells expressing each allele of *BrJMJ18-GFP* protein under NC conditions or HS condition were acquired using the confocal laser scanning system Zeiss LSM510, respectively. *BrJMJ18-GFP* protein was localized in the nucleus without exception. Sample size was chosen based on previous experience and standards in the field. The phenotyping, transient transformation and Q-PCR experiments were repeated for three times. All attempts of replication were successful and gave similar results.

We have supplemented the figure captions with descriptions regarding biological replicates and sample numbers. Please see details in lines 269, 277, 359-360 and 399-400.

10. I expect a clear mechanism to explain how *BrJMJ18Par* delay flowering time at HT in *Par*; a model will be helpful.

RE: Thanks for the insightful comments. We added a working model into Fig S25 as suggested.

Supplementary Figure 25 A working model of BrMJ18^{Par} under different temperature conditions. Under NC conditions, the induction of BrMJ18^{PC} and BrMJ18^{Par} downregulates *BrFLC3* by demethylating its H3K36me3, consequently promoting flowering. Under high temperature conditions, the flowering promotion function of BrMJ18^{PC} is strengthened in most *B. rapa* subspecies. However, some *B. rapa* crops have developed the *BrMJ18^{Par}* allele to stabilize flowering and vegetative growth to counter against high temperatures via a mechanism in which the binding and subsequent demethylation activity of BrMJ18^{Par} of its downstream loci, including *BrFLC3*, is notably weakened by heat.

Reviewer #3 (Remarks to the Author):

This manuscript identifies BrJMJ18 as a targeted locus for the domestication of a thermotolerant B. rapa crop, Chinese kale (B. rapa subsp. chinensis var. parachinensis). The authors utilized various domesticated strains of B. rapa to identify loci that are adapted to tropical regions. In this manuscript, the authors focused on gQTLA09-1 and identified BrJMJ18 as a candidate gene that confers local adaptation of parachinensis (Par). The authors presented data to show that BrJMJ18 is an H3K36me2/3 demethylase and nonsynonymous variations in BrJMJ18 protein confer thermotolerant traits of Par. This study may provide interesting insights into knowledge on the domestication of crops in various local environments, including different climate regimes. However, several issues need to be addressed to warrant the conclusion. Major issues are listed below.

RE: We thank you for your insightful comments and suggestions. We have implemented all of your applicable recommendations to improve the quality of the presented work, please see details in our point-by-point response below. With these improvements, we hope that you will find the manuscript much improved.

Major points.

1. *Re: Fig. 4 BrJMJ18 as an H3K36me2/3 demethylase*

In this manuscript, the authors used two methods to argue that BrJMJ18 is an H3K36me2/3 demethylase. 1) The authors used transgenic Arabidopsis plants in which BrJMJ18 is under the control of the Arabidopsis JMJ18 promoter. Affinity purified protein extract of BrJMJ18 from Arabidopsis was incubated together with calf thymus histones to detect activity (Fig. 4A; Fig. S14). 2) transiently expressed two variants of BrJMJ18 in tobacco and extracted histone to detect the level of H3K36me2/3 (Fig. 4B).

1-a. A previous study of Arabidopsis JMJ18 showed that the Arabidopsis ortholog of JMJ18 is an H3K4m2/3 demethylase (Yang et al., 2012). Neither of the two methods the authors used is appropriate to characterize the specificity of demethylases. Purified recombinant proteins with careful assays should be necessary.

RE: Thanks for the insightful comments. As suggested, we utilized affinity-purified His-BrJMJ18^{PC/Par} proteins and synthesized Histone H3 peptides with specific modifications, including H3K4me3, H3K9me3, H3K27me3, and H3K36me2/3, for conducting the analysis. Importantly, the outcomes align with the results obtained using immunoaffinity-purified BrJMJ18:gfp proteins in the before mentioned experiment. Specifically, the His-BrJMJ18 protein exhibited effective demethylation of H3K36me2/3 peptides, while keeping the methylation levels of H3K4me3, H3K9me3, and H3K27me3 peptides unchanged. These findings have been integrated into Figure 4A and S14A, accompanied by a detailed description in the main text (lines 320-326), and a comprehensive protocol in the Method section (lines 927-929 and 941-947).

Figure 4A *E. coli* expressed His-BrJMJ18^{PC} and His-BrJMJ18^{Par} demethylate H3K36me3 and H3K36me2 *in vitro*. Synthesized Histone H3 peptide with H3K4me3, H3K9me3, H3K27me3, H3K36me2 and H3K36me3 modifications were incubated with affinity-purified recombinant His-BrJMJ18^{PC/Par} protein for 4 h at 37°C. Purified His was used as a negative control.

Figure S14A The two allelic His-BrJMJ18 proteins were affinity-purified from *Escherichia coli* cells. And synthesized Histone H3 peptides with H3K4me3, H3K9me3 and H3K27me3 modification were used as substrates, respectively. Purified His was used as a negative control.

1-b. It is my understanding that Fig. 4B used tobacco with transiently expressed BrJMJ18. Then, how can the plants be exposed to 3 weeks of NC and 3 days of HS? Are these stable transgenic lines?

RE: Sorry about the mistake in Figure 4B (new Figure 4C). The tobacco leaves were treated under NC condition for 3 days and NC for 2 days following 1 day of HS, respectively. We corrected the description in the new text.

1-c. I do not think the designation of polymorphic alleles of BrJMJ18 as *wt* vs. *par* is appropriate. The "Par" version of BrJMJ18 is not a mutant. Perhaps *DG* vs. *Par* (or *PC* vs. *Par*) should be considered.

RE: As you pointed out, the "Par" version of BrJMJ18 is not a mutant since both BrJMJ18^{WT} and BrJMJ18^{Par} are naturally present in ancient *Brassica rapa* (Fig S10). Therefore, we replaced "BrJMJ18^{WT}" with "BrJMJ18^{PC}" in the updated text and figures as suggested.

2. Re: Table S8: ChIP-seq of BrJMJ18.

2-a. It appears that the authors generated a polyclonal antibody against BrJMJ18. However, no data is presented to show the specificity of the antibody used in ChIP-seq experiments.

RE: We performed Western blot (WB) analyses employing recombinant His-BrJMJ18^{PC/Par} proteins that were affinity-purified from *Escherichia coli* cells to assess the BrJMJ18 polyclonal antibody's specificity. The BrJMJ18 antibody successfully detected identical His-BrJMJ18 bands as the His antibody, affirming the antibodies' specificity. We have incorporated these results into new Figure S19.

Figure S19. BrJMJ18 antibody specifically recognized BrJMJ18 protein in *Par*. Recombinant His-BrJMJ18^{PC/Par} proteins were affinity-purified from *Escherichia coli* cells. About 0.8-4.8 μ g His-BrJMJ18 proteins were separated by SDS-PAGE. Immunoblotting analysis were conducted using anti-His (HT501, TransGen, China) and anti-BrJMJ18 antibodies, respectively. The His antibody identified the His-BrJMJ18 band at about 150 KD and several non-specific bands. BrJMJ18 antibody identified exactly the same His-BrJMJ18 bands with His antibody.

2-b. It was not possible for me to evaluate the quality of this ChIP-seq experiment. There is no indication of the number of replicates, and it is also unclear what was used for the control.

RE: For ChIP-seq experiment, two biological replicates were performed for each sample. ChIP-seq experiment was carried out by IGENEBOOK Biotechnology Company (Wuhan, China) according to a previously described method (Landt et al., 2012). Briefly, 1.5 g samples were washed twice in cold phosphate-buffered saline (PBS), cross-linked with 1% formaldehyde for 10 minutes at room temperature, and then quenched by the addition of glycine (125 mmol/L final concentration). Afterwards, samples were lysed and chromatin was obtained on ice. Chromatin was sonicated to get soluble sheared chromatin (average DNA length of 200–500 bp). Then, 20 μ L of chromatin was reserved at -20 $^{\circ}$ C as input DNA. MACS2 software (version 2.1.1.20160309) was used to call peaks using default parameters (bandwidth,

300 bp; model fold, 5, 50; q value, 0.05) using Input as control. And BrJMJ18 target genes under NC/HS were obtained using wt *Par* plants under NC/HS as control. Please see the details in the Method section (lines 958-991).

2-c. *Fig. 5A used GADPH as a loading control for BrJMJ18 ChIP-qPCR validation at BrFLC3. Is the GADPH also enriched by BrJMJ18? From ChIP-seq data, the authors should have been able to extract loci that are enriched with BrJMJ18 regardless of conditions, and those loci are better controls.*

RE: GADPH is not enriched by BrJMJ18. In ChIP-Seq assays, it is reported that the two most commonly used methods for normalizing ChIP-qPCR data are fold enrichment and percent input. Fold enrichment calculates a signal-to-noise ratio by comparing the target sequence's amount in the ChIP sample to that in a negative control sample. This calculation is beneficial for estimating the signal-to-noise ratio. To calculate fold enrichment, the enrichment fold was consistently determined using IgG as a negative control, and a typically non-enriched housekeeping gene that serves as a suitable candidate (Haring et al., 2007). We used GADPH as internal control and normalized fold enrichment of BrJMJ18 and H3K36me3 to GADPH previously. Since GADPH did not demonstrate enrichment by BrJMJ18, we utilized GADPH as a loading control to illustrate the antibody's specificity (Huang et al. 2020). We have reanalyzed the ChIP-QPCR results following the approach described in the study by Sharma et al. (2016). We have made revisions to Figure 5A, E, F, Figure S15A, and the corresponding figure legends (lines 394-400).

References:

- Haring, M., Offermann, S., Danker, T., Horst, I., Peterhansel, C., & Stam, M. (2007). Chromatin immunoprecipitation: optimization, quantitative analysis and data normalization. *Plant methods*, 3, 1-16.
- Huang, Y., Zhang, H., Wang, L., Tang, C., Qin, X., Wu, X., Pan, M., Tang, Y., Yang, Z., Babarinde, I.A., et al. (2020). JMJD3 acts in tandem with KLF4 to facilitate reprogramming to pluripotency. *Nature communications* 11, 5061.
- Sharma, V., Pandey, S.N., Khawaja, H., Brown, K.J., Hathout, Y., and Chen, Y.W. (2016). PARP1 Differentially Interacts with Promoter region of DUX4 Gene in FSHD Myoblasts. *Journal of genetic syndromes & gene therapy* 7.

3. Re: Flowering time of various genotypes and transgenic plants

3-a. *Fig. S11-B: It is unclear why the authors compared the flowering times of vernalized plants with high temperature-grown plants, and it cannot be interpreted as "Line 267 - in which HS conditions caused early flowering in BrJMJ18PC-carrying plants, but delayed flowering in BrJMJ18Par-carrying plants". The authors are comparing two different treatments.*

RE: Sorry about the mistake in Fig S11B. All the plants were grown under NC conditions or HS condition without vernalization. We corrected the figure and legend.

Figure S11. Long-term high temperature exerted a stronger effect on *BrJMJ18^{PC}*- than on *BrJMJ18^{Par}*-carrying lines.

(A) The flowering times of *BrJMJ18^{PC}*- and *BrJMJ18^{Par}*-carrying DG lines (n = 39) farmed under natural field conditions in Zhangjiakou (ZJK) and Beijing (BJ), respectively, were used for analysis. The daily temperature of BJ is an average of 5 °C higher than that of ZJK. The earlier flowering induced by high temperature was attenuated significantly in *BrJMJ18^{Par}*-carrying plants. Flowering time, days after germination (DAS), was defined as the number of days from sowing to the appearance of the visible buds.

(B) Flowering time of *BrJMJ18^{PC}*-, *BrJMJ18^{Par}*- and *BrJMJ18^{PC/Par}*-carrying lines of the F2 population, which was generated from the F1 crosses of *PC* and *Par*. The flowering time was evaluated under normal conditions (NC) and heat stress (HS) conditions, respectively. NC seedlings were grown at 22°C under a long-day regime (16/8 h day/night) for 4 weeks, and then transplanted in pots under NC until bolting. HS, 4-week seedlings grown under NC were transplanted and moved to HS conditions until bolting. Flowering time, days after sowing (DAS), was defined as the number of days from sowing to the appearance of the visible bud. HS conditions caused early flowering in *BrJMJ18^{PC}*-carrying plants, but delayed flowering in *BrJMJ18^{Par}*-carrying plants (as shown by red arrows).

3-b. Flowering time as DAS (the day after sowing). Do HS conditions the authors used (either greenhouse or field conditions) cause any differences in growth (or developmental) -rate? HS condition the authors described (Fig. S4) shows apparent detrimental effects on plant growth.

RE: From Figures 3 and 6, we can see the differences in yield at the final harvest between *BrJMJ18^{PC}*-OX and *BrJMJ18^{Par}*-OX under HS conditions. To further elucidate the impact of HS conditions on the growth rate of different plants, we cultivated *Par*, *BrJMJ18*-OX, and *BrJMJ18^{Par}*-CR plants under NC conditions. After four weeks, we subjected them to both NC and HS treatments. We recorded the rosette leaves numbers of the *Par*, *BrJMJ18*-OX and *BrJMJ18^{Par}*-CR plants grown under NC conditions and HS conditions weekly since the high temperature treatment started. Under NC conditions, *BrJMJ18*-OX plants had fewer leaves than *Par* plants, while *BrJMJ18^{Par}*-CR plants had a leaf count similar to *Par* plants. Under (HS) treatment, only *BrJMJ18^{Par}*-OX plants showed a noteworthy increase in true leaf number, whereas the

leaf counts for the other three plant types exhibited no significant difference or slight decrease compared to NC conditions. We put these data in new Figure S20.

Figure S20. High temperature exerts different impact of the growth of *Par*, *BrJMJ18-OX*, and *Par^{BrJMJ18CR}* plants.

4-week-old *Par*, *BrJMJ18-OX*, and *BrJMJ18^{Par}-CR* plants grown under NC condition were moved to NC and HS treatments. True leaves numbers were recorded weekly until flowering. Under NC conditions, the two allelic of *BrJMJ18-OX* plants had fewer true leaves than *Par* plants, while *Par^{BrJMJ18CR}* plants had a leaf count similar to *Par* plants. Under HS treatment, only *BrJMJ18^{Par}-OX* plants showed a noteworthy increase in true leaf number, whereas the leaf counts for the other three plant types exhibited no significant difference compared to NC condition. The box encompasses two middle quartiles, with central line showing median. Whiskers extend to the furthest data point within 1.5 times the interquartile range. , n = 15. *Par* represents wild type *Par* plants, PC-OX represents *BrJMJ18^{PC}-OX* plants, Par-OX represents *BrJMJ18^{Par}-OX* plants, CR represents *BrJMJ18^{Par}-CR* plants.

With regarding to “HS condition the authors described (Figure S4A) shows apparent detrimental effects on plant growth.”, as observed by the reviewer, the HS condition indeed showed evident detrimental effects on plant growth, but except for the *Par*. We supposed that this can be attributed to the early treatment of young seedlings at a continuous high temperature of 42 °C, which is actually resembling a “heat shock”. We have made corrections in the new version to clarify this.

Figure S4(A) For heat shock treatment, 14-day seedlings grown under normal conditions (NC) were moved to NC and heat-shock conditions (16/8 h day/night, 42°C/42° C), respectively, for another one week. After the heat treatment, the plants were recovered at 22°C for 5 days. At the end of recovery, photographs were taken.

4. Fig. 7

4-a. Similar to point #2, I could not evaluate the quality of this ChIP-seq experiment.

RE: We added descriptions regarding biological replicates and the number of samples in the the Method section (lines 958-991)

4-b. Line 498: “32.8% of the enriched genes in BrJMJ18PC-OX plants and 28.6% in BrJMJ18Par-OX plants under NC conditions overlapped with reported H3K36me3-targeted genes, while the corresponding frequencies under HS conditions decreased to 22.1% and 20.0%: 1)Have the authors evaluated the association with other modifications (i.e., H3K4me2/3, H3K27me2/3, etc.). 2) It is also quite surprising that more than 70% of BrJMJ18PC and BrJMJ18-Par are unique. Are the authors suggesting two variations of BrJMJ18 have evolved to target quite different sets of genes?

RE: Thank you for your reminding and it's highly valuable for assessing BrJMJ18's H3K36me2/3 demethylation activity. 1) Currently, we are examining the connection between the enriched genes in BrJMJ18-OX plants and H3K27me2/3 targeted genes. It is reported that H3K27me3 has an opposing relationship with H3K36me3 accumulation. According to Mehral et al. (2021), there are at least 10,445 B. rapa genes marked by H3K27me3. Under NC conditions, only 15.5% of enriched genes in BrJMJ18^{PC}-OX plants and 15.6% in BrJMJ18^{Par}-OX plants overlap with the reported H3K27me3-marked genes. Furthermore, these proportions decrease to 13.0% and 10.5%, respectively, under HS conditions. These data indicate that the genes precipitated by BrJMJ18 are indeed specifically targeted by H3K36me2/3. We

Table S9 Genes
enriched in anti-G

revised the description above and put it into Table S9 and description in lines 626-631.

2) Based on our data, the results indicate that BrJMJ18^{PC} and BrJMJ18^{Par} are evolved to target different sets of genes. ~70% of the enrichment genes under NC conditions were specific to BrJMJ18^{PC}-OX and BrJMJ18^{Par}-OX plants, respectively (Figure 7 A). However, interestingly, the Gene Ontology (GO) analysis revealed that under NC conditions, the significantly enriched items in both *BrJMJ18^{PC}-OX* and *BrJMJ18^{Par}-OX* plants exhibited remarkable consistency (Supplementary Figure S21 A, B). This indicates that even though BrJMJ18^{PC} and BrJMJ18^{Par} have evolved to target different genes, their regulatory physiological processes are fundamentally similar. However, under HS conditions the enriched GO items in these two plants exhibited significant divergence, suggesting their distinct responses to temperature stresses.

Reference:

Mehraj, H., Takahashi, S., Miyaji, N., Akter, A., Suzuki, Y., Seki, M., Dennis, E.S., and Fujimoto, R. (2021). Characterization of Histone H3 Lysine 4 and 36 Tri-methylation in *Brassica rapa* L. *Frontiers in plant science* 12.

4-c. Line 515: "18.2% of the DEGs in *BrJMJ18WT-OX* plants and 17.8% of the DEGs in *BrJMJ18Par-OX* plants are BrJMJ18-target genes identified by ChIP-seq under NC conditions, while the corresponding frequencies under HS conditions decreased to 5.0% and 5.7%." It is not clear whether these overlaps are significant. How many genes that are identified as BrJMJ18 targeted are also DEGs?

RE: Before addressing this concern, we conducted a comparison to determine whether the genes regulated by BrJMJ18 significantly intersects with the heat stress response genes in *B. rapa*, in order to confirm the involvement of BrJMJ18 in plant heat stress responses. In a study of Yue et al. (2021) they identified 5,559 DEGs in Chinese cabbage after 8 days of heat stress. We discovered that 35.07% and 29.32% of the *BrJMJ18^{PC}-OX* and *BrJMJ18^{Par}-OX* DEGs under NC conditions overlapped with the DEGs reported by Yue et al. (Table S10). These findings indicate that the genes regulated by BrJMJ18 are mostly involved in heat stress response.

On top of this, we further observed that 18.2%, including 455 genes, of the DEGs in *BrJMJ18^{PC}-OX* plants and 17.8% (386 genes) of the DEGs in *BrJMJ18^{Par}-OX* plants were identified as BrJMJ18-binding genes through ChIP-seq analysis under NC conditions. However, these percentages decreased to 5.0% (29 genes) and 5.7% (38 genes) under HS conditions (Table S10). Beyond of the above, we conducted a further randomness test using SPSS to evaluate the abovementioned frequencies are significant. In this test, we employed a simple random sample approach to select 100 genes from the genome, and subsequently

analyzed their overlap with BrJMJ18-binding genes at a significance level of $\alpha=0.05$. The resulting proportion of the overlap was determined to be 1.3%. This value is significantly lower than the observed overlap frequencies between the DEGs of *BrJMJ18-OX* plants and BrJMJ18-binding genes. We added the

Table S10

Differentially expr

statement in the result (lines 494-496 and 499-502).

Reference:

Yue, L., Li, G., Dai, Y., Sun, X., Li, F., Zhang, S., Zhang, H., Sun, R., and Zhang, S. (2021). Gene co-expression network analysis of the heat-responsive core transcriptome identifies hub genes in *Brassica rapa*. *Planta* 253, 111.

5. Fig. 8

5-a. Using a heterologous system to show biochemical specificity is not desirable, and it is hard to interpret the result. The assumption here is that *in vivo* system of *Arabidopsis* and *B. rapa* is similar to each other, but the authors concluded that BrJMJ18 is different from its *Arabidopsis* counterpart. Are they affecting *AtFLC* by acting as H3K36me2/3 demethylases or as H3K4me2/3 demethylases?

RE: Thanks for the insightful comments. In Figure S18, we investigated the H3K36me3 level of *AtFLC* loci in Col-0 and *AtBrJMJ18::BrJMJ18^{PC/Par}::gfp* plants under NC conditions and HS, respectively. BrJMJ18^{PC/Par} affected the H3K36me3 level of *AtFLC* in *Arabidopsis* in a similar pattern as *BrFLC3* in *Par*.

Figure S18. Chromatin immunoprecipitation (ChIP) analysis of H3K36me3 enrichment on the *AtFLC* locus in *BrJMJ18* overexpression *Arabidopsis* lines under normal conditions (NC) and heat stress (HS) conditions,

respectively. The H3K36me3 level at *AtFLC* was downregulated in *AtJMJ18^{pro}::BrJMJ18^{PC}:gfp* but upregulated in *AtJMJ18^{pro}::BrJMJ18^{Par}:gfp* plants by heat. Three-week old plants grown under NC, and two-week old plants grown under NC following 1-week of HS were used for the analysis. *Actin2* was used as internal control of H3K36me3 as reported by Yang, et al, 2012 previously. The values are the mean \pm standard deviation from three biological replicates.

Reference:

Yang, H., Howard, M., and Dean, C. (2014). Antagonistic roles for H3K36me3 and H3K27me3 in the cold-induced epigenetic switch at Arabidopsis FLC. *Current biology* : CB 24, 1793-1797.

5-b. *BrJMJ18^{Par}-OX dissociates more from FLC3 in HS conditions, but the catalytic activity is better. Are these variations contributing to BrJMJ18 bindings?*

RE: Under HS conditions, Figure 5A, D and E demonstrate a significant increase in the dissociation of BrJMJ18^{Par} from *BrFLC3*. Meanwhile, in Figure 4 C, BrJMJ18^{Par}'s catalytic activity is weaker than that of BrJMJ18^{PC}. Based on these results, as proposed by the reviewer, it can be inferred that the functional differentiation between BrJMJ18^{Par} and BrJMJ18^{PC} is primarily due to BrJMJ18^{Par}'s alteration in substrate binding activity.

6. *Figure legends should be carefully revised. For example, Fig. 2 legend: There are no horizontal blue lines – they are pink. There is no pink bar (perhaps peach bar?). There is no indication of what blue and orange Fst values represent.*

RE: We apologize for the mistakes. We corrected the figure legends (line 198) as suggested.

7. *Line 312: "We also analyzed the flowering time of BrJMJ18 overexpression lines (AtJMJ18^{pro}::BrJMJ18CDS:GFP) in Arabidopsis". These are not "overexpression lines". The authors used the Arabidopsis JMJ18 endogenous promoter.*

RE: The sentence was corrected as " We also analyzed the flowering time of *BrJMJ18* transgenic lines (*AtJMJ18^{pro}::BrJMJ18CDS:gfp*) in *Arabidopsis*" (line 300-301). Additionally, we have corrected the relevant descriptions in the text accordingly (lines 327 and 345).

8. *One of the conclusions by the authors is that BrFLC3 is adapted to mediate HS-specific delay in flowering in Par variations. BrFLC3 mutation in Par varieties can support it.*

RE: Thank you for your valuable feedback. Figure S27 illustrates that there are no specific haplotypes of *BrFLC3* associated with a particular *B. rapa* crop. This observation could provide an explanation for the limited genetic mapping of *BrFLC3* for flowering time (table S11). Additionally, it suggests that the

flexibility in *BrFLC3* expression might serve as an adaptive factor contributing to thermotolerance.

Reviewer #4 (Remarks to the Author):

This is a very interesting paper, in which the authors identify an H3K36me3 demethylase in Brassica rapa. Natural variation in this protein contributes to the temperature response of flowering time, specifically in relation to elevated temperatures. Through a considered, step-by-step approach, the authors show first that there is a difference in flowering time specific to the Par genotype in warm conditions. They then identify the gene and show it has H3K36me3 demethylase activity. They record flowering time, gene expression, histone modifications and protein binding to show that the action of this gene strongly affects flowering response in high temperatures, likely through the demethylation of H3K36me3 at the BrFLC3 gene.

RE: We thank you for your insightful comments and suggestions. We have implemented all of your applicable recommendations to improve the quality of the presented work, please see details in our point-by-point response below. With these improvements, we hope that you will find the manuscript much improved.

Major comments

1. *Can you clarify a bit more in the introduction what the desired trait is in Par? It seems that flowers are wanted in warm conditions, without vernalization, but at the same time a late flowering natural allele is favourable. Is longer vegetative growth desirable?*

RE: We apologize for not clarifying this matter clearly. *Par* possesses two distinct domestication traits, both of which are essential for its production: 1) It can flowering without vernalization, ensuring a continuous supply of edible stem and inflorescence organs throughout the year. 2) Its flowering time remains relatively consistent despite fluctuations in temperature, guaranteeing a steady availability of high-quality edible parts. However, this does not necessarily imply that longer vegetative growth is desirable.

Regarding the second trait, *Par* can be classified into early-flowering types and late-flowering variety types. Early-flowering types are suitable for sowing during the relatively warmer period from late spring to early autumn, while late-flowering types are suitable for sowing in early spring, late autumn, and winter. However, irrespective of the type, a stable flowering period is essential to ensure proper nutritional growth and obtain edible inflorescence. For instance, the "forty-nine" variety, a typical early flowering type of *Par*, can be sown from April to September each year. Despite variations in the frequency and intensity of high-temperature days in different months of the cultivation season, the "forty-nine" variety consistently flowerings and becomes ready for harvest in approximately 36 days, with no significant differences in flowering time and other commercial traits. This demonstrates that this type of *Par* has developed the ability to maintain a relatively stable flowering time, even under frequent temperature fluctuations,

ensuring its safety in production and supply.

We revised the above statements in to the introduction (lines 53-63) to highlight the typical domestication characters of *Par*.

2. Related to that and to point 3 below, it is not clear if the focus is thermotolerance/heat stress or flowering time.

RE: In the introduction, we introduced the significant phenotypic distinctions between *Par* and *PC*. Originating from southern China, the *Par* variety exhibits two unique domestication traits that are not present in other leafy *Brassica rapa* crops, as mentioned in our earlier response to the reviewer's concern. The domestication path of *Par* suggests that particular attention has been given to ensure a stable flowering time under HS conditions. In other words, the primary focus of *Par*'s domestication lies in its adaptation of flowering time to cope with high temperature conditions of the South China. We revised the statement and added it in the introduction (lines 53-63 and 75-88).

3. I am not sure if the term "thermotolerance" is appropriate for the HS conditions used, at 29°C. **1)** Heat stress studies as far as I am aware are usually done in much higher temperatures (above 30°C, and normally around 40°C, as in Fig. S4 in this manuscript). **2)** The terms "thermotolerance" and "heat stress" are not absolute and they depend on the optimum range of the plant in question as well as the temperature range in the region where it is grown. I would suggest some more explanation to be added to justify the choice of these temperatures and possibly changing the terms "thermotolerance" and "heat stress" if appropriate. **3)** The result of *Par* being more thermotolerant as shown in Fig. S4 is convincing, however the link between this increased tolerance and *BrJMJ18* is not made

RE: Thanks for the insightful comments.

1) Most of the *Brassica rapa* crops originated from the Mediterranean coast, belonging to temperate climate zones, and they are highly sensitive to heat stress. According to "The Plant Family Brassicaceae" book, temperatures with a day-night average higher than 30°C can be considered as heat stress for Brassicaceae plants (Anjum et al., 2012). While as reported by Morrison (2002), During the flowering stage, the critical threshold temperature causing seed yield losses for all Brassica species was found to be 29.5°C. Mean maximum temperatures more than 29°C during vegetative development led to a decline in flower numbers across all Brassica species. As the intensity of heat stress during flowering increased, seed yield showed a corresponding decrease.

References:

Anjum, N., Ahmad, I., Me, P., Duarte, A., Umar, S., and Khan, N. (2012). The Plant Family Brassicaceae: Contribution Towards Phytoremediation.
Morrison, M.J., and Stewart, D.W. (2002). Heat Stress during Flowering in Summer Brassica. *Crop Science* 42, 797-803.

2) As you mentioned, the definitions of "thermotolerance" and "heat stress" are influenced by the plant's optimal range and the temperature conditions in its cultivation region. To address this, we referred to data from the China Meteorological Administration regarding average temperatures in Guangdong province, the area of *Par* domestication, for the summer months from July to September. Over the last decade, which is also able to reflect historical trends. The average summer temperature has stood at 29.4°C. Consequently, we propose that around 29°C can be regarded as the critical temperature for *Par* domestication. In this study, our heat treatment involved maintaining a consistent day-night temperature of 29°C, which can be deemed as inducing heat stress for *Par*. And we apologize for any confusion regarding the description of Figure S4. In Figure S4, referring to it as "heat shock" might be more accurate, and we have accordingly revised it in the manuscript. We revised the statement and added it in the introduction (lines 75-84).

Figure S4(A) For heat shock treatment, 14-day seedlings grown under normal conditions (NC) were moved to NC and heat-shock conditions (16/8 h day/night, 42°C/42°C), respectively, for another one week. After the heat treatment, the plants were recovered at 22°C for 5 days. At the end of recovery, photographs were taken.

3) In the tunnel greenhouse experiment, we evaluated the aboveground biomass of both *BrJMJ18^{Par-CR}* and *Par* plants under field conditions (Fig. 6B, D-G). The tunnel greenhouse experienced day maximum and minimum temperatures of 52°C and 31°C, respectively, with night maximum and minimum temperatures of 33°C and 24°C, respectively. The plants were harvested simultaneously, and the yield of *Par* plants was 1.3 times higher than that of *BrJMJ18^{Par-CR}* plants. This indicates that under natural extreme high temperature conditions, *BrJMJ18^{Par}* is strongly involved in thermotolerance.

4. It needs to be clarified which of the different versions of HS was actually used in the different experiments (e.g. 29/29°C or 29/22°C). Not consistent night temperature and pre-treatment between Methods section, main text, Fig. 2D and table S6. Vernalization treatment is likely to be significant and should be clarified where this was used.

RE: Apologies for any confusion. We have thoroughly reviewed the HS condition and verified that it was set at 29°C for both day and night, with a long day period of 16 hours of light and 8 hours of darkness. These specific details have been included in the “Plant materials and growth condition” and “Planting of transgenic *B. rapa* and *Arabidopsis* plants” of the “Materials and Methods” section (lines 722-723 and 726-734). Therefore, in both Figure 2D and Table S6, the utilized HS conditions is 29/29°C.

Additionally, we have now indicated all the experiments related to vernalization treatment in the figure legend.

5. Some results are hidden in tables (e.g. different expression of *BrJMJ*) while there are supplementary figures that are not necessarily adding anything more than the numbers in the text could give (e.g. S6 and perhaps also S10).

RE: In Figure S6, we want to convey two points: 1) The candidate selective genes, which exhibited differentiation between *Par* and *DG* but not between *DG* and *PC*, were chosen for further analysis; 2) the total number of these candidate genes is 964.

Regarding Figure S10, our objective was to illustrate that *BrJMJ18^{PC}* (*BrJMJ18^{WT}* was replaced *BrJMJ18^{PC}* in the new text as suggested by reviewer 3) and *BrJMJ18^{Par}* were evenly represented in the ancient *subsp. rapifera* and *subsp. oleifera* groups, respectively. This supports the idea that variations at *BrJMJ18^{Par}* might not have played a significant role in the early speciation of *B. rapa*, however, these variations could have conferred advantages for subsequent speciation and/or local adaptation.

As you mentioned, the results of Figure S6 are indeed presented in Tables S5; however, the reason we kept these two images is primarily because they appear to be readable visually.

6. If *BrJMJ18^{Par}* is deactivated by warm, 1) then why do the CRISPR, OX and wt *Par* not all have the same phenotype? Or if it is not fully deactivated, why is wt *Par* not intermediate between these? 2) In fact, why would overexpression of a K36 demethylase, even an inactive one, lead to an increase in *FLC* expression and *H3K36me3* levels? Could you add some more discussion around these points and any possible explanations? 3) What other genes are affected? (Table s9?)

RE: I apologize for not describing this matter clearly. The results from enzyme activity assays (Figure 4A-C), RNA-seq, and ChIP-seq (Figure 7) data indicate that *BrJMJ18^{Par}* remains active under HS conditions, but there may have been changes in its binding capacity and targets. Gene Ontology (GO) analysis revealed that under NC, *BrJMJ18^{PC}* and *BrJMJ18^{Par}* exhibited similar enrichment patterns for target genes (Supplementary Figure S21 A, B). However, under HS, the enriched target genes differed between *BrJMJ18^{PC}* and *BrJMJ18^{Par}* (Supplementary Figure S21 C, D). Moreover, GO analysis of DEGs shows that under HS conditions, *BrJMJ18^{Par}* demonstrated opposite regulatory effects compared to *BrJMJ18^{PC}*

concerning cell division-related genes (indicated by the shared GO terms of upregulated genes under BrJMJ18^{Par} and downregulated genes under BrJMJ18^{PC}). Furthermore, BrJMJ18^{Par} specifically regulates the expression of heat tolerance genes involved in photosynthesis and chlorophyll biosynthesis. In summary, these findings suggest that BrJMJ18^{PC} functions as a "throttle" for flowering, regardless of whether it is under NC conditions or HS. On the other hand, BrJMJ18^{Par} exhibits functional divergence at different temperatures. Under NC conditions, BrJMJ18^{Par} accelerates flowering, but under HS conditions, it functions as a "brake" on flowering and a "throttle" on growth. This explains why both *BrJMJ18^{Par}-OX* and *BrJMJ18^{PC}-OX* exhibit early flowering under NC conditions, but under HS conditions, only *BrJMJ18^{PC}-OX* shows even earlier flowering, while *BrJMJ18^{Par}-OX* displays delayed flowering and robust growth.

2) In Figure 5 D and E, we demonstrated that under HS conditions, the expression of BrJMJ18^{Par} leads to an increase in H3K36me3 levels and *BrFLC3* expression of the plant. This is mainly due to the fact that BrJMJ18^{Par} loses its ability to bind to *BrFLC3* as shown in Figure 5A. As a result, it cannot demethylate *BrFLC3* chromatin, leading to an elevation in H3K36me3 levels and an upregulation of *BrFLC3* expression.

3) To demonstrate that this changing pattern in BrFLC3 is not specific, we conducted expression and ChIP-qPCR validation on the randomly selected "BraA06g002250.3C" gene from Table S9. The results showed that the H3K36 chromatin status and BrJMJ18 enrichment changes of the "BraA06g002250.3C" gene were similar to those of BrFLC3. This further confirms that BrJMJ18^{Par}'s loss of binding capacity to certain genes under HS conditions, resulting in an inability to regulate their expression, is a widespread mechanism.

Figure S15. The enrichments of BrJMJ18^{PC} and BrJMJ18^{Par} at BrFLC1, 2 and 5 under different temperatures.

(A) Together with Figure 5A, we showed that in *DG*, BrJMJ18^{PC} binds strongly to *BrFLC1-3*, and high temperature aggravates their binding markedly. While in *Par*, BrJMJ18^{Par} binds strongly to *BrFLC3*, and slightly to *BrFLC1*; and intriguingly, we noticed that high temperature thoroughly disassociated the binding of BrJMJ18^{Par} and *BrFLC3* (Figure 6A). (B) To investigate the binding of BrJMJ18 to targets other than *BrFLCs*, we conducted ChIP-qPCR using anti-BrJMJ18 antibody on the randomly selected BrJMJ18-binding gene BraA06g002250.3C from Table S9. Both allele of BrJMJ18 protein could bind to BraA06g002250.3C under NC. High temperature aggravates BrJMJ18^{PC}'s binding to BraA06g002250.3C while disassociated the binding of BrJMJ18^{Par}, sharing the same changing pattern of binding to *BrFLC3*. (C) The H3K36me3 level of BraA06g002250.3C was further tested by ChIP-qPCR using anti-H3K36me3 antibody. Compared to NC, H3K36me3 levels of BraA06g002250.3C loci decreased in *DG* while increased in *Par* under HS. Chromatin immunoprecipitation (ChIP) analysis of the BrJMJ18 and H3K36me3 level across *BrFLCs* were performed in *BrJMJ18^{Par}*-carrying *Par* and *BrJMJ18^{PC}*-carrying *DG* plants. Five-week old plants grown under normal conditions (NC) and four-week old plants grown under NC following 1-week of heat stress (HS) were used for the analysis. The fold enrichment of BrJMJ18 and H3K36me3 level was calculated using IgG as control. GADPH was used as a BrJMJ18-independent control. Control is a locus gene desert regions where BrJMJ18 does not bind. The values are the mean \pm standard deviation from three biological replicates.

7. Claims of tight association between BrJMJ18 and BrFLC3 based on diurnal expression must be weakened, since it not excluded (and indeed seems likely) that they could both be under circadian regulation by the clock.

Expression of clock genes could be shown to distinguish between claim and heat-shock misregulation of circadian clock. Furthermore, strong diurnal regulation of these genes suggests claims about expression in different subspecies should be tested with measurements at different times of the day, since expression is so sensitive to this, or at least report time of sampling for all work and use the same for comparisons.

RE: Thank you for your suggestion. We have removed Figure 5C and its related descriptions. And we actually have taken “circadian” into consideration during our experiments. All qPCR and RNA-seq samples were collected at the end of day. We have added this information to the Materials and Methods section (line 793 and 959).

Minor comments

Table 1: Not consistent use between π and PI.

RE: All of the “ π ” were replaced with “PI” in the text

Fig. S1A: missing X-axis label. Also, it is unclear if the finally selected K is 5, 6 or 7.

RE: We re-drew Figure S1A (Figure S1B in the new text) and added the X-axis and labels. We selected K=5 for the following analysis since K = 5 explained the population structure best and maximized the marginal likelihood. Most of the morphologically distinct crops long recognized as subspecies were largely resolved as distinct clusters in our STRUCTURE analyses, which was consistent with the empirical classification. We added these details in the figure legend.

Fig 2 A, B: No horizontal blue line. Also “ π values of the corresponding comparisons are shown at the bottom of (A) and (B), respectively” I could not find these. Also, no x-axis labels.

RE: 1) the line is pink, and we corrected it in the figure legend (line 198); 2) sorry for the mistake, and we deleted the description; 3) the x-axis labels was added in Fig 2 A, B.

Lines 188-190: I could not find the data related to the BSA-seq approach including the flowering time phenotypes in different conditions. If authors feel it is beyond the scope or unnecessary to report this fully, then perhaps some summary information could be added in text or diagram form.

RE: Thanks for the insightful comments. For BSA analysis under NC conditions, we obtained 31.2 and 25.4 Gb of high-quality bases from the DNA bulks of the early bolting (n=25) and late bolting (n=25) progenies after data filtration, respectively. The average depth was about 60× and 52× in the early and late bolting bulks, respectively. A total of 2,904,351 and 2,790,600 SNPs were identified in the early and late

bolting bulks, respectively, and the SNP index was calculated for each SNP. By examining the $\Delta(\text{SNP index})$ plot, we found highly contrasting patterns in the SNP-index graph for the early bolting and late bolting bulks in 9 regions (bQTL, BSA-QTL; Table S6). While for BSA analysis under HS conditions, a total of 2,304,121 (n=25, 24.5 Gb, 50 \times) and 2,369,714 SNPs (n=25, 25.5 Gb, 50 \times) were identified in the early and late bolting bulks, respectively. By examining the $\Delta(\text{SNP index})$ plot, a total of 14 heat-responsive flowering time QTLs were determined. Our focus then shifted towards QTLs that manifested solely under HS conditions and were absent under NC conditions. We add BSA-seq method in the Method section (Lines 814-825). And flowering time of the individuals used for BSA seq were add to Table S6.

Table S6
Flowering time QTL

Fig. 2D: 99% or 95% threshold?

RE: 95%.

Fig. S7: No "24h after treatment".

RE: "24h after treatment" was deleted in the figure legend.

Table S7 s4, line 238 and line 248: In table S7 s4, BrJMJ18 is not differently expressed in PC during flowering, unlike the statement in line 238. Also is there no significant difference among subspecies during flowering (as stated in line 248)? This is surprising considering the large difference between PC and the other two.

RE: We are sorry for the inaccurate description. By analyzing the RNA-seq data of *PC*, *DG*, and *Par*, we found that the expression of *BrJMJ18* during flowering remains unchanged in *PC* but shows a similar increase in both *DG* and *Par*. Correspondingly, we revised the statements as "We then found that *BrJMJ18* (*BraA09g034190.3C*) was the most highly expressed *BrJMJ*, while the other two were undetectable; and *BrJMJ18* was differently expressed during flowering in *DG* and *Par*, but not *PC*, under NC conditions (**Table S7, s4**)." (lines 232-234), and original line 248 to " Since no significant differences of the expression pattern of *BrJMJ18* were found during flowering (**Table S7, s4**) and in the response to high temperature (**Figure S9**) between *DG* and *Par*" (lines 243-245).

Figure S9. Expression of *BrJMJ18* under different temperatures.

BrJMJ18 was induced by heat, but to a similar degree, in *DG* and *Par*. 3 *BrJMJ18*^{PC}-carrying *PC* (PC016, PC126, PC249) accessions, 2 *BrJMJ18*^{PC}-carrying *DG* (DG003, DG123) accessions, 1 *BrJMJ18*^{Par}-carrying *DG* (DG016) accessions and 3 *BrJMJ18*^{Par}-carrying *Par* (Par268, Par270, Par278) accessions were randomly selected from our germplasm collection. 5-week-old plants grown under normal conditions (NC), and 4-week old plants grown under NC following heat stress (HS) for one week were used for *BrJMJ18* Q-PCR. *GADPH* was used as internal control. The values are the mean ± standard deviation from three biological replicates.

Fig. S11: What is different in S11A compared to 1F that means the DG accessions are now flowering?

RE: Figure S11A provides further details on the flowering time of *DG* in Figure 1F (the right panel). We observed that among the *DG* population, there were 30 lines carrying the *BrJMJ18*^{PC} allele and 9 lines carrying the *BrJMJ18*^{Par} allele. In Figure S11A, we found that lines carrying the *BrJMJ18*^{Par} allele exhibited smaller differences in flowering time between ZJK and BJ. It is important to note that the flowering data for *DG* lines presented in Figure S11A, or Figure 1F (right panel), were collected after four weeks of vernalization, and all the *DG* lines had flowered by that time.

Fig. 3A: Is the scale bar 2cm?

RE: The scale bar represents 5 cm. We corrected the figure legend (lines 265 and 267-268).

Fig. 3G: Where is the broadened flowering time delay of ParBrJMJ18CR plants under HS conditions shown?

RE: In Figure 3G, *Par* flowered at approximately 51 days under NC, while it flowered at around 70 days under HS, resulting in a difference of 19 days in flowering time. On the other hand, the *BrJMJ18*^{Par-CR} plants flowered at approximately 60 days under NC conditions, but within the observed time window of 120 days, they did not flower under HS conditions. Therefore, we concluded that the flowering time

delay in *BrJMJ18^{Par}-CR* is broadened. We now realize that Figure 3G may be confusing. The DAS of *BrJMJ18^{Par}-CR* plants under HS has been set as 120 days in the new text.

Figure 3G Flowering time (DAS) of *Par* and *BrJMJ18^{Par}-CR* under NC and HS conditions, respectively..

In HS, the CRISPR line never flowered, so how do you explain that in the tunnel greenhouse in the summer of 2021 it did? And why did the BrJMJ18PC-OX not flower early there?

RE: Yes, as you said, this result was also confusing for us. One potential explanation is that the heat stress experiment conducted in the tunnel greenhouse was carried out under natural field conditions, where we couldn't artificially regulate the temperature. Consequently, the day temperatures often exceeded 50°C, imposing an excessively severe stress on the plants. This extreme high-temperature stress could lead to early flowering in all plants and narrow down the phenotypic differences between different overexpression lines.

Fig. 4B: Activity under NC conditions also appears different in tobacco leaves.

RE: Thank you for your reminding. In Figure 4B (Figure 4C in the new text), there was a discrepancy in the loading of *BrJMJ18^{Par}:gfp* compared to *BrJMJ18^{PC}:gfp*, which resulted in the appearance of a difference in enzyme activity, as you mentioned. We repeated the experiment and ensured the loading of *BrJMJ18* in each reaction assay is equal, and the results showed that there was no difference in enzyme activity between *BrJMJ18^{Par}:gfp* and *BrJMJ18^{PC}:gfp* under NC. We have made the necessary corrections to the figure accordingly.

Figure 4 (C) H3K36me3 and H3K36me2 status in *35S::BrMJ18^{WT}:gfp* and *35S::BrMJ18^{Par}:gfp* expressing tobacco leaves under normal conditions (NC) for 3 days and NC for 2 days following 1-day of HS, respectively *35S::gfp* expressing tobacco leaves were used as a negative control. BrMJ18-gfp proteins were detected with anti-GFP antibody to confirm equal expression of exogenous genes. Ponceau S staining was used to assess equal loading.

Fig. 5B: Please indicate bolting time here.

RE: The *Par* bolted at 25DAP, we revised Figure 5B and legend as suggested (lines 403-404). DAP, days after planting (14-day *Par* seedlings grown in cultivation pots in greenhouse were transplanted into natural field).

Fig. S16: As the role of FLC is in the initiation of flowering (at least in Arabidopsis), why is FLC investigated after bolting? Also, no y-axis numbers.

RE: Apologies for the confusion in the description. We actually examined the expression of *FLC* at two stages, before and during bolting. Y-axis numbers has been added in the Figure S16

Fig. S17: AtFT levels do not seem to match flowering time pattern. Can you comment?

RE: I apologize for presenting incorrect data. At that time, we examined two genes, *AtTSF* and *AtFT*, and mistakenly displayed the data for *AtTSF* here. We have repeated the Q-PCR experiment and provided the data for *AtFLC* and *AtFT*, and it aligns with our expectations. The results for *AtTSF* were not as anticipated, which could be attributed to its low expression level, potentially affecting the accuracy of detection.

Figure S17. Expression of *AtFLC* and *AtFT* in *AtJMJ18^{pro}::BrJMJ18^{PC}:gfp* and *AtJMJ18^{pro}::BrJMJ18^{Par}:gfp* plants under different temperatures.

(A) *AtFLC* expression decreased in *AtJMJ18^{pro}::BrJMJ18^{PC}:gfp* but increased in *AtJMJ18^{pro}::BrJMJ18^{Par}:gfp* plants upon heat treatment. (B) The induction degree of the *AtFT*'s expression by high temperature was in strongly weakened in *AtJMJ18^{pro}::BrJMJ18^{Par}:gfp* transgenic plants, which is consistent with their flowering time variations. Three-week old plants grown under normal conditions (NC), and two-week old plants grown under NC following 1-week of heat stress (HS) were used for the analysis. Actin was used as internal control. The values are the mean \pm standard deviation from three biological replicates.

Fig. 5F: It is surprising that there is no *FLC3* expression in *DG* (according to Fig. S16) but high *K36*. Can you comment?

RE: Figure S16 presents the data obtained from RNA-seq analysis. Technically, RNA-seq has lower sensitivity compared to qPCR. Therefore, we retested the expression of *BrFLC3* in *DG* using qPCR. Our findings confirmed that *BrFLC3* is expressed in *DG*.

Figure S16. *BrFLC3* is one of the two expressed *BrFLCs* and the only downregulated *BrFLC* during floral transition in *Par*.

A) The expression data derived from RNA sequencing data of *PC*, *DG* and *Par* plants grown under natural field conditions

before and during bolting. (B) *BrFLC3* expression in *PC*, *DG* and *Par* grown under natural field conditions before and during bolting detected by Q-PCR. *GADPH* was used as internal control.

Fig. S19: Claims are too strong since the results are not consistent in the two lines.

RE: We have now excluded this data from the main text. Furthermore, we have made substantial reduction to the description of the *Arabidopsis* results, in line with the recommendations provided by both you and reviewer 2.

Fig. 7A, B: Why are there so few overlapping? Is this expected? In NC the plants generally behaved similarly, right?

RE: We supposed that, despite not being entirely satisfactory, this result is acceptable after conducting multiple biological replicates. Based on the ChIP-seq data, we found that 32.8% of the enriched genes in *BrJMJ18^{PC}-OX* plants and 28.6% in *BrJMJ18^{Par}-OX* plants under NC conditions overlapped with previously reported H3K36me3-targeted genes. The Gene Ontology (GO) assay demonstrated that under NC conditions, the significantly enriched items of *BrJMJ18^{PC}* and *BrJMJ18^{Par}* showed remarkable consistency (Supplementary Figure S21 A, B), which align with the fact that under NC the plants generally behaved similarly. On the other hand, it is worth noting that this finding is corroborated by the RNA-seq data, where approximately 60% of the differentially expressed genes (DEGs) in both *BrJMJ18^{PC}-OX* and *BrJMJ18^{Par}-OX* plants were found to be overlapping under NC (Table S9).

Fig. S23: Not clear what is shown here. Can you clarify in text and with arrows or clearer image?

RE: Here we want to show that under NC conditions, all three *BrJMJ18* proteins (*BrJMJ18^{Par(A345T)}*, *BrJMJ18^{Par(C633Y)}*, and *BrJMJ18^{Par(C633Y)/(F654L)}*) exhibited nuclear localization, similar to *BrJMJ18^{PC}* and *BrJMJ18^{Par}*. Additionally, our findings indicated that the nuclear localization of *BrJMJ18^{Par(A345T)}*, *BrJMJ18^{Par(C633Y)}*, and *BrJMJ18^{Par(C633Y)/(F654L)}* remained unaffected by HS (Supplementary Figure 23). We removed Figure S23 and description as the subcellular localization is not logically related to the demethylase activity difference between *BrJMJ18^{PC}* and *BrJMJ18^{Par}* under HS.

Fig. 8A: It seems that maybe the terms "activity" and "H3K36 signal" are mixed up in the discussion of this figure and in figure legend. e.g. where it is said that in NC F654L had higher activity than WT.

RE: I apologize for the confusion in using the terms "activity" and "H3K36 signal." We have made the necessary corrections throughout the entire manuscript, and all relevant descriptions have been unified as "activity."

Reviewer #5 (Remarks to the Author):

Note to the authors: I have been asked to review the population genetics part only, as there was obviously a gap in the reviews before. I therefore read the whole manuscripts, but focus my review to the first part.

RE: We thank you for your insightful comments and suggestions. We have implemented all of your applicable recommendations to improve the quality of the presented work, please see details in our point-by-point response below. With these improvements, we hope that you will find the manuscript much improved.

Some typos:

Abstract

l27: provide, not provides

RE: “provides” was corrected as “provide” in line 26.

Introduction:

First sentence confusing, please rewrite

RE: Thanks for reminding. We revised the sentence as: Temperature plays a crucial role in determining the pace of plant development. Given the current scenario of climate change, rising temperatures can accelerate flowering and shorten developmental phases in crops. This may lead to significant reductions in agricultural yields, posing a widespread risk of food insecurity (lines 33-36).

l52 exists, not exits

RE: we reworded the corresponding text (line 51).

More critical:

MM:

The methods used to perform the QTL study are not described. There is a hint that population genetics were performed as described in Su et al 2018, 1)but the description there does not fit to the presented figure (2D), as this presents physical distances and SNP indices instead of genetic distances (in cM) and a LOD score. So please either clarify the figure or add a proper description of the method. 2) Moreover, looking at Supplementary Figure S8, it looks as if the JMJ8 locus is not closely linked to the QTL, at least falls out of the major LD group.

RE: 1) With regarding to the figure 2D, we now present an explanation of how we carried out the QTL analysis below. An F2 population, comprising approximately 350 offspring, was generated through a cross between Par and PC for our analysis. The germinated seeds were planted in soil and cultivated under NC

conditions at 22°C/22°C with a 16-hour light and 8-hour dark period for 4 weeks. Following this, half of the seedlings were kept under NC conditions until their flowering time data were recorded, while the other half were transferred to HS conditions at 29°C/29°C with a 16-hour light and 8-hour dark period until they flowered. Flowering-related quantitative trait loci (ftQTLs) responding to different temperatures were then identified by using BSA analysis. For BSA analysis under NC conditions, we obtained 31.2 and 25.4 Gb of high-quality bases from the DNA bulks of the early bolting (n=25) and late bolting (n=25) progenies after data filtration, respectively. The average depth was about 60× and 52× in the early and late bolting bulks, respectively. A total of 2,904,351 and 2,790,600 SNPs were identified in the early and late bolting bulks, respectively, and the SNP index was calculated for each SNP. By examining the Δ (SNP index) plot at 95 % confidence as reported by Abe et al, (2012), we found highly contrasting patterns in the SNP-index graph for the early bolting and late bolting bulks in 9 regions (bQTL, BSA-QTL; Table S6). While for BSA analysis under HS conditions, a total of 2,304,121 (n=25, 24.5 Gb, 50×) and 2,369,714 SNPs (n=25, 25.5 Gb, 50×) were identified in the early and late bolting bulks, respectively. By examining the Δ (SNP index) plot, a total of 14 heat-responsive flowering time QTLs were determined. Our focus then shifted towards QTLs that manifested solely under HS conditions and were absent under NC conditions (please see details in Table S6).

Building upon the analyses above, we further investigated the interval on chromosome A09 (indicated by the yellow column in the diagram below) since it overlaps with the region under specific selection in the domestication analysis of *Par/DG*. The enlarged version of this interval is shown in Figure 2D. To plot it, we extracted the delta index values of each SNP variation from the BSA data and their corresponding physical positions for this interval. We then plotted Figure 2D in Excel, with delta index values on the vertical axis and physical positions on the horizontal axis.

We added the details into the lines 187-191, 814-825 and Table S6.

2) We sincerely apologize for the inaccurate annotation. Upon careful examination of BrJMJ18's physical position on the chromosome, we realized that the originally marked location was slightly off-center. We have made the necessary adjustment to position it more accurately. However, it's important to note that even with this correction, BrJMJ18 is not situated at the highest point or the center of this interval, as mentioned by the reviewer. Nevertheless, our subsequent transgenic, biochemical, and phenotypic experiments all provide compelling evidence that BrJMJ18 is the candidate gene.

It's also worth noting that such examples are not uncommon in research. In a study by Yano et al. (2016, Nature Genetics), they also observed cases where some genes were not located at the peak of the significant GWAS interval, and some were even outside the interval but closely linked to the neighboring block. They attributed this challenge to two main factors. Firstly, diversity panels of crop species often

have strong population structure, leading to spurious associations between phenotypes and unrelated markers. Secondly, the large extent of linkage disequilibrium (LD) in plants, especially for the self-pollinating crops (the population used in our study are all self-incompatible lines.), can span several hundred kilobases. This includes many candidate genes within a single LD block showing significant signals, requiring further experiments to pinpoint the causal gene(s). This might be influenced by different computational methods used for analysis.

References:

- Abe A, Kosugi S, Yoshida K, Natsume S, Takagi H, Kanzaki H, Matsumura H, Yoshida K, Mitsuoka C, Tamiru M, Innan H, Cano L, Kamoun S, Terauchi R. Genome sequencing reveals agronomically important loci in rice using MutMap. *Nat Biotechnol.* 2012 Jan 22;30(2):174-8. doi: 10.1038/nbt.2095. PMID: 22267009.
- Yano, K., Yamamoto, E., Aya, K., Takeuchi, H., Lo, P.-c., Hu, L., Yamasaki, M., Yoshida, S., Kitano, H., Hirano, K., et al. (2016). Genome-wide association study using whole-genome sequencing rapidly identifies new genes influencing agronomic traits in rice. *Nature genetics* 48, 927-934.

Figure S8 is also not really described, the description keeps talking about a qQTL (which indicates a phenotype-genotype-association) but only π -values (nucleotide diversity) is shown.

RE: Thank you for your reminder. As you mentioned that there is some overlap display between the data in Figure S8 and Figure 2E. In the new figure, we have retained only the data related to the LD block and provided a brief discussion on it: In addition, BrJMJ18 was further found to be located on a LD block on 09, from 26.6 to 27.3 Mb, indicating a strong signature of selection of the locus.

Table S6 also does not show LOD scores for the respective peaks. In other words, the statement that this QTL was the highest is not shown here.

RE: Apologies for the inaccuracies in the description; we have since removed the relevant statement in the revised text.

Figure 1A: really small, not really readable. I would advice either to move to Supplementary or to change size.

RE: The original Figure 1A and C have been moved to the supplementary Figure S1 A and C, respectively. We have also revised Figure 1 to ensure its readability by enlarging it appropriately.

Figure 1B: please add the phylogenetic outgroup to the caption.

RE: In Figure 1B, an unrooted phylogenetic tree was constructed without an outgroup. In previous studies, it has been confirmed that turnip is the most ancestral population of *Brassica rapa* (Takuno et al., 2007; Cheng et al., 2016; Bird et al., 2017; Qi et al., 2017). Therefore, in our research, turnip was consider as the root to judge the evolution of *Brassica rapa* family.

References:

- Takuno, S., Kawahara, T., & Ohnishi, O. (2007). Phylogenetic relationships among cultivated types of *Brassica rapa* L. em. Metzg. as revealed by AFLP analysis. *Genetic resources and crop evolution*, 54, 279-285.
- Cheng, F., Sun, R., Hou, X., Zheng, H., Zhang, F., Zhang, Y., & Wang, X. (2016). Subgenome parallel selection is associated with morphotype diversification and convergent crop domestication in *Brassica rapa* and *Brassica oleracea*. *Nature genetics*, 48(10), 1218-1224.
- Bird, K. A., An, H., Gazave, E., Gore, M. A., Pires, J. C., Robertson, L. D., & Labate, J. A. (2017). Population structure and phylogenetic relationships in a diverse panel of *Brassica rapa* L. *Frontiers in plant science*, 8, 321.
- Qi, X., An, H., Ragsdale, A. P., Hall, T. E., Gutenkunst, R. N., Chris Pires, J., & Barker, M. S. (2017). Genomic inferences of domestication events are corroborated by written records in *Brassica rapa*. *Molecular ecology*, 26(13), 3373-3388.

Figure 1C: no black arrows as indicated in the caption

RE: The description was removed in the legend.

Figure 1D: do the estimated times since the respective splits make sense in a historical context? This should possibly be added to the discussion.

RE: It is reported that East Asian leafy *B. rapa* crops likely evolved from turnips (McAlvay et al, 2021; Qi et al, 2017). Through genome sequencing, we found that PC is the closest to the ancient group, while Par is the most recent. Our analysis suggests that Par split from DG around 1,838-3,050 years ago, and DG split from PC approximately 3,000 years ago. In **Figure S24**, we compared the estimated demographic modeling with the written history of leafy *B. rapa* domestication events in China. Turnips was described as “Feng” (葍) in the oldest Chinese poetry collection, Shi Jing (Classic of Poetry), about 3,100-2,600 years ago (Luo, 1992; Ye, 1989). And PC was called “Song” (菘) and was firstly described in the oldest Chinese encyclopedia, Er Ya (Literary Expositor), about 3,000-2,700 years ago in our country. Additionally, the word “Song” was used as a general term for leafy *B. rapa* crops, and different types of “Song” were recorded. For instance, three types of “Song” were recorded in Xin Xiu Ben Cao (Newly Revised Canon of Material Medica, 1364 years ago): “Niu Du Song (Big-Tummy Song)” with large and curved leaves, “Zi Song (Purple Song)” with purple and slightly bitter leaves (probably purple Pak Choi), and “Bai Song (White Song)” with white petioles and dark green leaves. According to the morphological features, the Bai Song is likely to be the DG. With regarding to *Par*, *Par* was recorded to be cultivated in the Taihu Lake area of China in the Song Dynasty (AD 960-AD 1,279) and was mentioned in a poetry Cai Geng by poet Lu You (AD 1,125 – AD 1,210). We have organized the mentioned timelines (shown in **Figure S24**), and we found that the recorded order of PC, DG, and *Par* corresponds to the appearance times inferred from genome sequencing. However, historical records of these times are indeed behind the predicted times from genome sequencing. We speculate that this could be due to the fact that species are generally not recorded immediately after their formation but before they are fully developed and widespread. We added these

sentences into Discussion (lines 592-597) and Figure S24 legend.

Figure S24 A combined summary of estimated demographic modeling and written history of leafy *B. rapa* domestication events.

The black horizontal arrow represents time scale. The pink callouts above represents the historical written records of *PC*, *DG* and *Par* (Subspecies, source, time). The green and red bars below represents the estimated split time of *DG* and *Par* based on our $\partial a\partial i$ analysis.

References:

McAlvay, A.C., Ragsdale, A.P., Mabry, M.E., Qi, X., Bird, K.A., Velasco, P., An, H., Pires, J.C., and Emshwiller, E. (2021). Brassica rapa Domestication: Untangling Wild and Feral Forms and Convergence of Crop Morphotypes. *Molecular biology and evolution* **38**, 3358-3372

Qi, X., et al. Genomic inferences of domestication events are corroborated by written records in Brassica rapa. *Mol Ecol* **26**, 3373-3388(2017).10.1111/mec.14131

Luo, G. A analysis of historical materials about the time and place of the origin of the peking cabbage. *Studies in the History of Natural Sciences* **2**, 171-176(1992).

Ye, J. The name and reality of the rape (Brassica campestris and Brassica juncea) in China. *Studies in the History of Natural Sciences* **8**, 158-165(1989).

Figure 1 E: I personally find it hard to grasp the message here. What's the information in the vertical direction?

RE: In the process of generation of this TreeMix figure, information regarding gene flow was incorporated onto the *B. rapa* phylogenetic tree. All these manipulations were performed using the TreeMix computational software implementing the model described above. As a result, the vertical axis here essentially represents the stacked evolutionary branches of various *B. rapa* groups.

Figure 2A. the horizontal line is red, not blue

RE: Sorry about the mistake. We corrected the figure legend (line 198) as suggested.

Figure 2C this does not show genes, but the results of GO enrichment, caption has to be adapted

RE: We corrected the caption as "GO terms associated with reproduction and abiotic stress were specifically enriched in the *Par/DG* comparison" as suggested. And Figure 2C was moved to Figure S6A in the new text.

Figure 2D: this is at least an unusual QTL representation, why isn't it presented as LOD score, and why not over the full chromosome?

RE: Please refer to the detailed procedure for generating Figure 2D in our response to the "more critical Comment" 1 above.

Figure 2F: the caption should explain what JmjN, JmjC, Zf-C5HC, FYR and FYRC stands for.

RE: In Lu et al., (2008), by merging information between the phylogenetic tree and the domain architecture of each protein, they classified the JmjC domain-containing proteins into eight groups: KDM6/JMJD3 group, KDM5/JARID1 group, KDM4/JHDM3 group, KDM3/JHDM2 group, KDM2/JHDM1 group, PHF group, JMJD6 group, and JmjC domain-only group. Based on its structural features, BrJMJ18 is categorized within the KDM5/JARID1 group, characterized by five distinct domains: JmjN (Jumonji N domain), JmjC (Jumonji C domain), zf-C5HC2 (Zinc finger of C5HC2-type), FYRN ("FY-rich" domain N-terminal), and FYRC ("FY-rich" domain C-terminal).

The statement above was substantially revised and added in the Figure 2F legend (lines 214-216).

Reference:

Lu, F., Li, G., Cui, X., Liu, C., Wang, X. J., & Cao, X. (2008). Comparative analysis of JmjC domain-containing proteins reveals the potential histone demethylases in Arabidopsis and rice. *Journal of Integrative Plant Biology*, 50(7), 886-896.

Figure 2G; Was there a difference in FTi in the DG lines carrying the *Par* allele? Did the three DG lines flowering without vernalisation (Figure 1F) carry a *Par* allele?

RE: I supposed the question is "Was there a difference in flowering time in the DG lines carrying the *Par* allele"? Yes, as depicted in Figure S11, there are significant differences in flowering time among the DG lines harboring the *Par* allele. In the three DG lines flowering without vernalisation, two of them carry *Par* alleles, while the other one is carrying the DG allele.

REVIEWER COMMENTS

Reviewer #1 (Remarks to the Author):

I am pleased to see that the authors have addressed my previous concerns and improved the manuscript in this second version. While some of the claims made by the authors may be too strong for the data presented, overall, this work represents a significant step forward in the field and will be relevant for different areas of plant biology.

However, my major concern now is that although the authors have included the bioinformatic methods and GEO accession number, I cannot find a table indicating the sequencing metrics (such as the number of total reads, mapped reads, etc.) of the different sequencing experiments performed (RNA-seq, ChIP-seq). I suggest that the authors include this information in the supplementary data to improve the reproducibility and transparency of their work

Minor points:

Line 28. NC, not defined, it is better to say normal conditions in the abstract

Line 372. FigS19 is mentioned before other FigS, please number them according to the text.

Figure 5. A cartoon showing the analyzed BrFLC3 region by ChIP would be useful.

Figure 5. Please indicate in the legend the tissue used for ChIP and qPCR.

Line 627. I suppose you mean Mehraj

Reviewer #2 (Remarks to the Author):

The authors have addressed my questions, and the manuscript now reads more smoothly. I am pleased with the improvements in the updated manuscript. However, I can still find areas where further improvement is needed:

1. While the authors have added information about biological replicates, this information is still missing in most of the western blot results, which is also important for evaluation.

2. The authors should also carefully review their writing again, as evidenced by the issue on line 428: "AtJMJ18pro::BrJMJ18PC/Par:gfArabidopsis plants.

Reviewer #3 (Remarks to the Author):

This manuscript has been improved in response to this and other reviewers' comments. However, I still find there are outstanding issues to be addressed.

1. Line 64-66: "Vernalization-requiring morphotypes often exhibit delayed flowering under high temperatures due to de-vernalization effects."

I was unable to retrieve the content from two referenced sources (5, 6), and thus I am not sure what the authors exactly meant here. Nonetheless, it is my understanding that de-vernalization refers to a phenomenon where interruption of elevated temperatures counteracts the effects of vernalization "during" cold treatment, rather than delaying flowering under high growth temperatures, as the authors might suggest. Therefore, it is not appropriate to state that the vernalization-requiring morphotypes delay flowering under high temperatures.

2. Line 257: "F2 population" it looks like that the authors refer to "Par X PC F2 population in Fig. 2C". Authors can clarify this.

3. In Par, JMJ18 decreases at BrFLC3 with HS 1w (Fig. 5A), but H3K36me3 does not change (Fig. 5D). In addition, BrFLC expression actually decreases under HS (Fig. 5C). BrFLC3 increases only in BrJMJ18par-OX lines (Fig. 5C).

The authors interpreted this as "Line: 389-390: Under NC, BrFLC3 expression decreased markedly in both BrJMJ18PC- and BrJMJ18Par-OX plants, while under HS, it was decreased in BrJMJ18PC-OX but increased in BrJMJ18Par-OX plants (Figure 5 C), which is in line with their flowering time variation under HS (Figure 3 C,D)." However, Par does not show such correlation (Fig. 5C,D). How do the authors explain this?

4. Line 432-434: "These results proposed that the late flowering of BrJMJ18Par-OX under HS conditions primarily results from the dissociation of BrJMJ18Par from BrFLC3, leading to the inability to repress its expression"

This is a testable model. Do JMJ18-PC and JMJ18-Par enrich at BrFLC3 under HS differentially? The authors already have the antibody to perform CHIP-qPCR experiments and indeed the authors already have CHIP-seq data using these lines. Is this the case?

5. Related to CHIP-seq analysis:

a. One of authors' interpretation regarding the variations in BrJMJ18 pertains to their differential bindings to target loci. The authors only indicated the changes in targets (presumably based on different peak callings of CHIP-seq). It is more relevant to identify loci that exhibit quantitative changes in terms of BrJMJ18 enrichment.

b. In addition, two replicates are not sufficient for statistically sound analysis of CHIP-seq data.

6. While the manuscript includes descriptions of statistical analyses, there are no indications of statistical significance on the graphs. For example, Line 400: Asterisks indicate significant differences, Student's t-

test (* $p < 0.05$, ** $p < 0.01$, *** $p < 0.001$). But there is no asterisk on the graph.

7. Fig. 5A. Control graphs are collapsed, and it is impossible for me to interpret the data.

8. Growth phenotypes of BrJMJ18 should be carefully interpreted. After all, Par lines do not show such phenotypes under HS (Fig. 6A) and the authors are analyzing ectopically expressed transgenic lines.

Reviewer #4 (Remarks to the Author):

The authors have addressed most comments in this revision. There are three points I would like to comment again on, based on their response.

I appreciate the added text in the introduction (lines 75-84), which explains the context for the temperature used. However, I still think that the average summer temperature (and as they suggest “the approximate threshold for Par domestication”) is not a good indicator of “heat stress”. My suggestion would be to change the term “heat stress” when referring to 29°C.

The model in Fig S25 suggests high temperature represses vegetative growth by repressing BrJMJ18Par. Is that right? Furthermore, the term “stabilize flowering and vegetative growth” in the legend is unclear.

I feel the answer to my question 6 does not address the issue of why the Par OX line might lead to more H3K36me3 at BrFLC3, and higher BrFLC3 expression, as shown in figure 5, compared to the wt Par line, in HS conditions. (this is a minor comment)

Reviewer #5 (Remarks to the Author):

I am happy with the changes made and thank you for your valuable work. There is an inconsistency in one of your explanations (“Secondly, the large extent of linkage disequilibrium (LD) in plants, especially for the self-pollinating crops (the population used in our study are all self-incompatible lines.)”)  so LD should be low here! but I still accept the other part of the explanation and the changes made.

REVIEWER COMMENTS

Reviewer #1 (Remarks to the Author):

I am pleased to see that the authors have addressed my previous concerns and improved the manuscript in this second version. While some of the claims made by the authors may be too strong for the data presented, overall, this work represents a significant step forward in the field and will be relevant for different areas of plant biology.

RE: We thank you for your insightful comments. We have implemented all of your applicable recommendations to improve the quality of the presented work, please see details in our point-by-point response below.

However, my major concern now is that although the authors have included the bioinformatic methods and GEO accession number, I cannot find a table indicating the sequencing metrics (such as the number of total reads, mapped reads, etc.) of the different sequencing experiments performed (RNA-seq, ChIP-seq). I suggest that the authors include this information in the supplementary data to improve the reproducibility and transparency of their work.

RE: Thank you very much for the comments. Due to variations in data formats and quality assessment criteria of different sequencing experiments, it is not applicable to present them in a unified table. Alternatively, we have separately provided relevant sequencing metrics in Tables S7-10. Please refer to the details in the corresponding Supplementary Tables.

Minor points:

Line 28. NC, not defined, it is better to say normal conditions in the abstract

RE: “NC” was replaced with “normal conditions” as suggested in line 28.

Line 372. FigS19 is mentioned before other FigS, please number them according to the text.

RE: We renumbered Figures S 15-19 according to their order. Please see changes in lines 376, 383, 385, 387, 395, 426, 435 and 685 in the new text and Supplementary Figures.

Figure 5. A cartoon showing the analyzed *BrFLC3* region by ChIP would be useful.

RE: We added cartoons showing the analyzed region of *BrFLC1-3, 5* and *BraA06g002250.3C* in the new Fig. S16 E.

Figure S16 E Cartoons showing the analyzed region of *BrFLCs* and *BraA06g002250.3C* by ChIP-QPCR. Black boxes represent the exons and black bars between them represent introns. Analyzed regions are represented by the red bars.

Figure 5. Please indicate in the legend the tissue used for ChIP and qPCR.

RE: The 5th-7th young and healthy rosette leaves of *Par* plants grown under NC and HT conditions were used for ChIP and qPCR tests (*HS (heat stress) was replaced with HT (high temperature) as required by one of the reviewers*). We added these details into the Figure 5 legend (lines 400-401, 408-409 and 414-415).

Line 627. I suppose you mean Mehraj

RE: We corrected the name as “Mehraj” in line 637.

Reviewer #2 (Remarks to the Author):

The authors have addressed my questions, and the manuscript now reads more smoothly. I am pleased with the improvements in the updated manuscript. However, I can still find areas where further improvement is needed:

RE: We thank you for your insightful comments. We have implemented all of your applicable recommendations to improve the quality of the presented work, please see details in our point-by-point response below.

1. While the authors have added information about biological replicates, this information is still missing in most of the western blot results, which is also important for evaluation.

RE: All the western blot experiments were repeated at least three times with similar results. We have presented data from one representative experiment. We added the details to legends of Figure 4 and Figure 8. Please see lines 359 and 597.

2. The authors should also carefully review their writing again, as evidenced by the issue on line 428: "AtJMJ18pro::BrJMJ18PC/Par:gfArabidopsis plants."

RE: We corrected the writing as "AtJMJ18pro::BrJMJ18^{PC/Par}:gfp Arabidopsis plants" in line 436 as suggested. And we have also carefully checked and corrected the spelling and formatting of the text, Please see changes in revision mode of the MS word file.

Reviewer #3 (Remarks to the Author):

This manuscript has been improved in response to this and other reviewers' comments. However, I still find there are outstanding issues to be addressed.

RE: We thank you for your insightful suggestions. We have implemented all of your applicable recommendations to improve the quality of the presented work, please see details in our point- by-point response below.

1. Line 64-66: *"Vernalization-requiring morphotypes often exhibit delayed flowering under high temperatures due to de-vernalization effects.*

I was unable to retrieve the content from two referenced sources (5, 6), and thus I am not sure what the authors exactly meant here. Nonetheless, it is my understanding that de-vernalization refers to a phenomenon where interruption of elevated temperatures counteracts the effects of vernalization "during" cold treatment, rather than delaying flowering under high growth temperatures, as the authors might suggest. Therefore, it is not appropriate to state that the vernalization-requiring morphotypes delay flowering under high temperatures.

RE: We apologize for our incomplete description. Your feedback has made us realize that dividing *B. rapa* morphotypes based on the requirement for vernalization may not accurately represent previous research on the impact of high temperatures on *B. rapa* crops.

Here, we would like to emphasize the effect of high temperatures on the flowering time of *B. rapa* plants during their vegetative growth. We thus reworded the text as "High temperatures can impact the flowering time of *B. rapa* in various ways. Please see lines 64-75.

2. Line 257: *"F2 population" it looks like that the authors refer to "Par X PC F2 population in Fig. 2C". Authors can clarify this.*

RE: "... confirmed in the natural DG and *Par* group and the F2 population" were replaced with "confirmed in the natural DG group and the "*Par* × *PC*" F2 population" in line 259.

3. In *Par*, JMJ18 decreases at *BrFLC3* with HS 1w (Fig. 5A), but H3K36me3 does not change (Fig. 5D). In addition, *BrFLC* expression actually decreases under HS (Fig. 5C). *BrFLC3* increases only in *BrJMJ18par-OX* lines (Fig. 5C).

The authors interpreted this as “Line: 389-390: Under NC, *BrFLC3* expression decreased markedly in both *BrJMJ18PC-* and *BrJMJ18Par-OX* plants, while under HS, it was decreased in *BrJMJ18PC-OX* but increased in *BrJMJ18Par-OX* plants (Figure 5 C), which is in line with their flowering time variation under HS (Figure 3 C,D).” However, *Par* does not show such correlation (Fig. 5C,D). How do the authors explain this?

RE: Previous researches demonstrate that the expression of FLC is regulated at multiple levels, including chromatin regulation, transcription level, and co-transcriptional RNA metabolism. In our study, by our demethylation assay, RNA-seq and ChIP-qPCR, we demonstrated that *BrJMJ18* is a novel H3K36me_{2/3} demethylase affecting the H3K36me_{2/3} level of *BrFLC3*. And HT condition represses the binding and H3K36me_{2/3} demethylation of *BrFLC3* by *BrJMJ18^{Par}* in *Par* (HS (heat stress) was replaced with HT (high temperature) as required by one of the reviewers). However, we also believe that other regulatory pathways continue to function, leading to changes in the expression of *BrFLC3* under HS. As pointed out by the reviewer, *BrJMJ18* decreases after 1 week of HT at *BrFLC3* (Fig. 5A) in *Par*, but H3K36me₃ levels do not change (Fig. 5D). We speculate that this lack of change in H3K36me₃ levels is not solely attributed to the action of *BrJMJ18* but rather involves other factors, such as H3K36me_{2/3} writers or erasers, working in concert to bring about this change. Also as mentioned by the reviewer, the fact that H3K36me₃ remains unchanged logically suggests that *BrFLC3* expression should also remain unchanged. However, *BrFLC3* expression actually decreases under HT in *Par*. This implies that *BrFLC3* expression is not only subject to epigenetic regulation but may also involve regulation at other levels when responding to high temperatures. This is the outcome of the collaborative action of multi-level regulation. However, in *BrJMJ18^{Par}-OX* plants, the expression of *BrFLC3* significantly increases under HT as expected. We propose that in *BrJMJ18^{Par}-OX* plants an excessive amount of *BrJMJ18^{Par}* proteins disrupts the balanced regulation of *BrFLC3* conferred by other regulatory mechanisms in wildtype *Par*, and amplifies the effect of *BrJMJ18^{Par}*,

leading to a dominant output of an increase in H3K36me3 levels and *BrFLC3* expression under HT as expected. We therefore appreciate the reviewer's insights, which have reminded us that while overexpression is a valuable research strategy to help elucidating gene regulation mechanisms, we should also consider other factors and data carefully to make better judgments.

4. Line 432-434: “These results proposed that the late flowering of *BrJMJ18Par-OX* under HS conditions primarily results from the dissociation of *BrJMJ18Par* from *BrFLC3*, leading to the inability to repress its expression”. This is a testable model. Do *JMJ18-PC* and *JMJ18-Par* enrich at *BrFLC3* under HS differentially? The authors already have the antibody to perform ChIP-qPCR experiments and indeed the authors already have ChIP-seq data using these lines. Is this the case?

RE: We conducted Anti-GFP ChIP-qPCR using *Par* and *BrJMJ18-OX* plants grown under NC and HS, respectively. The ChIP-qPCR was performed as previously described in the Method section using GFP antibody (ab290, Abcam, Cambridge, MA, USA) and normal rabbit IgG (12-370, Sigma-Aldrich) was use as negative control. GADPH was used as a *BrJMJ18*-independent control. Under NC, both *BrJMJ18^{PC}:gfp* and *BrJMJ18^{Par}:gfp* proteins bind to *BrFLC3*. Under HT, the binding of *BrJMJ18^{PC}:gfp* to *BrFLC3* was aggravated by heat, while *BrJMJ18^{Par}:gfp* protein thoroughly disassociated from *BrFLC3*. We put these results into Fig. S16 B, and please see description changes in lines 384-385.

Figure S16 B To further test the binding of BrMJ18 to *BrFLC3*, anti-GFP ChIP-qPCR was conducted using *Par* and *BrMJ18^{PC/Par}-OX* plants grown under NC for 5 weeks or 4 weeks followed by 1 week HT treatment. Under NC, both BrMJ18^{PC}:gfp and BrMJ18^{Par}:gfp proteins bind to *BrFLC3*. Under HT, the binding of BrMJ18^{PC}:gfp to *BrFLC3* was aggravated by heat, while BrMJ18^{Par}:gfp protein thoroughly disassociated from *BrFLC3*.

5. Related to ChIP-seq analysis:

a. One of authors' interpretation regarding the variations in BrMJ18 pertains to their differential bindings to target loci. The authors only indicated the changes in targets (presumably based on different peak callings of ChIP-seq). It is more relevant to identify loci that exhibit quantitative changes in terms of BrMJ18 enrichment.

RE: We focused solely on BrMJ18 binding genes in the original manuscript is because BrMJ18^{Par}- and BrMJ18^{PC}-OX plants exhibited significant phenotypic differences under HT conditions. We supposed this might be due to their regulation of different gene sets. The reviewer's suggestion highlights the importance of identifying loci that exhibit quantitative changes in terms of BrMJ18 enrichment as a crucial basis for studying the functional changes of BrMJ18 under HS. We totally agree with this, and therefore we have included these data in the new Table S9.

b. In addition, two replicates are not sufficient for statistically sound analysis of ChIP-seq data.

RE: We performed two independent biological replicates of the ChIP-seq experiments, based mainly on two publications: "ChIP-seq guidelines and practices of the ENCODE and modENCODE consortia, Genome Res. 2012. 22: 1813-1831" and "PeakSeq enables systematic scoring of ChIP-seq experiments relative to controls. Nat. Biotechnol. 2209. 27: 66-75." These authors showed that more than two replicates did not significantly improve site discovery for ChIP-seq analysis; thus, they set the standard that all the ENCODE ChIP measurements should be performed on two independent biological replicates. In the newly published paper (Yang et al., 2020, *Nature Plants*), ChIP-seq assays were also performed on two replicates. Therefore, we proposed that the two-replicate guideline is appropriate for ChIP-seq analysis.

References:

Landt, S. G., Marinov, G. K., Kundaje, A., Kheradpour, P., Pauli, F., Batzoglou, S., ... & Snyder, M. (2012). ChIP-seq guidelines and practices of the ENCODE and modENCODE consortia. *Genome research*, 22(9), 1813-1831.

Rozowsky, J., Euskirchen, G., Auerbach, R. K., Zhang, Z. D., Gibson, T., Bjornson, R., ... & Gerstein, M. B. (2009). PeakSeq enables systematic scoring of ChIP-seq experiments relative to controls. *Nature biotechnology*, 27(1), 66.

Yang, X., Yan, J., Zhang, Z., Lin, T., Xin, T., Wang, B., ... & Huang, S. (2020). Regulation of plant architecture by a new histone acetyltransferase targeting gene bodies. *Nature Plants*, 6(7), 809-822.

6. While the manuscript includes descriptions of statistical analyses, there are no indications of statistical significance on the graphs. For example, Line 400: Asterisks indicate significant differences, Student's t-test ($*p < 0.05$, $**p < 0.01$, $***p < 0.001$). But there is no asterisk on the graph.

RE: Sorry for the mistake. We added indications of statistical significance to Fig. 5A, C-E and Fig. 6 B, C.

7. Fig. 5A. Control graphs are collapsed, and it is impossible for me to interpret the data.

RE: We redraw the Figure 5A to make it more readable.

Figure 5A Both BrJM18^{PC} and BrJM18^{Par} bind to the chromatin of *BrFLC3*, and high temperature aggravates the binding of BrJM18^{WT}, but not BrJM18^{Par}, to *BrFLC3*. Chromatin immunoprecipitation (ChIP) analysis of the BrJM18 level across *BrFLC3* were performed in *BrJM18^{Par}*-carrying *Par* and *BrJM18^{PC}*-carrying *DG* plants. Yong and healthy rosette leaves from five-week old plants grown under normal conditions (NC), and four-week old plants grown under NC following 1-week of heat stress (HS) conditions, were used for the analysis. Rabbit IgG was used as control. *GADPH* was used as a BrJM18-independent control. Control is a locus gene desert regions where BrJM18 does not bind. The values are the mean \pm standard deviation from three biological replicates. Asterisks indicate significant differences, Student's t-test ($*p < 0.05$, $**p < 0.01$, $***p < 0.001$).

8. *Growth phenotypes of BrJMJ18 should be carefully interpreted. After all, Par lines do not show such phenotypes under HS (Fig. 6A) and the authors are analyzing ectopically expressed transgenic lines.*

RE: We have changed the subheading to "Overexpression of BrJMJ18^{Par} Promotes Vegetative Growth under Both Greenhouse and Field High-Temperature Conditions." Additionally, we have reworded the text to provide a distinction in the phenotypic description between BrJMJ18 overexpressing and *Par* plants. Please see lines 445, 447-449 and 470.-

Reviewer #4 (Remarks to the Author):

The authors have addressed most comments in this revision. There are three points I would like to comment again on, based on their response.

RE: We thank you for your insightful comments. We have implemented all of your applicable recommendations to improve the quality of the presented work, please see details in our point- by-point response below.

I appreciate the added text in the introduction (lines 75-84), which explains the context for the temperature used. However, I still think that the average summer temperature (and as they suggest “the approximate threshold for Par domestication”) is not a good indicator of “heat stress”. My suggestion would be to change the term “heat stress” when referring to 29°C.

RE: Thanks for the kind reminding. We have changed “heat stress (HS)” to “high temperatures (HT)” in the text and Figures.

The model in Fig S25 suggests high temperature represses vegetative growth by repressing BrJMJ18^{Par}. Is that right? Furthermore, the term “stabilize flowering and vegetative growth” in the legend is unclear.

RE: Sorry for the unclear presentation, here we want to say that high temperature suppresses early flowering and promotes vegetative growth by repressing the H3K36me_{2/3} demethylase activity of BrJMJ18^{Par}. We redraw the Fig S25 to make it more readable, and also reworded the corresponding legend mentioned as “some *B. rapa* crops have developed the BrJMJ18^{Par} allele to repress early flowering and promote vegetative growth to counter against high temperatures...”

Figure S25 A working model of *BrJMJ18^{Par}* under different temperature conditions.

Under NC, the induction of *BrJMJ18^{PC}* and *BrJMJ18^{Par}* downregulates *BrFLC3* by demethylating its H3K36me3, consequently promoting flowering. Under high temperature conditions, the flowering promotion function of *BrJMJ18^{PC}* is strengthened in most *B. rapa* subspecies. However, some *B. rapa* crops have developed the *BrJMJ18^{Par}* allele to repress early flowering and promote vegetative growth to counter against high temperatures via a mechanism in which the binding and subsequent demethylation activity of *BrJMJ18^{Par}* of its downstream loci, including *BrFLC3*, is notably weakened by heat. The symbol "↓" signifies positive regulation towards downstream factors, "⊥" indicates negative regulation towards downstream factors, while the light blue bar arrow indicates the meaning of "results in".

I feel the answer to my question 6 does not address the issue of why the Par OX line might lead to more H3K36me3 at BrFLC3, and higher BrFLC3 expression, as shown in figure 5, compared to the wt Par line, in HS conditions. (this is a minor comment).

RE: I apologize for the previous response, where I may not have fully understood your question. In comparison to "Par", under HT conditions the H3K36 level of *BrFLC3* in the *BrJMJ18^{Par}-OX* line should logically decrease. However, in reality, it increases, resulting in higher *BrFLC3* expression, which causes delayed flowering in the plant. Previous researches demonstrate that the expression of *FLC* is regulated at multiple levels, including chromatin regulation, transcription level, and co-transcriptional RNA metabolism. In our study, by our demethylation assay, RNA-seq and ChIP-qPCR, we demonstrated that *BrJMJ18* is a novel H3K36me2/3 demethylase affecting the H3K36me2/3 level of *BrFLC3*. And HT condition represses the binding and H3K36me2/3 demethylation of *BrFLC3* by *BrJMJ18^{Par}* in *Par*. However, we also believe that other regulatory factors continue to function, leading to changes in the H3K36me2/3 status of *BrFLC3* under HS. We speculate that this change at *BrFLC3* in

H3K36me3 levels is not solely attributed to the action of BrJMJ18 but rather involves other factors, such as H3K36me3 writers or erasers, working in concert to bring about this change.

Here we propose that in *BrJMJ18^{Par}-OX* plants an excessive amount of BrJMJ18^{Par} proteins disrupts the balanced regulation of *BrFLC3* conferred by other regulatory mechanisms in wildtype *Par*, and amplifies the effect of BrJMJ18^{Par}, leading to a dominant output of an increase in H3K36me3 levels and *BrFLC3* expression under HT as expected. We therefore appreciate the reviewer's insights, which have reminded us that while overexpression is a valuable research strategy to help elucidating gene regulation mechanisms, we should also consider other factors and data carefully to make better judgments.

Reviewer #5 (Remarks to the Author):

I am happy with the changes made and thank you for your valuable work. There is an inconsistency in one of your explanations ("Secondly, the large extent of linkage disequilibrium (LD) in plants, especially for the self-pollinating crops (the population used in our study are all self-incompatible lines.)")  so LD should be low here! but I still accept the other part of the explanation and the changes made.

RE: Sorry for the mistake. Here we actually confused the concepts of LD (Linkage Disequilibrium) and LD block. What we meant to convey is that the population we utilized consists of self-incompatible lines, which have low LD but relatively large LD blocks (Luikart et al., 2003).

Reference:

Luikart, G., England, P. R., Tallmon, D., Jordan, S., & Taberlet, P. (2003). The power and promise of population genomics: from genotyping to genome typing. *Nature reviews genetics*, 4(12), 981-994.

REVIEWER COMMENTS

Reviewer #3 (Remarks to the Author):

Regarding responses to the previous points #3:

I understand that the regulation of FLC can be influenced by multiple factors, making the interpretation of transgenic line behaviors complex.

However, the main conclusion in this manuscript is the demonstration of an intriguing link between natural variations of H3K36me_{2/3} demethylase, BrJMJ18, and its regulatory impact under high-temperature conditions. One locus the authors focused on is the BrFLC3.

Par plants flower later at high temperatures (Fig. 3C) and the authors suggested that the reduction in BrJMJ18 activity (H3K36 demethylation) at BrFLC3 occurs due to diminished catalytic activity (Fig. 8) and decreased enrichment at BrFLC3 (Fig. 5A) at high temperatures. Therefore, the authors implicated that BrFLC3 levels increase to delay flowering at high temperatures.

However, this correlation is observed only in BrJMJ18-par OX lines. In non-transgenic Par plants, the H3K36me₃ level at BrFLC3 does not change significantly (Fig. 5D), and most notably, BrFLC3 levels are not increased at high temperatures (Fig. 5C).

Reviewer #4 (Remarks to the Author):

I thank the authors for their thoughtful responses. I am happy with the changes made and I believe this is a very valuable and interesting study.

REVIEWER COMMENTS

Reviewer #3 (Remarks to the Author):

Regarding responses to the previous points #3:

I understand that the regulation of FLC can be influenced by multiple factors, making the interpretation of transgenic line behaviors complex.

However, the main conclusion in this manuscript is the demonstration of an intriguing link between natural variations of H3K36me2/3 demethylase, BrJMJ18, and its regulatory impact under high-temperature conditions. One locus the authors focused on is the BrFLC3.

Par plants flower later at high temperatures (Fig. 3C) and the authors suggested that the reduction in BrJMJ18 activity (H3K36 demethylation) at BrFLC3 occurs due to diminished catalytic activity (Fig. 8) and decreased enrichment at BrFLC3 (Fig. 5A) at high temperatures. Therefore, the authors implicated that BrFLC3 levels increase to delay flowering at high temperatures. However, this correlation is observed only in BrJMJ18-par OX lines. In non-transgenic Par plants, the H3K36me3 level at BrFLC3 does not change significantly (Fig. 5D), and most notably, BrFLC3 levels are not increased at high temperatures (Fig. 5C).

RE: We thank you for your insightful comments.

The reviewer pointed out that under high-temperature (HT) conditions, there is an increase in BrFLC3 levels to delay flowering, which is supported by the examination of H3K36me2/3 enrichment and BrFLC3 expression in BrJMJ18^{Par}-OX plants. This correlation is specifically observed in BrJMJ18^{Par}-OX lines. However, in non-transgenic Par plants, the situation is different: the H3K36me3 level at BrFLC3 remains largely unchanged (Fig. 5D), and there is no significant increase in BrFLC3 levels at high temperatures (Fig. 5C). We propose that these differences in response are likely due to the multi-level regulation of FLC, including chromatin remodeling, transcriptional control, and co-transcriptional RNA metabolism. Specifically, in BrJMJ18^{Par}-OX plants, an excess of BrJMJ18^{Par} proteins may disrupt the balanced regulation of BrFLC3 seen in wild-type Par plants. This disruption could intensify the influence of BrJMJ18^{Par} and result in significant increases in H3K36me2/3 levels and BrFLC3 expression under HT conditions. On the other hand, in WT Par plants, the regulation of BrFLC3 expression involves a complex interplay of mechanisms, resulting in less pronounced changes compared to the BrJMJ18-

overexpressing lines. Moreover, both WT *Par* and BrJMJ18^{Par}-OX plants exhibit consistent flowering time variations under HT conditions, but the latter experiences more significant changes. These findings suggested that in BrJMJ18^{Par}-OX plants, BrJMJ18^{Par} can adjust the expression of *BrFLC3* and flowering time, while the effect of *BrFLC3* is less straightforward in the WT *Par* plants. Furthermore, another distinction is that in BrJMJ18^{Par}-OX plants, the expression of BrJMJ18^{Par} is consistently persistent, while in WT *Par* plants, its expression is likely to be temporally and spatially specific. Therefore, variations in the observation window and sampling locations may influence the interpretation of the intrinsic scenario. Additionally, it is plausible that other downstream genes and pathways of BrJMJ18 contribute to *Par*'s thermotolerance regulation, potentially influencing the observation of the *in vivo* scenario.

In light of your feedback and that of the editor, we have revised our major conclusion regarding this section to more accurately the findings. Specifically, we have amended the statement as follows: "The overexpression of BrJMJ18^{Par} can adjust the expression of *BrFLC3* and flowering time, while the effect of *BrFLC3* is less than straightforward in the WT *Par* plants.". The comments above have been revised and included in the sections of "Results-Overexpression of BrJMJ18^{Par} moderates flowering by altering the expression of *BrFLC3*" and "Discussion-Functional divergence of BrFLCs". For more details, please refer to lines 442-446 and 694-716, respectively.

Furthermore, we have also updated the Introduction, Abstract and Results sections in line with the editor's suggestions, ensuring that certain claims are adjusted to accurately represent the experimental discoveries. Please see major changes in lines 16-28, 89-97, 166-167, 226, 319, 374, 390-391, 396-397, 400, 447-448, 468-469, and 485. We invite you to review these modifications (highlighted in yellow) and share any additional feedback you may have.